# LookHere: Vision Transformers with Directed Attention Generalize and Extrapolate

**Anthony Fuller, Daniel G. Kyrollos, Yousef Yassin, James R. Green**
Department of Systems and Computer Engineering
Carleton University
Ottawa, Ontario, Canada
*anthony.fuller@carleton.ca

## Abstract

High-resolution images offer more information about scenes that can improve model accuracy. However, the dominant model architecture in computer vision, the vision transformer (ViT), cannot effectively leverage larger images without finetuning — ViTs poorly extrapolate to more patches at test time, although transformers offer sequence length flexibility. We attribute this shortcoming to the current patch position encoding methods, which create a distribution shift when extrapolating.

We propose a drop-in replacement for the position encoding of plain ViTs that restricts attention heads to fixed fields of view, pointed in different directions, using 2D attention masks. Our novel method, called LookHere, provides translation-equivariance, ensures attention head diversity, and limits the distribution shift that attention heads face when extrapolating. We demonstrate that LookHere improves performance on classification (avg.$\uparrow 1.6\%$), against adversarial attack (avg.$\uparrow 5.4\%$), and decreases calibration error (avg.$\downarrow 1.5\%$) — on ImageNet *without* extrapolation. *With* extrapolation, LookHere outperforms the current SoTA position encoding method, 2D-RoPE, by $21.7\%$ on ImageNet when trained at $224^2$ px and tested at $1024^2$ px. Additionally, we release a high-resolution test set to improve the evaluation of high-resolution image classifiers, called ImageNet-HR.

## 1 Introduction

There is a decades-long trend in computer vision towards higher-resolution imagery, which contains more detailed scene information. Increasing resolution is a reliable way to improve model accuracy [13, 14, 15, 16, 17, 18, 19, 20], but this comes at a cost; training models for hundreds of epochs on large-scale datasets is expensive, especially at high-resolutions. There are two ways to reduce this cost and still see accuracy benefits from high-resolutions: ❶ high-resolution finetuning, which pretrains models at a lower resolution, like $224^2$ px, then finetunes them at a higher resolution, like $384^2$ px; and ❷ extrapolating, which deploys models at a higher resolution, without further training. Of these two options, we should aim for models that can *effectively extrapolate*, as it presents a zero-cost solution that does not require finetuning at every target resolution. Finetuning costs aside, improvements to extrapolation should benefit high-resolution finetuning since models that are better at extrapolating can adapt to higher resolutions more easily. Although extrapolation is a significant and exciting challenge, state-of-the-art (SoTA) model architectures extrapolate poorly.

Vision transformers (ViTs [9]) offer SoTA performance on many computer vision tasks. ViTs are simple; they split images into non-overlapping patches, linearly project pixels to form patch embeddings, and process these "tokens" with a stack of architecturally identical transformer layers — maintaining a constant feature map size throughout. This non-hierarchical design enables learning *patch* representations, which are useful for dense prediction tasks [21, 22, 23] and are fundamental

---

*AF, DGK, and YY made significant technical contributions. AF and DGK initiated the project. AF and JRG led the project. Code and data are available at: `https://github.com/GreenCUBIC/lookhere`

38th Conference on Neural Information Processing Systems (NeurIPS 2024).

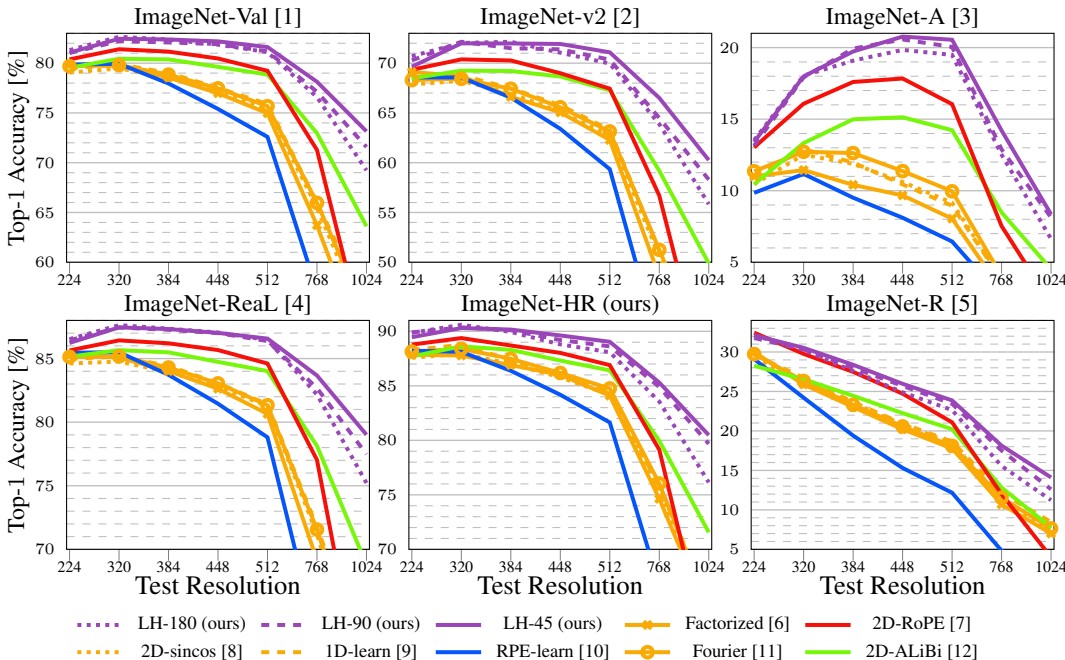

Figure 1: ViT-B/16 models trained for $150$ epochs on ImageNet at $224^2$ px and tested up to $1024^2$ px. Model architectures are consistent between runs other than *position encoding* methods. We perform an $8$-run hyperparameter sweep, per method, to ensure fair comparisons. Our three LookHere variants improve extrapolation ability, with more narrow fields of view performing best at $1024^2$.

for vision-language models [24, 25, 26]. The design enables efficient processing of only a subset of patches, known as token dropping [27, 28]. Lastly, it enables model scaling by increasing the embedding size and the layer count [29, 30].

Image-size extrapolation with ViTs can be achieved in three ways: ❶ increasing the patch size, which packs more pixels into each patch embedding; ❷ increasing the "patchification" stride, which skips-over pixels; and ❸ increasing the number of patches. Of these three options, we should aim for models that can effectively ingest more patches — called "sequence length extrapolation" in the natural language processing (NLP) community [31] — as a greater number of patches presents models with more (uncompressed) information that we hope to leverage into higher accuracy. Furthermore, methods that improve sequence length extrapolation, like our proposed method, can be fused with methods that adjust patch sizes, like FlexiViT [32]. We strongly believe that patch *position encoding* is a primary cause of the poor sequence length extrapolation ability of ViTs — like it is in NLP, where significant advancements have been made by improving position encoding [31, 33, 34, 35].

Adding learnable or fixed sinusoidal position embeddings to patch embeddings before the first layer is the most common way ViTs encode positions. Recently, the rotary position embeddings (RoPE [36]) used in SoTA language models [37, 38] were extended to ViTs, as 2D-RoPE [7], showing exciting results. RoPE is a different approach to position encoding that injects positional information in each self-attention layer by rotating queries and keys with fixed sinusoidal embeddings. But for these methods to ingest more patches at test time, they must either introduce new position embeddings or modify existing embeddings — both options create a significant distribution shift. Motivated by these observations and more, we make the following contributions:

❶ **LookHere** — We introduce a novel position encoding method for plain ViTs that restricts attention heads to fixed fields of view (FOV) and points them in different directions via 2D masks. This design provides: ⓐ translation-equivariance, ⓑ attention head diversity, ⓒ improved interpretability, and ⓓ limits the distribution shift that attention heads face when extrapolating.

❷ **Controlled Experiments** — We perform an apples-to-apples comparison between *seven* position encoding methods for plain ViTs alongside our three LookHere variants. We demonstrate that LookHere: ⓐ improves classification, segmentation, adversarial robustness, and model calibration

when tested *at* the training resolution; ❺ significantly improves performance when tested *beyond* the training resolution; and ❻ increases its performance advantage after high-resolution finetuning.

❸ **Extrapolation Insights** — We show that extrapolation: ❹ benefits images with small objects the most, as they occupy more patches at test time; ❺ produces class-level and dataset-level effects; and ❻ creates distribution shifts that can be visualized via attention maps.

❹ **ImageNet-HR** — We introduce the first natively high-resolution ImageNet test set ($1024^2$ px) aimed to benchmark classifiers on images that were not upsampled to achieve the target image size.

## 2 Background and Related Work

A ViT splits an image into a grid of non-overlapping patches, flattens the grid into a sequence, and flattens the patches into vectors; i.e., $\mathbb{R}^{Y \times X \times C} \to \mathbb{R}^{N_y \times N_x \times P^2 \times C} \to \mathbb{R}^{(N_y \cdot N_x) \times (P^2 \cdot C)}$, where $Y$ is the image-height, $X$ is the image-width, $C$ is the number of channels, $N_y$ is the grid-height, $N_x$ is the grid-width, $P$ is the patch height and width. A linear layer maps each vector of pixels to a patch embedding; i.e., $\mathbb{R}^{P^2 \cdot C} \to E_i^{patch} \in \mathbb{R}^D$, where $D$ is the embedding dimension also known as the transformer width. We define $i$ and $(i_y, i_x)$ as the sequence position and the 2D position of the $i^{\text{th}}$ patch, respectively, where $N$ is the total number of patches, equal to $N_y \cdot N_x$, $i \in \{1, 2, \ldots, N\}$, $i_y \in \{1, 2, \ldots, N_y\}$, and $i_x \in \{1, 2, \ldots, N_x\}$. Finally, sequence length extrapolation occurs when $N_{test} > N_{train}$.

A patch embedding represents the *content* of a patch, and contains no information representing its original location within the image. Thus, we must encode patch positions to enable spatial reasoning; otherwise, a ViT will operate on a bag of patches.

We define a "plain ViT" as attention-only and non-hierarchical. Our primary goal is to improve the extrapolation ability — i.e., generalize to more patches at test time — of plain ViTs. Our work is motivationally aligned with FlexiViT [32] and NaViT [6], improving the flexibility of plain ViTs. Next, we briefly describe seven position encoding methods and refer the reader to the cited studies for further details; we include them *all* in our controlled experiments. Another method, iRPE [39], is also compatible with plain ViTs. However, we exclude it because it is more than twice as slow as other methods; nonetheless, we benchmark iRPE with our best training recipe in Appendix A.2.1.

**Input Embeddings.** This group leverages learned or fixed position embeddings, $E_i^{pos} \in \mathbb{R}^D$, that are added to patch embeddings at the transformer input; i.e., $z_i = E_i^{patch} + E_i^{pos}$, where $z$ is the input to the first transformer layer. Position embeddings represent the absolute positions of patches in an image.

❶ 1D position embeddings [9] (**1D-learn** for short) map $i$ to learnable embeddings. ❷ 2D sinusoidal embeddings [8] (**2D-sincos** for short) individually map $i_y$ and $i_x$ to fixed 1D-sinusoidal embeddings ($E_i^y, E_i^x \in \mathbb{R}^{\frac{D}{2}}$), then concatenate them along the embedding dimension. ❸ Factorized position embeddings [6] (**Factorized** for short) individually map $i_y$ and $i_x$ to learnable embeddings ($E_i^y, E_i^x \in \mathbb{R}^D$), then add them. ❹ Learnable Fourier features [11] (**Fourier** for short) map $(i_y, i_x)$ to Fourier features [40, 41], then to embeddings with a multi-layer perceptron (MLP).

**Attention Biases.** This group leverages learned or fixed operations that encode positions by modifying the pairwise interactions between patches in self-attention *without* adding position embeddings to patch embeddings. Recall that self-attention first applies three separate linear transformations to project internal patch representations and splits the resultant vectors into $H$ smaller vectors of length $D_H$; i.e., $\mathbb{R}^{N \times D} \to \mathbb{R}^{3 \times N \times H \times D_H}$ — creating queries, keys, and values for each attention head. We denote a specific head by $h$. Next, attention scores ($A \in \mathbb{R}^{H \times N \times N}$) are calculated by measuring the similarity between all pairs of queries ($q_{hi} \in \mathbb{R}^{D_H}$) and keys ($k_{hj} \in \mathbb{R}^{D_H}$), separately, for each head; i.e., $a_{hij} = q_{hi} \cdot k_{hj} / \sqrt{D_H}$, where $i$ and $j$ are query and key sequence positions, and we define $(i_y, i_x)$ and $(j_y, j_x)$ as their 2D positions. Attention scores ($a_{hij}$) represent the *amount* of information moving from patch position $j$ to $i$ — whereas values ($v_{hj} \in \mathbb{R}^{D_H}$) represent the *content* of the moving information.

❺ Learnable relative position encoding [10] (**RPE-learn** for short) biases attention scores by mapping all possible relative positions between queries and keys to learnable embeddings ($B_{ij} \in \mathbb{R}^H$); i.e., biases are a function of $i_y - j_y$, $i_x - j_x$, and $h$. ❻ A 2D extension of Attention with Linear Biases

(ALiBi [31]), **2D-ALiBi** [12] penalizes attention scores as a function of the Euclidean distance between $(i_y, i_x)$ and $(j_y, j_x)$, and a head-specific scalar, called a slope. Slopes bias attention heads at different rates. ❼ A 2D extension of rotary position embeddings (RoPE [36]), **2D-RoPE** [7] rotates queries and keys as a function of their positions. Each query is rotated by the sinusoidal embedding of $i_y$ for half its dimensions and the sinusoidal embedding of $i_x$ for the other half of its dimensions; likewise, keys are rotated as a function of $j_y$ and $j_x$.

**Non-plain ViTs.** Many hybrid or hierarchical architectures have been invented that often encode positions differently [42, 43, 44, 45, 46, 47, 48, 49, 50, 51, 52, 53]. Although these architectures may be favored in some circumstances, the plain ViT is the most common single architecture due to its simplicity, flexibility, and scalability. We benchmark many non-plain ViTs and large SoTA ViTs on extrapolation in Appendix A.2.1.

**ViT Extrapolation.** Some ViTs have been tested at higher resolutions than they were trained [54, 43, 12, 55]. NaViT [6] benchmarked input embedding methods on extrapolation, none see the gains at higher resolutions that we observe.

## 3 LookHere

**Design Motivation.** We introduce 2D attention masks that assign each attention head a direction and a FOV, preventing attention outside the head's FOV. Within a head's FOV, attention scores are penalized based on relative patch distances. Three ideas motivate this design. ❶ Attention head diversity: heads often learn redundant algorithms that can be pruned with little accuracy penalty [56, 57, 58]. Head redundancy has also been observed in NLP [59, 60, 61], where diversity-encouraging loss functions have been leveraged to improve generalization [62, 63, 64, 65]. From a mechanistic point of view, we can think of attention heads as an ensemble of sub-networks that "operate completely in parallel, and each add their output back into the residual stream," [66] and the residual stream is mapped to logits. Diversity has long been a desirable property of ensembles [67, 68], and constraining attention heads to focus in different directions ensures it. ❷ Attention head consistency: heads often learn interpretable spatial algorithms, like "attend to the area above the query," which reliably retrieves information from the internal representations above the query; however, we believe these types of spatial algorithms might fail when new or modified position embeddings are introduced to encode *new* patch positions during extrapolation — misleading the model about the information above the query, for example. We believe hard-coding both directions and distances (via attention masks and biases) will reduce the need for models to learn their own spatial algorithms. ❸ Translation-equivariance has long been a desirable property of vision models, contributing to the success of convolutional networks [69, 70, 71]. ViTs are critiqued for weak inductive biases, leading to poor sample efficiency when trained from scratch [72, 73, 74]. We believe that LookHere's stronger inductive biases, achieved via directional masking and distance penalties, can improve ViT sample efficiency.

**Design Specifics.** Let $H$ be the number of heads, $L$ be the number of layers, and $N$ be the number of patches (plus one for the CLS token). We denote the LookHere matrices by $\mathcal{A}_{\text{FIX}} \in \mathbb{R}^{L \times H \times (N+1) \times (N+1)}$. We encode positions by subtracting the LookHere matrix for a layer $l$, $\mathcal{A}_{\text{FIX}}^l$, from the learned attention matrix, $\mathcal{A}_{\text{LRN}}^l = QK^T/\sqrt{D_H}$, before the softmax that normalizes the attention matrix prior to multiplying it by values [75], i.e., $\mathcal{A}^l = \texttt{softmax}(\mathcal{A}_{\text{LRN}}^l - \mathcal{A}_{\text{FIX}}^l)$. We do not add position embeddings to patch embeddings.

Let $i$ and $j$ be query and key sequence positions, respectively, with 2D-coordinates $(i_y, i_x)$ and $(j_y, j_x)$. Crucially, $j$ is visible to $i$ if $j$ lies within $i$'s FOV. This attention masking technique is inspired by the 1D causal masks used in autoregressive transformer decoders used in NLP [75]. When $j$ is visible, we bias the attention score based on the Euclidean distance between $i$ and $j$ to encode the relative distance between patches. We scale distances via a slope function $m : \mathbb{N}_L \times \mathbb{N}_H \to \mathbb{R}$, $m(l, h) = s_l(l) \cdot s_h(h) \cdot s_g$ that strengthens or weakens the distance penalty as a function of the head ($s_h : \mathbb{N}_H \to \mathbb{R}$) and layer ($s_l : \mathbb{N}_L \to \mathbb{R}$), scaled by a global slope $s_g \in \mathbb{R}$. Finally, the CLS token is visible to all positions.

$$\text{LookHere}(l, h, i, j) = \begin{cases} m(l, h) \cdot \text{Distance}(i, j) & \text{if } j \text{ is visible to } i \\ \infty & \text{otherwise} \end{cases} \tag{1}$$

$$\text{Distance}(i, j) = \sqrt{(i_y - j_y)^2 + (i_x - j_x)^2} \tag{2}$$

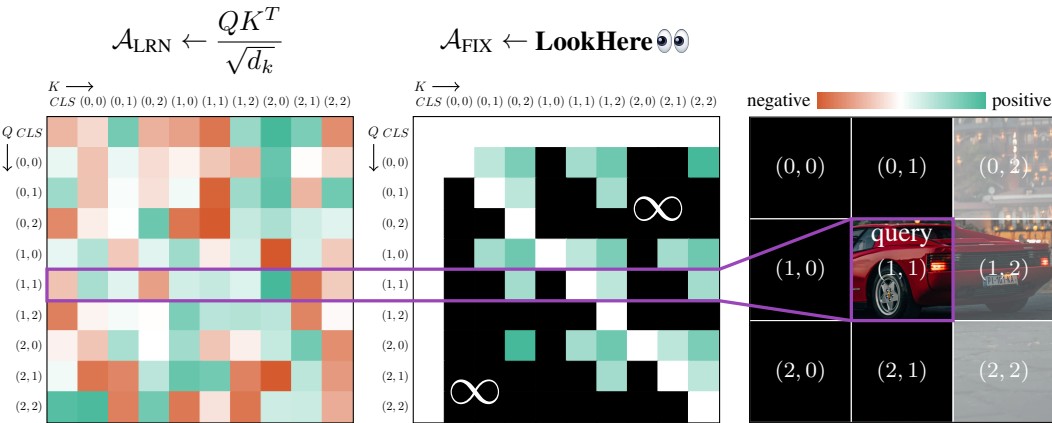

Figure 2: LookHere masks and biases (center) the learned attention matrix (left, where colors are random). Masked cells are **black**, encoding directions ($\rightarrow$ with a 90° FOV); biased cells are shaded **bluish-green**, encoding relative patch distances. (Right) An example of the FOV of the center query patch. The final attention matrix is computed as $\mathcal{A}^l = \texttt{softmax}(\mathcal{A}^l_{\text{LRN}} - \mathcal{A}^l_{\text{FIX}})$, at each layer $l$.

For example, Figure 2 displays attention matrices of a head that "looks right" with a 90° FOV. We create three LookHere variants, the first two have FOVs of 180° and 90° (**LH-180** and **LH-90**). We direct attention heads eight different ways, selecting the four cardinal directions ($\uparrow, \downarrow, \leftarrow, \rightarrow$) and the four intercardinal directions ($\nearrow, \searrow, \swarrow, \nwarrow$). ViT-B models have twelve attention heads; we leave the last four attention heads undirected to allow them unrestricted attention over the full image. We create a final variant that cuts the first four LH-90 masks in two, creating eight 45° views that cover the full image without overlapping (**LH-45**). Visualizations of the bias matrices are in Appendix A.3.

**Design Ablations.** We offer four takeaways through extensive ablations (Appendix A.6): ❶ LookHere is robust to the choice of slope function. We set our default $s_l$ to linearly decrease from 1.5 to 0.5 with increasing depth (inspired by depth-wise attention distance findings [76]). This helped in preliminary experiments, but the benefits disappear in our ablations. We arbitrarily set our default $s_h$ to $(\frac{1}{2}, \frac{1}{8}, \frac{1}{32}, \frac{1}{128})$ for the four undirected heads, but distance penalties on undirected heads can be removed entirely. We set $s_g = 1$; LookHere is also robust to the choice of the global slope. We believe precisely tuning slopes is unnecessary because models can learn to scale attention logit magnitudes. ❷ Increasing penalties with the square or square root of the distance harms extrapolation. ❸ Removing all distance penalties harms extrapolation. ❹ Our main contribution, 2D directional masks, are crucial to retain performance, but our method is robust to *many* directional configurations.

**Compute.** $\mathcal{A}_{\text{FIX}}$ is precomputed and fixed, subtracting it element-wise from the learned attention matrices $\mathcal{A}_{\text{LRN}}$ only costs $H \cdot (N + 1) \cdot (N + 1)$ floating point operations (FLOPs) per layer. For a ViT-B/16 model, these subtractions account for 0.016% of the total FLOPs. LookHere reduces FLOPs by *not* adding position embeddings to patch embeddings, but this amount is also negligible. Additionally, LookHere matrices offer structured sparsity (up to 7/8 for a 45° FOV) that can speedup attention — although exciting, this speedup requires custom kernels that we leave for future work.

## 4 Experiments

Deep neural networks — including ViTs — can be sensitive to seemingly minor hyperparameter changes when trained from scratch. Dosovitskiy et al. [9] finetuned the original ViT at a higher resolution, reaching 77.9% top-1 accuracy on ImageNet (we refer to ILSVRC2012 or ImageNet-1k as ImageNet). Steiner et al. [77] searched 28 hyperparameter configurations, achieving best and average runs of 80.0% and 76.9%, respectively (average calculation omits runs without data augmentation, as they were poor). Touvron et al. [78] ablated repeat augmentation [79], dropping accuracy by 4.8%. Touvron et al. [17] replaced cross-entropy loss with binary cross-entropy loss, raising accuracy by 1.3%. Importantly, these are all ViT-B/16 models trained from scratch for 300 epochs on ImageNet. Informed by these observations and more, we design a controlled experiment: We search 8 hyperparameter configurations for *each* position encoding method using a single codebase; this offers an apples-to-apples comparison between our three LookHere variants and seven baselines.

### 4.1 Setup

Our 80 training runs result from the following Cartesian product:

**Position encoding:** 1D-learn, 2D-sincos, Factorized, Fourier, RPE-learn, 2D-ALiBi, 2D-RoPE, LH-180, LH-90, LH-45

**Augmentations:** RandAugment(2, 15) [80], 3-Augment [17]
**Learning rate:** $1.5 \cdot 10^{-3}$, $3.0 \cdot 10^{-3}$
**Weight decay:** 0.02, 0.05

For each configuration, we train a ViT-B/16 on 99% of the ImageNet training set, holding the last 1% as a validation set called "minival", following [77, 81] (see Appendix A.4.1 for other hyperparameters). We train all models from scratch for 150 epochs on $224^2$ px images. Our results are competitive and sometimes surpass ViTs trained for much longer, which validates our setup. The best models (according to minival accuracy), among our 8-run hyperparameter sweep per method, are always trained using 3-Augment [17], a $3.0 \cdot 10^{-3}$ learning rate, and a 0.05 weight decay.

**Test sets.** We test all 80 models on six ImageNet test sets. This includes ❶ the original "validation" set used as a test set (Val for short [1]), ❷ the reassessed labels of the original validation set (ReaL for short [4]), ❸ the independently collected and in-distribution test set (v2 for short [2]), ❹ the natural adversarial test set (-A for short [3]), ❺ the ImageNet rendition test set (-R for short [5]), and ❻ the high-resolution test set that we introduce (-HR for short).

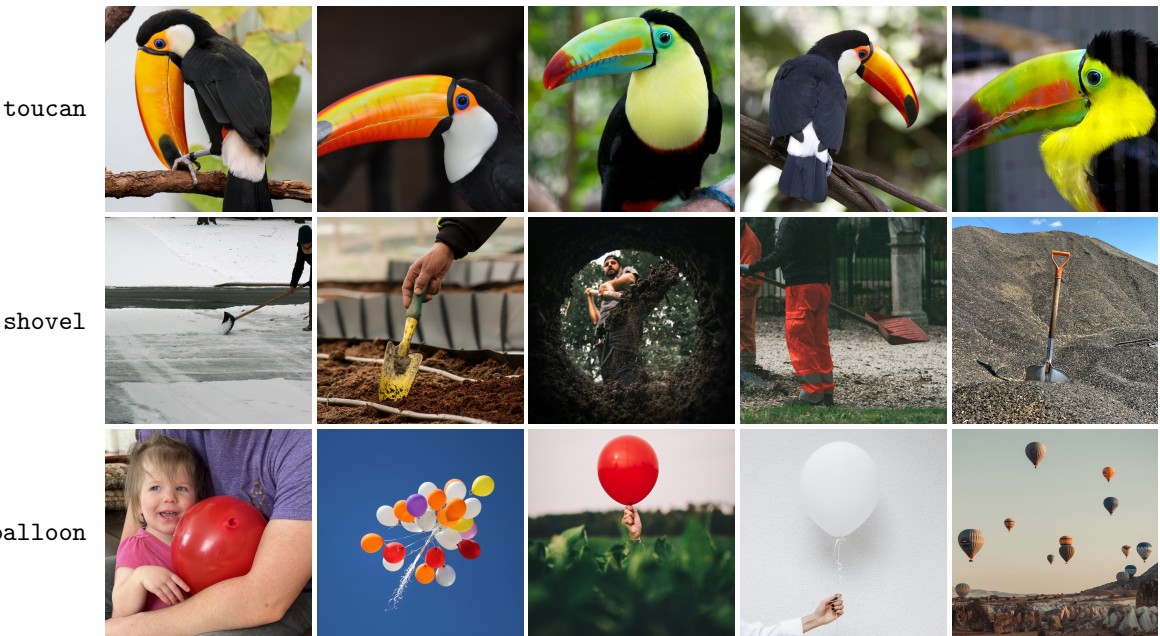

Figure 3: Images of three classes from ImageNet-HR. (Bottom left is Anthony's niece Addison.)

**ImageNet-HR.** Since there are no natively high-resolution ImageNet test sets, there are two options to test the extrapolation ability of models trained on ImageNet: ❶ upsample existing test sets to higher resolutions, and ❷ collect a high-resolution test set ourselves. However, upsampling low-resolution images introduces another distribution shift (i.e., interpolated pixels) that we may not want to test. Thus, we collect a high-resolution test set to remove this confounding variable from our analysis. We manually collect 5 images for each ImageNet class, resulting in 5k total images, and manually crop them to $1024^2$ px. This is smaller than other test sets (v2 is 30k images, -A is 7.5k images). However, we invest considerable resources to ensure its quality with two priorities: annotation accuracy and image diversity. See Appendix A.1 for details. ImageNet-HR can be accessed: `https://huggingface.co/datasets/antofuller/ImageNet-HR`

**Adversarial Attacks.** We perform Fast Gradient Sign Method (FGSM [82]) adversarial attacks with two strengths ($\frac{1}{255}$, $\frac{3}{255}$) on all models using Val images.

**Calibration Estimates.** We calculate the Expected Calibration Error (ECE [83]) with 15 bins of all models using Val images.

**Higher-Resolution Finetuning.** With the best model per method, we continue training on ImageNet for 5 epochs at $384^2$ px. We test at $384^2$ px without extrapolating.

**Segmentation.** With the best model per method, we finetune following the Segmenter protocol with a linear decoder [84]. Additionally, we probe the patches by only training a linear layer to produce a low-resolution logit map which is upsampled to obtain a full resolution segmentation map, following [85]. We run these experiments on ADE20k [86] at $512^2$ px and Cityscapes [87] at $768^2$ px.

**Patch Logit-lens.** Inspired by interpretability research [88], we evaluate the quality of the learned patch representations for models leveraging LookHere compared with other methods. Following prior work [89, 90], we project frozen patch representations onto the learned class embedding space using the MLP classifier head that was learned for the CLS token. We leverage the ImageNet-S dataset [91], which contains partial segmentation maps for 12k images from Val, covering 919 ImageNet classes.

**Extrapolating.** With the best model per method, we test on images larger than $224^2$ px, increasing the number of patches and we test on images smaller than $224^2$ px, decreasing the number of patches; for both experiments, no further training is performed — the models are tested on their resolution generalization ability. For 1D-learn and 2D-sincos, we bilinearly interpolate the position embeddings used during training. For Factorized, we linearly interpolate the position embeddings for each axis. Fourier does not require adjustment since fractional positions along each axis are used as input. For RPE-learn, we interpolate the learned relative biases using the official BEiT implementation [10]. 2D-ALiBi does not require adjustment either. However, we tune a parameter on minival that scales the distance penalty at each test resolution. For 2D-RoPE, we tune its base frequency on minival — this is a SoTA method to extrapolate RoPE used in NLP [33]. Lastly for LookHere, we tune the global slope on minival. The benefits of tuning slopes are minimal, see Appendix A.4.4.

## 4.2 Results and Analysis

Table 1: Top-1 acc. (%) for ViT-B models trained on ImageNet for 150 epochs; trained and tested at $224^2$. We report the best and average results across our 8-run hyper-parameter sweep.

| Method | Val [1] | | ReaL [4] | | v2 [2] | | -A [3] | | -R [5] | | -HR (ours) | |
|---|---|---|---|---|---|---|---|---|---|---|---|---|
| | Best | Avg. | Best | Avg. | Best | Avg. | Best | Avg. | Best | Avg. | Best | Avg. |
| 1D-learn | 79.45 | 77.35 | 84.97 | 82.87 | 68.49 | 65.17 | 10.97 | 7.58 | 29.64 | 25.73 | 88.28 | 85.22 |
| 2D-sincos | 79.05 | 77.44 | 84.62 | 82.96 | 67.86 | 65.31 | 10.45 | 7.76 | 29.11 | 26.07 | 87.58 | 85.36 |
| Factorized | 79.86 | 77.29 | 85.30 | 82.99 | 69.11 | 65.34 | 11.00 | 7.16 | 29.99 | 26.18 | 87.86 | 85.37 |
| Fourier | 79.69 | 77.37 | 85.13 | 82.89 | 68.30 | 65.33 | 11.36 | 7.79 | 29.73 | 24.62 | 88.14 | 85.39 |
| RPE-learn | 79.86 | 77.26 | 85.46 | 82.88 | 68.57 | 65.19 | 9.85 | 7.18 | 29.10 | 24.62 | 88.22 | 85.17 |
| 2D-ALiBi | 79.54 | 77.29 | 85.15 | 82.92 | 68.47 | 65.15 | 10.45 | 7.27 | 28.26 | 24.41 | 87.70 | 85.13 |
| 2D-RoPE | 80.38 | 78.37 | 85.64 | 83.78 | 69.34 | 66.56 | 13.03 | 8.84 | 32.45 | 28.55 | 88.78 | 86.35 |
| **LH-180** | 81.31 | 80.01 | 86.53 | 85.30 | 70.70 | 68.52 | 13.53 | 10.45 | 32.10 | 28.94 | 89.86 | 87.80 |
| **LH-90** | 81.02 | 79.89 | 86.44 | 85.28 | 70.28 | 68.54 | 13.15 | 10.80 | 31.77 | 29.47 | 89.90 | 87.86 |
| **LH-45** | 81.06 | 79.74 | 86.23 | 85.07 | 69.65 | 68.18 | 13.41 | 10.21 | 32.12 | 29.51 | 89.46 | 87.43 |

LookHere improves ViT sample efficiency (Table 1). Our three variants outperform the best baseline, 2D-RoPE, under almost all test conditions (the single exception being the best 2D-RoPE model on -R). LookHere further improves gains when considering averaged results — i.e., when accuracy values are averaged over 8 hyperparameter configurations (please see the Appendix A.5 for individual results). For instance, LH-180 outperforms 2D-RoPE by 0.93% / 1.36% on on Val / v2 on our best runs and by 1.64% / 1.96% on Val / v2 on our averaged runs — indicating that LookHere decreases hyperparameter sensitivity. Surprisingly, LH-180 *averages* 80.01% on Val, which matches the *best* run trained for twice as long by Steiner et al. [77].

LookHere improves ViT adversarial robustness and model calibration (Tables 2 3); both have been linked to ensemble diversity [92, 93, 94], which we offer as a potential explanation. This is an interesting finding because adversarial robustness and calibration can be at odds with accuracy [95, 96]. We show that LookHere learns more diverse attention heads by measuring the generalized Jensen-Shannon divergence [97] between heads (Figure 4). In the Appendix A.8, we measure more properties of models leveraging different position encoding methods. LookHere significantly outperforms other methods on segmentation linear probing, demonstrating its ability to learn spatially-

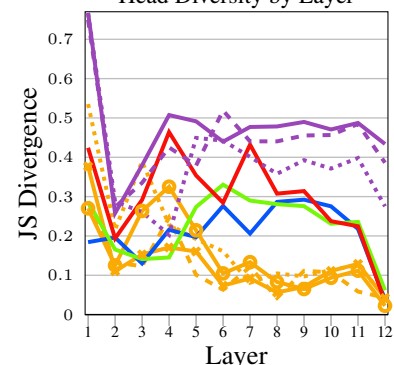

Figure 4: **LookHere** learns more diverse attention heads and prevents attention collapse. Legend follows Figures 1 7.

Table 2: Fast Gradient Sign Method attack [82] (% top-5 acc. on Val), best and average runs.

| Method | $\epsilon = 1/255$ Best | Avg. | $\epsilon = 3/255$ Best | Avg. |
|---|---|---|---|---|
| 1D-learn | 58.87 | 54.36 | 44.23 | 41.37 |
| 2D-sincos | 60.38 | 55.16 | 45.37 | 41.61 |
| Factorized | 60.86 | 56.19 | 46.34 | 42.32 |
| Fourier | 59.91 | 54.74 | 44.99 | 41.90 |
| RPE-learn | 59.81 | 53.36 | 45.04 | 40.19 |
| 2D-ALiBi | 58.07 | 53.68 | 43.32 | 40.30 |
| 2D-RoPE | 60.59 | 57.16 | 47.11 | 43.77 |
| **LH-180** | 65.06 | 62.59 | 51.81 | 49.06 |
| **LH-90** | 63.89 | 61.88 | 50.87 | 48.07 |
| **LH-45** | 64.71 | 61.71 | 50.21 | 47.86 |

Table 3: Expected Calibration Error % [83] (↓) on Val, best and average runs.

| Best | Avg. |
|---|---|
| 10.13 | 12.21 |
| 10.14 | 11.85 |
| 10.01 | 11.37 |
| 9.65 | 12.13 |
| 8.66 | 11.42 |
| 9.26 | 11.24 |
| 9.60 | 11.48 |
| 8.28 | 9.76 |
| 8.68 | 9.91 |
| 8.87 | 9.99 |

Table 4: Semantic Segmentation (% mIoU), linear probing (LP) and finetuning (FT).

| | ADE20k LP | FT | Cityscapes LP | FT |
|---|---|---|---|---|
| | 29.5 | 38.05 | 47.1 | 72.93 |
| | 29.2 | 38.39 | 45.3 | 72.91 |
| | 29.4 | 37.95 | 45.9 | 72.51 |
| | 29.8 | 38.26 | 46.2 | 73.60 |
| | 26.4 | 37.25 | 42.9 | 73.87 |
| | 26.2 | 37.56 | 48.4 | 73.92 |
| | 29.9 | 39.74 | 47.0 | 75.53 |
| | 32.4 | 40.29 | 55.0 | 75.05 |
| | 32.6 | 40.60 | 55.3 | 74.90 |
| | 32.7 | 40.07 | 55.5 | 74.42 |

aware patch representations. LookHere also performs well with segmentation finetuning, achieving comparable performance to 2D-RoPE (Table 4).

High-resolution finetuning increases the performance advantage of all three LookHere variants over 2D-RoPE (Table 5). This aligns with our intuition that improving extrapolation methods can improve high-resolution finetuning. Lower initial finetuning loss has been linked to better retaining the general representations learned during pretraining [98], and better extrapolating models have lower initial loss at a higher-resolution, by definition.

Using a "logit lens" [88] approach, we project patch representations onto the class embedding space [89]. We observe that LookHere encodes semantic information in its patches faithful to

Table 5: Top-1 acc. (%) for models trained at $224^2$ px, finetuned and tested at $384^2$ px.

| Method | Val | ReaL | -v2 | -A | -R | -HR |
|---|---|---|---|---|---|---|
| 1D-learn | 81.46 | 86.46 | 70.69 | 18.80 | 29.80 | 89.82 |
| 2D-sincos | 81.33 | 86.50 | 70.53 | 17.73 | 29.26 | 89.62 |
| Factorized | 81.50 | 86.62 | 70.95 | 18.05 | 29.98 | 89.50 |
| Fourier | 81.71 | 86.73 | 71.01 | 19.73 | 29.68 | 89.90 |
| RPE-learn | 82.01 | 87.17 | 71.66 | 18.13 | 29.53 | 90.20 |
| 2D-ALiBi | 81.41 | 86.73 | 70.50 | 18.01 | 28.60 | 89.46 |
| 2D-RoPE | 82.31 | 87.21 | 71.82 | 21.68 | 33.38 | 89.92 |
| **LH-180** | 83.28 | 88.05 | 73.12 | 22.85 | 32.95 | 91.38 |
| **LH-90** | 83.08 | 87.99 | 72.99 | 23.51 | 32.63 | 91.24 |
| **LH-45** | 83.10 | 87.83 | 72.43 | 22.39 | 33.10 | 90.92 |

the original patch location; these patch-level predictions act as a segmentation map that can be generated without additional training. The officer in Figure 5 is not a one-off example; using ImageNet-S [91], we see that LookHere outperforms 2D-RoPE by at least 22% mIoU using this patch-projection method (Figure 5). Our best explanation is that, by restricting attention, LookHere

1D-learn (4.1%)   2D-sincos (2.1%)   Factorized (8.7%)   Fourier (3.3%)   RPE-learn (6.1%)

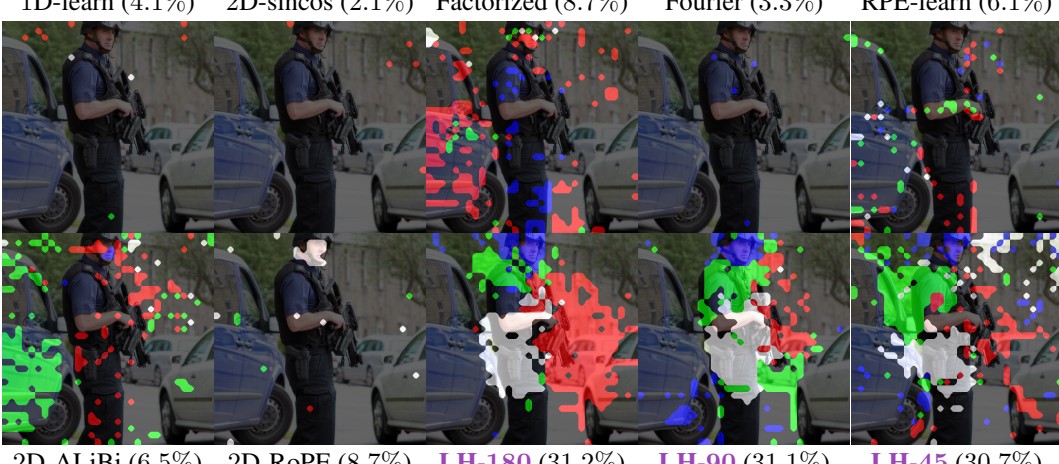

2D-ALiBi (6.5%)   2D-RoPE (8.7%)   **LH-180** (31.2%)   **LH-90** (31.1%)   **LH-45** (30.7%)

Figure 5: We apply frozen MLP classifying heads (learned on the CLS token) on frozen patch representations. We visualize ImageNet class predictions: `assault rifle` (**red**), `bulletproof vest` (**green**), `crash helmet` (**blue**), and `holster` (white). In parentheses, we show mIoU results (@224px) on ImageNet-S [91], where we apply this technique to segment images *without* training.

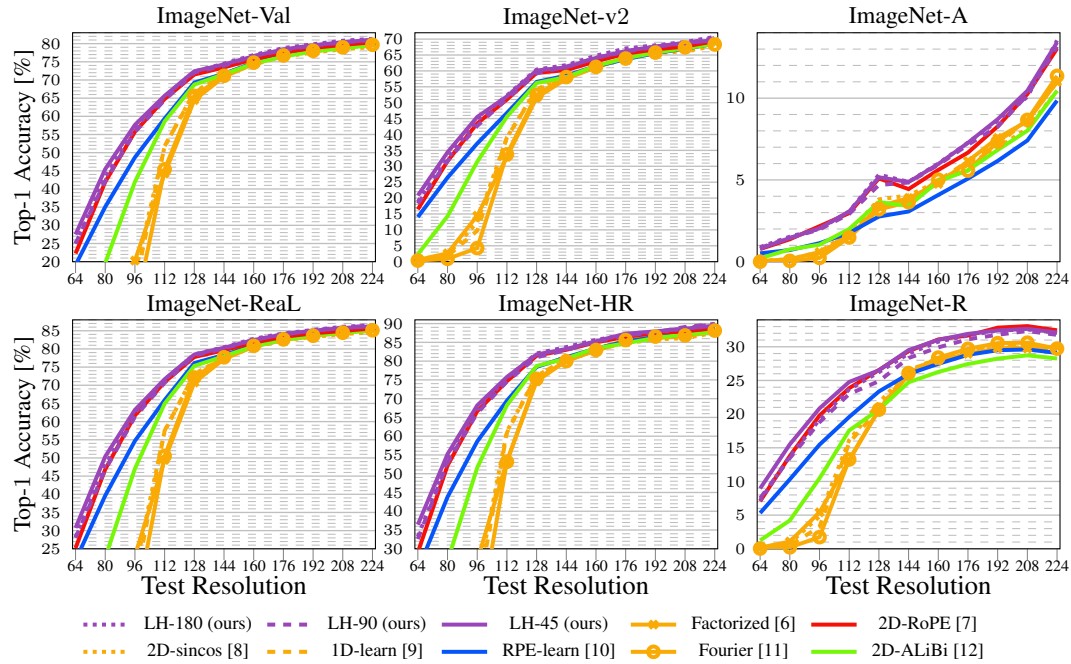

Figure 6: ViT-B/16 models trained for $150$ epochs on ImageNet at $224^2$ px and tested down to $64^2$ px. Model architectures are consistent between runs other than *position encoding* methods.

prevents the attention collapse at deeper layers observed in Figure 4 that divorces patch representations from their original patch locations; this collapse has been observed in other ViTs [99, 100]. We also expect that preventing attention collapse will benefit vision-language models, where frozen patch representations are used as "image tokens" that *should* represent their original patch locations [24, 25, 26]. More examples and detailed analysis are in Appendix A.7

LookHere significantly improves extrapolation ability (Figure 1). Our smallest FOV variant (LH-45) sees improving relative performance as resolution increases. LH-45 outperforms 2D-ALiBi, which is equivalent to LookHere without our 2D directional masks, by $9.5\%$ on Val at $1024^2$ px. These two results demonstrate the extrapolation benefits of restricting attention to fixed FOVs. LH-45 gains $1.3\%$ on Val when extrapolating from $224^2$ to $384^2$ px; this is the largest gain we find in the literature, including our extensive benchmarking of SoTA models in Appendix A.2. LookHere also outperforms other methods when tested on *smaller* images, but the advantage narrows (Figure 6).

Interestingly, smaller *objects* benefit most from extrapolation (Figure 7), which are distributed over more patches at test time. We believe this effect also explains the $6-8\%$ that LookHere models gain when extrapolating on ImageNet-A from $224^2$ to $448^2$ px; by inspection, ImageNet-A seems to have small objects, and other work found zooming-in on center-cropped ImageNet-A images improves

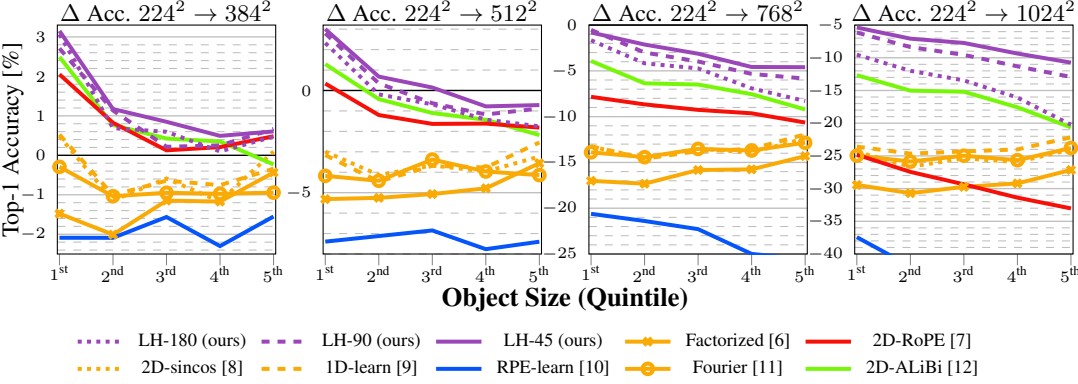

Figure 7: The effect of object size on accuracy gains or losses due to extrapolation. Object size is measured using annotations from Kaggle's ImageNet Object Localization Challenge [101].

performance [102]. Finally, all LookHere variants outperform other methods on ImageNet-HR, indicating better handling of interpolated pixels generated when upsampling lower-resolution imagery is *not* the reason why LookHere extrapolates better.

Reducing the distribution shift faced by attention heads during extrapolation is our best explanation for LookHere's large relative improvement. Figure 8 shows attention maps that are "unflattened" to visualize the image regions to which heads attend, averaged over the same 5k images. We show one head per model that exhibits similar behavior at a $224^2$ resolution. Models leveraging RPE-learn and 2D-ALiBi learn variants of an algorithm that retrieve information from above the query; however, both models retrieve information elsewhere in the image when extrapolating. LookHere hard-codes this type of algorithm, which it continues to execute when extrapolating. In Appendix A.8 we find more examples of interesting attention head behaviour.

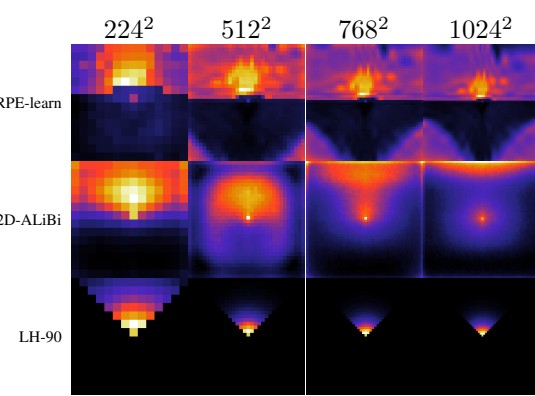

Figure 8: Attention maps of three attention heads across four resolutions, where the query is in the center. We use the colormap:

Extrapolation affects different datasets differently; it also affects different classes differently. For example, when extrapolating, all models underpredict certain classes (`bakery`, `church`, and `tights`) and overpredict other classes (`mobile home`, `threshing machine`, and `sports car`). This investigation is inspired by the class-level effects of data augmentation [103]. In Appendix A.9 we find more class-level effects of extrapolation.

# 5 Closing

**Limitations**. The primary limitation of LookHere is it requires hand-designed directional masks and distance penalties. However, our extensive ablations demonstrate that LookHere is robust to the choice of directional masks and distance penalties. The primary limitation of our experiments is we do not scale ViTs to giant sizes. Instead, we select the most common size, the ViT-B/16, and focus our computational resources on a controlled experiment — that extensively and fairly tunes the appropriate baselines for plain ViTs; this allows us to make confident conclusions based on our thorough experiments.

**Conclusion.** LookHere position encoding significantly improves the ability of plain ViTs to make inferences when provided a greater number of patches than seen during training. We thoroughly demonstrate that LookHere outperforms other methods with and without extrapolation on standard image benchmarks and our high-resolution ImageNet test set called ImageNet-HR. We provide new insights into ViT extrapolation by showing object-size, class-level, and dataset-level effects. We believe LookHere will help the vision community transform higher-resolution into higher accuracy.

**Future Work.** We are excited to realize the computational gains that LookHere makes available via sparse attention kernels, as well as bring LookHere to video and 3D point-cloud applications.

**Acknowledgments.** Anthony thanks NSERC's Postgraduate Scholarships Doctoral program for funding his PhD.

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

# A  Appendix / supplemental material

## A.1  ImageNet-HR

We invest considerable resources to ensure ImageNet-HR's quality with two priorities. ❶ Annotation accuracy — we only include images for which we are confident of their label; we achieve this by: ⓐ 5 rounds of quality control consisting of manually reviewing all cases where models disagreed with our annotations, using a SoTA model (`eva02_large_patch14_448.mim_m38m_ft_in22k_in1k` from timm [104]) and a weaker model that disagrees more often (`tiny_vit_5m_224.dist_in22k_ft_in1k` from timm [104]), ⓑ consulting someone with wildlife expertise to limit the annotation errors made by other test sets [105], ⓒ using multiple labels where necessary, for example, combining the "sunglass" and "sunglasses" classes, and labeling a "tusker" as also an "Asian elephant," if the image of the tusked animal is an Asian elephant. ❷ Image diversity — when collecting images, we try to maximize the diversity of images belonging to a class. Models achieve high accuracy on ImageNet-HR, likely due to less label ambiguity than other ImageNet test sets. Finally, we manually crop all images to $1024^2$ px, resulting in the first natively high-resolution ImageNet test set.

We collect the vast majority of images from flickr and Unsplash. Unsplash images "are made to be used freely" for commercial and non-commercial uses. flickr images were selected from the "All creative commons" license option. However, for some classes, we could not find enough open-access high-resolution images like "oil filter" or "hand or block plane," so we used Google search to find more. We estimate that around 50 of 5k images were not collected on flickr or Unsplash. Nine images were taken by an author or his family, with consent of everyone involved.

## A.2 Extrapolation Results

Table 6: Extrapolation results for ViT-B models trained on ImageNet for $150$ epochs; trained at $224^2$ and tested at various resolutions.

| Method | Res | Val [1] | | ReaL [4] | | v2 [2] | | -A [3] | | -R [5] | | -HR (ours) | |
|---|---|---|---|---|---|---|---|---|---|---|---|---|---|
| | | top-1 | top-5 | top-1 | top-5 | top-1 | top-5 | top-1 | top-5 | top-1 | top-5 | top-1 | top-5 |
| 1D-learn | $320^2$ | 79.89 | 94.59 | 85.17 | 96.26 | 68.78 | 87.88 | 12.91 | 34.36 | 26.73 | 40.04 | 88.76 | 96.94 |
| 2D-sincos | $320^2$ | 79.48 | 94.50 | 84.76 | 96.34 | 68.23 | 87.69 | 12.52 | 33.59 | 26.30 | 39.84 | 87.94 | 96.76 |
| Factorized | $320^2$ | 79.73 | 94.70 | 85.08 | 96.37 | 68.71 | 87.83 | 11.44 | 32.01 | 25.89 | 39.33 | 87.78 | 97.08 |
| Fourier | $320^2$ | 79.80 | 94.58 | 85.17 | 96.45 | 68.43 | 87.95 | 12.72 | 34.16 | 26.33 | 39.64 | 88.40 | 97.08 |
| RPE-learn | $320^2$ | 79.94 | 94.73 | 85.50 | 96.63 | 68.56 | 87.83 | 11.17 | 30.51 | 24.21 | 36.71 | 88.12 | 97.26 |
| 2D-ALiBi | $320^2$ | 80.44 | 95.10 | 85.66 | 96.66 | 69.25 | 88.65 | 13.33 | 35.00 | 26.54 | 40.47 | 88.62 | 97.36 |
| 2D-RoPE | $320^2$ | 81.40 | 95.36 | 86.43 | 96.77 | 70.38 | 88.94 | 16.08 | 39.52 | 29.76 | 44.21 | 89.38 | 97.38 |
| LH-180 | $320^2$ | 82.65 | 95.88 | 87.63 | 97.28 | 72.03 | 89.78 | 18.04 | 40.28 | 30.26 | 43.01 | 90.60 | 97.98 |
| LH-90 | $320^2$ | 82.22 | 95.66 | 87.44 | 97.13 | 72.15 | 89.62 | 17.92 | 40.28 | 30.33 | 43.34 | 90.36 | 97.94 |
| LH-45 | $320^2$ | 82.45 | 95.84 | 87.45 | 97.17 | 71.99 | 89.92 | 17.99 | 40.96 | 30.53 | 43.31 | 90.30 | 97.66 |
| 1D-learn | $384^2$ | 79.02 | 94.14 | 84.42 | 95.99 | 67.47 | 87.13 | 11.97 | 32.23 | 23.66 | 36.93 | 87.36 | 96.74 |
| 2D-sincos | $384^2$ | 78.56 | 94.06 | 83.95 | 96.02 | 66.96 | 87.16 | 11.87 | 31.31 | 23.53 | 36.63 | 87.02 | 96.38 |
| Factorized | $384^2$ | 78.56 | 94.06 | 84.01 | 95.94 | 66.62 | 87.21 | 10.41 | 29.72 | 23.03 | 35.69 | 86.90 | 97.00 |
| Fourier | $384^2$ | 78.85 | 94.07 | 84.30 | 96.07 | 67.43 | 87.33 | 12.63 | 32.59 | 23.31 | 35.95 | 87.48 | 96.84 |
| RPE-learn | $384^2$ | 77.96 | 93.64 | 83.72 | 95.85 | 66.54 | 86.51 | 9.51 | 26.85 | 19.41 | 30.38 | 86.38 | 96.44 |
| 2D-ALiBi | $384^2$ | 80.38 | 94.93 | 85.49 | 96.51 | 69.21 | 88.34 | 14.99 | 37.88 | 24.45 | 38.02 | 88.30 | 97.24 |
| 2D-RoPE | $384^2$ | 81.16 | 95.27 | 86.20 | 96.71 | 70.27 | 88.91 | 17.60 | 41.16 | 27.48 | 41.05 | 88.70 | 97.44 |
| LH-180 | $384^2$ | 82.38 | 95.79 | 87.35 | 97.25 | 72.15 | 89.67 | 19.09 | 42.32 | 27.58 | 39.97 | 89.94 | 97.84 |
| LH-90 | $384^2$ | 82.08 | 95.70 | 87.26 | 97.18 | 71.50 | 89.74 | 19.93 | 42.68 | 27.99 | 40.58 | 90.10 | 97.66 |
| LH-45 | $384^2$ | 82.38 | 95.85 | 87.32 | 97.15 | 71.98 | 90.00 | 19.73 | 43.29 | 28.38 | 40.95 | 90.16 | 97.76 |
| 1D-learn | $448^2$ | 77.52 | 93.45 | 83.06 | 95.47 | 65.77 | 86.10 | 10.44 | 28.79 | 20.93 | 33.86 | 86.04 | 96.36 |
| 2D-sincos | $448^2$ | 77.12 | 93.54 | 82.80 | 95.52 | 65.11 | 85.62 | 10.61 | 29.32 | 20.84 | 33.66 | 85.78 | 96.24 |
| Factorized | $448^2$ | 76.98 | 93.33 | 82.63 | 95.32 | 65.06 | 85.70 | 9.67 | 26.63 | 20.21 | 32.50 | 86.00 | 96.46 |
| Fourier | $448^2$ | 77.47 | 93.46 | 83.05 | 95.51 | 65.57 | 86.05 | 11.37 | 29.59 | 20.57 | 32.76 | 86.18 | 96.46 |
| RPE-learn | $448^2$ | 75.40 | 92.41 | 81.45 | 94.87 | 63.40 | 84.29 | 8.11 | 23.79 | 15.31 | 25.31 | 84.18 | 95.44 |
| 2D-ALiBi | $448^2$ | 79.63 | 94.61 | 84.74 | 96.32 | 68.66 | 87.86 | 15.13 | 37.79 | 22.25 | 35.13 | 87.32 | 96.96 |
| 2D-RoPE | $448^2$ | 80.47 | 94.92 | 85.67 | 96.47 | 68.99 | 88.27 | 17.84 | 41.92 | 24.72 | 37.36 | 88.02 | 97.20 |
| LH-180 | $448^2$ | 81.86 | 95.56 | 87.05 | 97.16 | 71.05 | 89.62 | 19.84 | 43.35 | 24.90 | 36.83 | 88.80 | 97.74 |
| LH-90 | $448^2$ | 81.91 | 95.54 | 87.00 | 97.08 | 71.39 | 89.49 | 20.57 | 43.39 | 25.71 | 37.70 | 89.20 | 97.52 |
| LH-45 | $448^2$ | 82.19 | 95.67 | 87.02 | 97.08 | 71.93 | 89.58 | 20.77 | 43.76 | 25.97 | 38.35 | 89.62 | 97.66 |
| 1D-learn | $512^2$ | 75.89 | 92.54 | 81.41 | 94.79 | 63.31 | 84.37 | 8.95 | 25.65 | 18.65 | 31.03 | 84.44 | 96.02 |
| 2D-sincos | $512^2$ | 75.43 | 92.57 | 81.26 | 94.87 | 62.74 | 84.31 | 9.16 | 25.27 | 18.41 | 30.80 | 84.18 | 95.66 |
| Factorized | $512^2$ | 74.97 | 92.22 | 80.65 | 94.47 | 62.29 | 83.56 | 8.05 | 22.81 | 17.78 | 29.80 | 84.10 | 95.76 |
| Fourier | $512^2$ | 75.68 | 92.56 | 81.33 | 94.78 | 63.14 | 83.90 | 9.96 | 26.20 | 18.11 | 30.06 | 84.76 | 95.94 |
| RPE-learn | $512^2$ | 72.59 | 90.74 | 78.81 | 93.52 | 59.35 | 81.83 | 6.45 | 21.00 | 12.17 | 21.45 | 81.64 | 94.36 |
| 2D-ALiBi | $512^2$ | 78.86 | 94.18 | 84.02 | 95.96 | 67.28 | 87.05 | 14.23 | 35.83 | 20.23 | 32.23 | 86.42 | 96.56 |
| 2D-RoPE | $512^2$ | 79.23 | 94.39 | 84.61 | 96.15 | 67.44 | 87.22 | 16.05 | 38.53 | 21.09 | 33.74 | 86.90 | 96.90 |
| LH-180 | $512^2$ | 81.11 | 95.26 | 86.46 | 96.87 | 69.96 | 88.97 | 19.49 | 41.81 | 22.64 | 33.98 | 88.08 | 97.48 |
| LH-90 | $512^2$ | 81.19 | 95.16 | 86.38 | 96.84 | 70.35 | 88.92 | 20.05 | 42.09 | 23.44 | 35.18 | 88.62 | 97.16 |
| LH-45 | $512^2$ | 81.62 | 95.44 | 86.57 | 96.91 | 71.09 | 88.80 | 20.55 | 43.21 | 23.87 | 35.76 | 89.04 | 97.40 |
| 1D-learn | $768^2$ | 65.95 | 87.11 | 71.75 | 90.33 | 51.11 | 75.27 | 3.79 | 13.16 | 12.13 | 22.47 | 75.76 | 92.64 |
| 2D-sincos | $768^2$ | 65.48 | 86.90 | 71.19 | 90.19 | 50.64 | 75.36 | 3.84 | 12.77 | 11.45 | 21.22 | 75.46 | 92.06 |
| Factorized | $768^2$ | 63.71 | 85.36 | 69.15 | 88.81 | 48.58 | 73.17 | 3.04 | 10.95 | 10.70 | 20.32 | 74.64 | 92.24 |
| Fourier | $768^2$ | 65.97 | 86.92 | 71.56 | 90.01 | 51.25 | 75.24 | 4.05 | 13.45 | 11.53 | 21.38 | 76.04 | 92.54 |
| RPE-learn | $768^2$ | 57.16 | 79.87 | 63.00 | 83.99 | 41.56 | 65.96 | 2.55 | 8.91 | 4.83 | 9.98 | 66.68 | 86.32 |
| 2D-ALiBi | $768^2$ | 72.97 | 90.64 | 78.13 | 93.26 | 59.19 | 81.53 | 8.48 | 24.00 | 12.83 | 22.54 | 79.96 | 93.76 |
| 2D-RoPE | $768^2$ | 71.28 | 89.93 | 77.03 | 92.54 | 56.70 | 79.93 | 7.53 | 22.20 | 12.00 | 21.23 | 79.14 | 93.68 |
| LH-180 | $768^2$ | 76.59 | 92.92 | 82.17 | 95.09 | 63.88 | 84.63 | 12.52 | 29.41 | 15.56 | 25.88 | 83.56 | 95.68 |
| LH-90 | $768^2$ | 77.12 | 93.38 | 82.68 | 95.46 | 64.49 | 85.23 | 12.89 | 30.44 | 17.52 | 28.68 | 84.90 | 96.08 |
| LH-45 | $768^2$ | 78.13 | 93.76 | 83.67 | 95.68 | 66.51 | 86.17 | 14.21 | 31.96 | 18.14 | 28.64 | 85.30 | 96.10 |
| 1D-learn | $1024^2$ | 55.67 | 80.00 | 61.00 | 83.85 | 40.97 | 65.86 | 1.95 | 7.77 | 8.46 | 17.28 | 65.14 | 86.02 |
| 2D-sincos | $1024^2$ | 53.71 | 78.36 | 58.91 | 82.32 | 39.57 | 64.82 | 1.48 | 6.04 | 7.08 | 14.93 | 64.62 | 86.16 |
| Factorized | $1024^2$ | 50.46 | 75.08 | 55.22 | 79.17 | 37.41 | 62.22 | 1.33 | 5.43 | 7.00 | 14.51 | 63.86 | 85.08 |
| Fourier | $1024^2$ | 54.58 | 78.17 | 59.80 | 82.14 | 39.22 | 64.61 | 1.87 | 7.01 | 7.65 | 15.49 | 64.66 | 86.46 |
| RPE-learn | $1024^2$ | 36.80 | 60.77 | 41.10 | 65.25 | 24.37 | 46.37 | 0.99 | 3.88 | 1.85 | 4.81 | 48.40 | 72.34 |
| 2D-ALiBi | $1024^2$ | 63.62 | 84.36 | 68.80 | 87.80 | 49.88 | 73.57 | 4.51 | 14.48 | 7.78 | 15.24 | 71.56 | 88.74 |
| 2D-RoPE | $1024^2$ | 51.41 | 75.05 | 56.71 | 79.27 | 37.18 | 60.56 | 2.12 | 8.04 | 4.00 | 9.06 | 60.30 | 82.64 |
| LH-180 | $1024^2$ | 69.24 | 88.81 | 75.22 | 91.84 | 55.84 | 78.89 | 6.63 | 18.92 | 11.23 | 20.18 | 76.14 | 92.44 |
| LH-90 | $1024^2$ | 71.58 | 90.19 | 77.48 | 92.99 | 58.30 | 80.83 | 8.13 | 21.29 | 12.61 | 22.50 | 79.68 | 93.34 |
| LH-45 | $1024^2$ | 73.15 | 91.09 | 79.01 | 93.63 | 60.26 | 82.38 | 8.35 | 22.15 | 14.11 | 23.78 | 80.48 | 93.78 |

### A.2.1 Other Models

Table 7: Top-1 acc. (%) on Val [1] for models outside our controlled experiment, using the timm library [104].

| Name | $224^2$ | $320^2$ | $384^2$ | $448^2$ | $512^2$ | $768^2$ | $1024^2$ |
|---|---|---|---|---|---|---|---|
| beitv2_large_patch16_224.in1k_ft_in22k_in1k[106] | 87.97 | 87.73 | 80.76 | 60.80 | 40.48 | 10.63 | 5.19 |
| caformer_b36.sail_in1k[52] | 85.28 | 85.61 | 84.93 | 84.08 | 83.08 | 77.90 | 70.35 |
| caformer_b36.sail_in22k_ft_in1k[52] | 87.24 | 87.29 | 86.34 | 84.78 | 82.96 | 73.55 | 62.60 |
| convformer_b36.sail_in1k[52] | 84.59 | 85.06 | 83.98 | 82.34 | 79.48 | 58.24 | 37.54 |
| convformer_s18.sail_in1k[52] | 82.89 | 83.34 | 81.72 | 78.86 | 73.36 | 39.49 | 17.57 |
| eva_giant_patch14_224.clip_ft_in1k[107] | 88.75 | 88.86 | 88.50 | 87.83 | 87.22 | 83.37 | 78.31 |
| iRPE (our implementation)[39] | 80.53 | 81.59 | 81.47 | 80.77 | 79.94 | 72.86 | 60.30 |
| swin_base_patch4_window7_224.ms_in22k_ft_in1k[44] | 84.40 | 84.80 | 84.31 | 83.77 | 82.90 | 78.56 | 70.58 |
| swin_s3_base_224.ms_in1k[49] | 83.86 | 82.61 | 81.34 | 80.39 | 79.30 | 73.54 | 63.78 |
| swin_tiny_patch4_window7_224[44] | 80.85 | 80.92 | 79.96 | 79.09 | 78.33 | 72.24 | 61.06 |
| twins_pcpvt_base.in1k[51] | 82.54 | 83.20 | 82.27 | 81.06 | 79.68 | 72.22 | 61.24 |
| twins_pcpvt_small.in1k[51] | 80.94 | 81.67 | 80.92 | 79.76 | 78.46 | 70.56 | 58.74 |
| twins_svt_large.in1k[51] | 83.38 | 83.44 | 82.64 | 82.03 | 80.99 | 76.49 | 69.01 |
| vit_base_patch16_clip_224.laion2b_ft_in12k_in1k[108] | 85.79 | 85.84 | 85.03 | 84.24 | 83.25 | 76.35 | 66.60 |
| vit_base_patch16_rope_reg1_gap_256.sbb_in1k[104] | 81.26 | 82.33 | 81.88 | 80.89 | 79.66 | 72.66 | 63.25 |
| vit_large_patch14_clip_224.openai_ft_in12k_in1k[108] | 87.93 | 88.08 | 87.57 | 87.02 | 86.18 | 81.51 | 75.51 |
| vit_mediumd_patch16_rope_reg1_gap_256.sbb_in1k[104] | 81.55 | 82.67 | 82.23 | 81.43 | 80.08 | 73.24 | 64.69 |
| vit_small_r26_s32_224[9] | 81.38 | 82.89 | 82.46 | 81.75 | 80.32 | 72.30 | 61.38 |
| xcit_medium_24_p8_224.fb_dist_in1k[53] | 84.86 | 85.36 | 84.97 | 84.41 | 83.75 | 79.58 | 72.94 |
| xcit_small_12_p16_224.fb_in1k[53] | 81.68 | 82.65 | 82.10 | 81.51 | 80.43 | 74.48 | 65.46 |

Table 8: Top-1 acc. (%) on -HR (ours) for models outside our controlled experiment, using the timm library [104].

| Name | $224^2$ | $320^2$ | $384^2$ | $448^2$ | $512^2$ | $768^2$ | $1024^2$ |
|---|---|---|---|---|---|---|---|
| beitv2_large_patch16_224.in1k_ft_in22k_in1k[106] | 95.16 | 95.24 | 90.36 | 73.72 | 52.60 | 14.32 | 7.40 |
| caformer_b36.sail_in1k[52] | 93.06 | 93.08 | 92.84 | 92.10 | 90.88 | 85.68 | 79.30 |
| caformer_b36.sail_in22k_ft_in1k[52] | 94.40 | 94.56 | 94.02 | 93.00 | 91.34 | 82.16 | 70.94 |
| convformer_b36.sail_in1k[52] | 92.44 | 92.26 | 90.94 | 88.94 | 85.40 | 70.04 | 59.80 |
| convformer_s18.sail_in1k[52] | 90.98 | 90.84 | 88.40 | 83.30 | 77.24 | 53.18 | 41.02 |
| eva_giant_patch14_224.clip_ft_in1k[107] | 95.96 | 95.86 | 95.70 | 95.58 | 95.36 | 92.94 | 89.24 |
| iRPE (our implementation)[39] | 89.10 | 89.76 | 89.56 | 88.88 | 88.12 | 82.40 | 72.60 |
| swin_base_patch4_window7_224.ms_in22k_ft_in1k[44] | 91.82 | 92.20 | 91.18 | 90.24 | 90.06 | 85.22 | 79.26 |
| swin_s3_base_224.ms_in1k[49] | 91.78 | 90.22 | 88.52 | 86.82 | 85.72 | 78.00 | 68.46 |
| swin_tiny_patch4_window7_224[44] | 89.06 | 89.10 | 87.56 | 86.76 | 85.46 | 77.58 | 68.54 |
| twins_pcpvt_base.in1k[51] | 90.54 | 90.92 | 90.30 | 89.24 | 87.60 | 79.50 | 69.74 |
| twins_pcpvt_small.in1k[51] | 89.62 | 89.40 | 88.76 | 87.14 | 85.62 | 77.40 | 67.30 |
| twins_svt_large.in1k[51] | 91.46 | 91.24 | 90.34 | 89.36 | 87.82 | 82.42 | 76.22 |
| vit_base_patch16_clip_224.laion2b_ft_in12k_in1k[108] | 93.22 | 93.38 | 93.24 | 92.68 | 92.02 | 88.14 | 81.92 |
| vit_base_patch16_rope_reg1_gap_256.sbb_in1k[104] | 89.36 | 90.84 | 90.14 | 89.22 | 88.22 | 83.36 | 76.54 |
| vit_large_patch14_clip_224.openai_ft_in12k_in1k[108] | 94.62 | 94.78 | 94.68 | 94.62 | 94.10 | 91.10 | 86.64 |
| vit_mediumd_patch16_rope_reg1_gap_256.sbb_in1k[104] | 89.42 | 90.56 | 90.38 | 89.74 | 89.26 | 83.90 | 77.48 |
| vit_small_r26_s32_224[9] | 89.76 | 90.52 | 90.30 | 89.56 | 88.80 | 82.06 | 72.12 |
| xcit_medium_24_p8_224.fb_dist_in1k[53] | 92.54 | 92.94 | 92.34 | 91.94 | 91.20 | 87.70 | 81.80 |
| xcit_small_12_p16_224.fb_in1k[53] | 89.54 | 90.14 | 89.88 | 89.02 | 88.22 | 82.82 | 74.86 |

## A.3 LookHere Bias Matrices

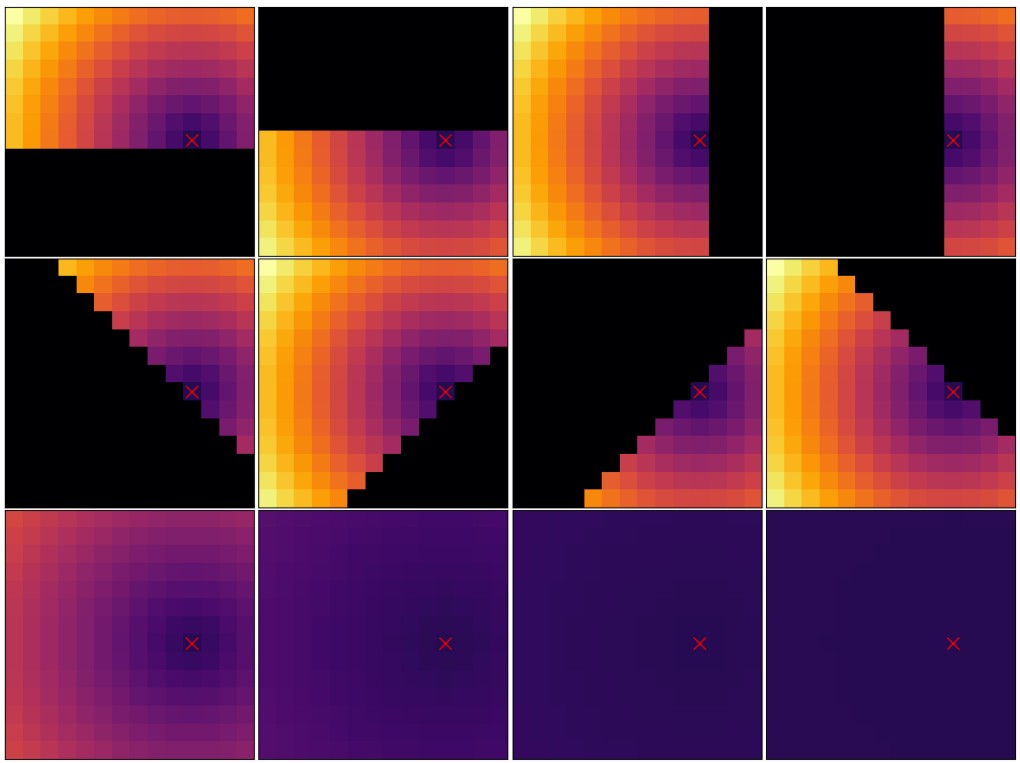

Figure 9: LH-180 bias matrices for query patch (11,8), grid size of 14x14.

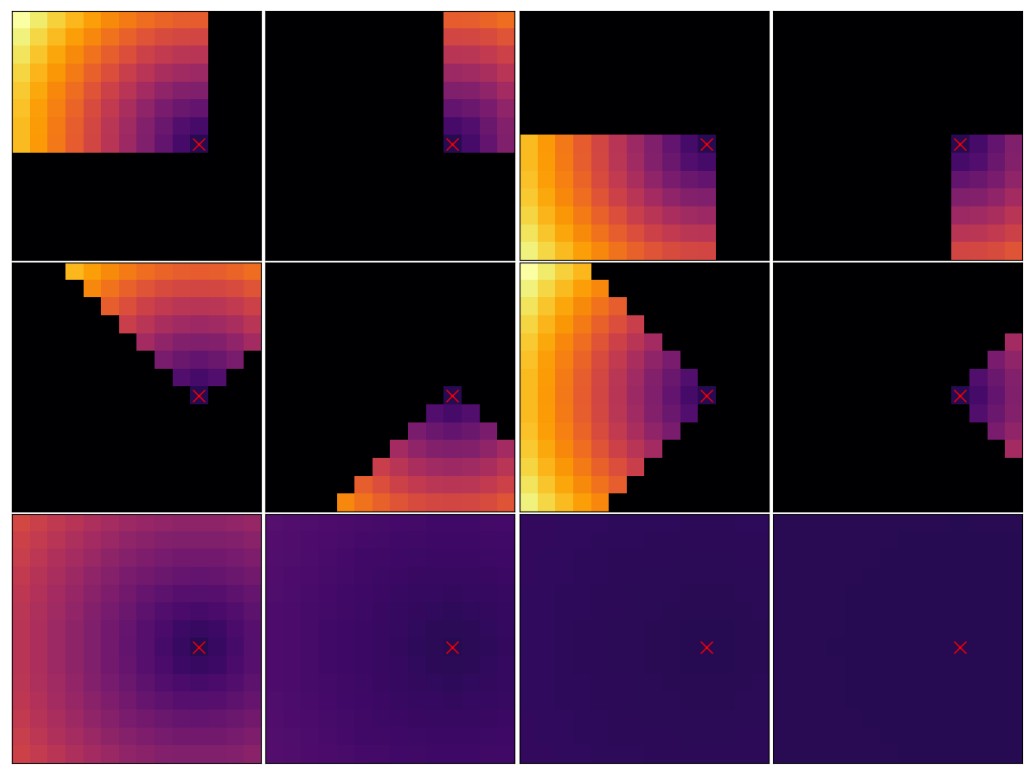

Figure 10: LH-90 bias matrices for query patch (11,8), grid size of 14x14.

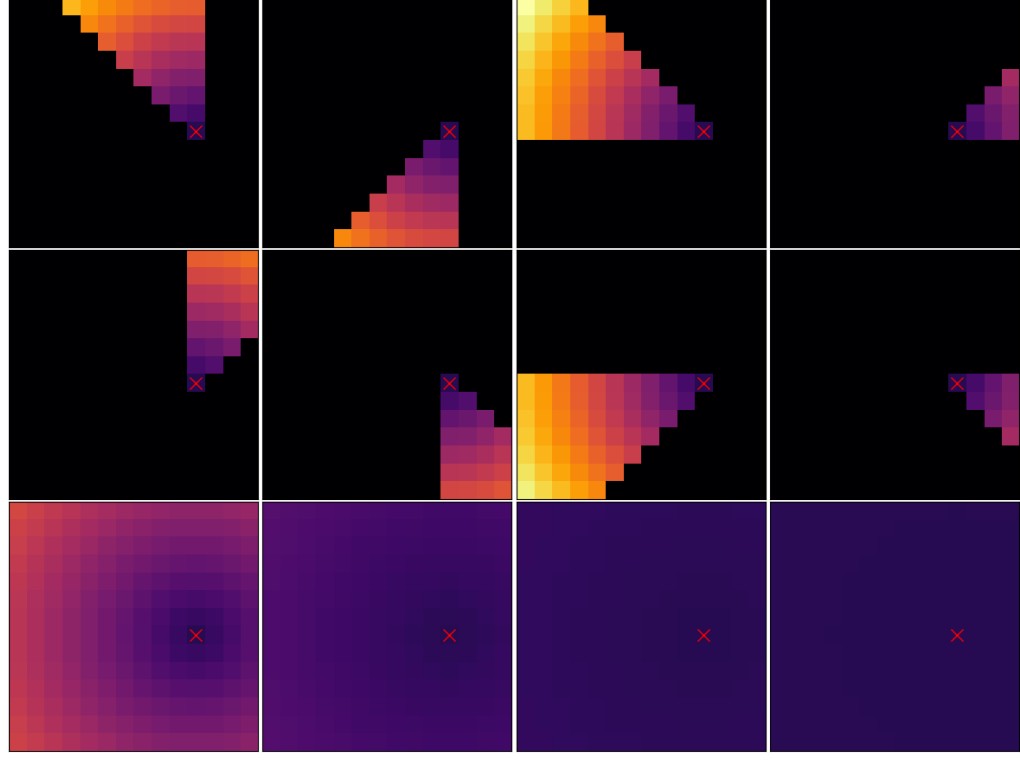

Figure 11: LH-45 bias matrices for query patch (11,8), grid size of 14x14.

### A.4 Experimental Details

#### A.4.1 Training ViTs

**Recipe.** Our training recipe that is consistent across configurations:

- AdamW [109] — using the default PyTorch implementation that does not fully decouple learning rate and weight decay
- Binary cross-entropy loss — summing along the class dimension, averaging along the batch dimension
- Linear warm-up for $10\%$ of steps and cool-down using a cosine decay schedule to a zero learning rate
- Batch size of $2048$
- Mixup [110] $\alpha = 0.8$, cutmix [111] $\alpha = 1$
- CLS token with an MLP classifying head — final linear layer weights are initialized to $0$ and biases to $-6.9$ (so all class probabilities start at $\frac{1}{1000}$)
- layer drop rate of $0.1$ and MLP dropout of $0$
- Train for $150$ epochs on the first $99\%$ of ImageNet-1k — using Huggingface's datasets library, i.e., `load_dataset("imagenet-1k", split="train[:99%]")`
- Choose checkpoint according to the best minival top-1 accuracy (run after each epoch), where minival is the last $1\%$ of the ImageNet-1k training set, i.e., `load_dataset("imagenet-1k", split="train[99%:]")`

#### A.4.2 Compute

Training takes around 3 days on an RTX $4090$ GPU. Thus, all $80$ training runs take around $240$ GPU-days. We spend another $54$ GPU-days on $18$ ablations. Ablations and our iRPE run always use our best training recipe, which is 3-Augment [17] data augmentation, $3 \cdot 10^{-3}$ learning rate, and $0.05$ weight decay. iRPE [39] takes around 7 days on an RTX $4090$ GPU, even with the official custom CUDA kernel. As a result, we exclude it from our apples-to-apples comparisons.

#### A.4.3 High-resolution finetuning

Following DEiT III's finetuning recipe [17], we increase the drop rate to $0.2$ and the weight decay to $0.1$, and fix the learning rate to $10^{-5}$ with a $512$ batch size.

#### A.4.4 Extrapolation Tuning

For 2D-ALiBi, 2D-RoPE, and LookHere models, we tune a single parameter at the target resolution on minival (Table 9). LookHere models benefit less from tuning than 2D-ALiBi and 2D-RoPE models. For example at a $512^2$ resolution, the difference in top-1 accuracy on minival when using the tuned parameter versus the default value is $2.1\%$ for 2D-ALiBi, $1.3\%$ for 2D-RoPE, and $0.15\%$ for LH-45. Thus, LookHere does not require tuning its global slope value to effectively extrapolate.

Table 9: Tuned Parameter Values

| Name | Tuning Parameter | $224^2$ | $320^2$ | $384^2$ | $448^2$ | $512^2$ | $768^2$ | $1024^2$ |
|------|------------------|---------|---------|---------|---------|---------|---------|----------|
| 2D-ALiBi | $s_g$ | 1.0 | 1.4 | 1.4 | 1.4 | 1.4 | 1.5 | 1.6 |
| 2D-RoPE | base frequency | 100 | 160 | 190 | 250 | 700 | 1250 | 1250 |
| LookHere | $s_g$ | 1.0 | 1.00 | 0.95 | 0.95 | 0.95 | 0.75 | 0.6 |

#### A.4.5 Segmentation

For both linear probing and full finetuning we use a linear decoder. The linear decoder consists of a linear layer applied to the frozen patch representations which is then upsampled to the original image size. Similar to [85] we add a BatchNorm layer before the linear layer.

For full finetuning, we followed the Segmenter training recipe [84] exactly. For ADE20k, the base learning rate is $10^{-3}$ for 160k iterations with a batch size of 8, at $512^2$ px. For Cityscapes, the base learning rate is $10^{-2}$ for 80k iterations with a batch size of 8, at $384^2$ px. We train with SGD.

For linear probing, we freeze the backbone and pre-compute the patch representations. We use the AdamW optimizer [109] and sweep the following learning rates: $\{0.0001, 0.0002, 0.0005, 0.001, 0.002, 0.005, 0.01, 0.02, 0.05, 0.1, 0.2, 0.3, 0.5\}$. For both ADE20k and Cityscapes we set the batch size to 16 and train the linear decoder for 50 epochs.

## A.5 Full Experimental Results

Table 10: First half of our hyper-parameter sweep. ViT-B models trained on ImageNet for 150 epochs; trained and tested at $224^2$. RA is for RandAugment and 3A for 3-Augment.

| Method | WD $10^{-2}$ | LR $10^{-3}$ | Data Aug | Val [1] top-1 | top-5 | ReaL [4] top-1 | top-5 | v2 [2] top-1 | top-5 | -A [3] top-1 | top-5 | -R [5] top-1 | top-5 | -HR (ours) top-1 | top-5 |
|---|---|---|---|---|---|---|---|---|---|---|---|---|---|---|---|
| 1D-learn | 2 | 3.0 | 3A | 78.51 | 93.42 | 83.91 | 95.45 | 66.86 | 86.11 | 9.64 | 27.71 | 28.21 | 41.85 | 86.54 | 95.88 |
| 2D-sincos | 2 | 3.0 | 3A | 77.96 | 93.29 | 83.63 | 95.34 | 66.35 | 85.84 | 9.55 | 27.57 | 28.29 | 42.01 | 86.24 | 96.16 |
| Factorized | 2 | 3.0 | 3A | 78.42 | 93.47 | 84.09 | 95.64 | 66.89 | 85.84 | 8.36 | 25.65 | 28.79 | 42.68 | 86.70 | 96.48 |
| Fourier | 2 | 3.0 | 3A | 78.78 | 93.37 | 84.21 | 95.50 | 67.32 | 85.89 | 9.56 | 27.08 | 28.90 | 42.29 | 87.24 | 96.28 |
| RPE-learn | 2 | 3.0 | 3A | 78.92 | 93.76 | 84.46 | 95.74 | 67.75 | 86.28 | 10.00 | 28.01 | 28.70 | 42.45 | 87.06 | 96.24 |
| 2D-ALiBi | 2 | 3.0 | 3A | 78.47 | 93.68 | 84.19 | 95.80 | 66.66 | 86.18 | 9.01 | 26.44 | 26.51 | 39.74 | 86.46 | 96.50 |
| 2D-RoPE | 2 | 3.0 | 3A | 79.36 | 93.96 | 84.70 | 95.96 | 67.98 | 86.93 | 10.37 | 28.36 | 30.58 | 44.46 | 87.42 | 96.64 |
| **LH-180** | 2 | 3.0 | 3A | 80.76 | 94.78 | 86.23 | 96.56 | 69.43 | 88.02 | 11.47 | 29.87 | 31.09 | 44.20 | 88.86 | 97.08 |
| **LH-90** | 2 | 3.0 | 3A | 80.75 | 94.71 | 86.17 | 96.45 | 69.85 | 87.97 | 12.27 | 30.24 | 31.19 | 44.20 | 88.90 | 97.06 |
| **LH-45** | 2 | 3.0 | 3A | 80.49 | 94.55 | 86.06 | 96.42 | 69.27 | 87.55 | 11.44 | 30.56 | 31.70 | 45.38 | 88.90 | 97.02 |
| 1D-learn | 5 | 3.0 | 3A | 79.45 | 94.30 | 84.97 | 96.10 | 68.49 | 87.59 | 10.97 | 30.59 | 29.64 | 43.48 | 88.28 | 96.76 |
| 2D-sincos | 5 | 3.0 | 3A | 79.05 | 94.25 | 84.62 | 96.14 | 67.86 | 87.01 | 10.45 | 29.41 | 29.11 | 43.24 | 87.58 | 96.48 |
| Factorized | 5 | 3.0 | 3A | 79.86 | 94.73 | 85.30 | 96.41 | 69.11 | 87.87 | 11.00 | 31.32 | 29.99 | 44.22 | 87.86 | 97.02 |
| Fourier | 5 | 3.0 | 3A | 79.69 | 94.41 | 85.13 | 96.36 | 68.30 | 87.66 | 11.36 | 30.93 | 29.73 | 43.90 | 88.14 | 96.96 |
| RPE-learn | 5 | 3.0 | 3A | 79.86 | 94.64 | 85.46 | 96.64 | 68.57 | 87.72 | 9.85 | 29.27 | 29.10 | 43.28 | 88.22 | 97.32 |
| 2D-ALiBi | 5 | 3.0 | 3A | 79.54 | 94.57 | 85.15 | 96.38 | 68.47 | 87.58 | 10.45 | 29.33 | 28.26 | 41.91 | 87.70 | 96.74 |
| 2D-RoPE | 5 | 3.0 | 3A | 80.38 | 94.86 | 85.64 | 96.49 | 69.34 | 87.89 | 13.03 | 33.95 | 32.45 | 46.96 | 88.78 | 96.92 |
| **LH-180** | 5 | 3.0 | 3A | 81.31 | 95.11 | 86.53 | 96.71 | 70.70 | 88.38 | 13.53 | 32.72 | 32.10 | 45.07 | 89.86 | 97.54 |
| **LH-90** | 5 | 3.0 | 3A | 81.02 | 94.92 | 86.44 | 96.68 | 70.28 | 88.34 | 13.15 | 32.89 | 31.77 | 44.74 | 89.90 | 97.20 |
| **LH-45** | 5 | 3.0 | 3A | 81.06 | 94.87 | 86.23 | 96.46 | 69.65 | 88.60 | 13.41 | 32.96 | 32.12 | 45.25 | 89.46 | 97.06 |
| 1D-learn | 2 | 3.0 | RA | 76.51 | 92.08 | 81.84 | 94.32 | 63.89 | 83.41 | 6.12 | 19.93 | 23.56 | 36.15 | 84.26 | 94.96 |
| 2D-sincos | 2 | 3.0 | RA | 76.38 | 92.22 | 81.77 | 94.53 | 63.87 | 84.05 | 6.57 | 20.23 | 23.62 | 36.95 | 84.40 | 95.28 |
| Factorized | 2 | 3.0 | RA | 76.45 | 92.18 | 82.16 | 94.53 | 64.31 | 84.10 | 6.57 | 20.97 | 24.30 | 37.35 | 84.34 | 94.90 |
| Fourier | 2 | 3.0 | RA | 76.59 | 92.08 | 82.07 | 94.49 | 64.51 | 84.13 | 7.28 | 21.72 | 24.20 | 37.39 | 83.76 | 94.68 |
| RPE-learn | 2 | 3.0 | RA | 76.37 | 92.28 | 81.90 | 94.54 | 63.99 | 83.41 | 6.12 | 18.76 | 23.05 | 36.01 | 83.58 | 94.96 |
| 2D-ALiBi | 2 | 3.0 | RA | 76.08 | 92.16 | 81.52 | 94.45 | 63.67 | 83.22 | 5.61 | 19.08 | 22.17 | 34.74 | 83.20 | 94.78 |
| 2D-RoPE | 2 | 3.0 | RA | 77.31 | 93.10 | 82.84 | 95.22 | 65.06 | 84.75 | 6.07 | 20.63 | 27.05 | 41.09 | 85.24 | 95.76 |
| **LH-180** | 2 | 3.0 | RA | 80.02 | 94.07 | 85.15 | 95.79 | 68.32 | 86.73 | 9.21 | 25.53 | 27.69 | 40.24 | 87.18 | 96.70 |
| **LH-90** | 2 | 3.0 | RA | 79.36 | 93.83 | 84.67 | 95.64 | 67.64 | 86.43 | 10.00 | 24.99 | 27.86 | 41.01 | 87.20 | 96.16 |
| **LH-45** | 2 | 3.0 | RA | 79.77 | 93.99 | 84.93 | 95.68 | 68.30 | 86.41 | 9.36 | 25.91 | 28.35 | 41.63 | 86.40 | 96.40 |
| 1D-learn | 5 | 3.0 | RA | 78.06 | 93.38 | 83.35 | 95.44 | 65.33 | 85.67 | 8.07 | 25.07 | 25.98 | 39.42 | 84.94 | 95.64 |
| 2D-sincos | 5 | 3.0 | RA | 77.95 | 93.27 | 83.26 | 95.37 | 65.51 | 85.64 | 7.57 | 25.37 | 26.11 | 39.63 | 85.16 | 95.88 |
| Factorized | 5 | 3.0 | RA | 78.55 | 93.96 | 84.00 | 95.91 | 66.88 | 86.19 | 8.05 | 24.05 | 27.08 | 40.76 | 86.36 | 96.28 |
| Fourier | 5 | 3.0 | RA | 78.16 | 93.47 | 83.43 | 95.41 | 66.28 | 85.90 | 8.28 | 25.16 | 26.25 | 39.94 | 85.98 | 95.98 |
| RPE-learn | 5 | 3.0 | RA | 78.15 | 93.57 | 83.50 | 95.51 | 66.50 | 85.91 | 7.56 | 23.64 | 25.10 | 38.51 | 85.62 | 95.64 |
| 2D-ALiBi | 5 | 3.0 | RA | 77.00 | 92.89 | 82.49 | 94.88 | 64.75 | 85.05 | 6.88 | 21.55 | 23.65 | 36.51 | 84.22 | 95.46 |
| 2D-RoPE | 5 | 3.0 | RA | 79.29 | 94.05 | 84.46 | 95.85 | 67.73 | 86.67 | 10.77 | 29.89 | 28.99 | 43.18 | 86.50 | 96.30 |
| **LH-180** | 5 | 3.0 | RA | 80.20 | 94.32 | 85.22 | 96.02 | 68.27 | 86.70 | 10.60 | 27.40 | 27.80 | 40.05 | 87.74 | 96.60 |
| **LH-90** | 5 | 3.0 | RA | 80.35 | 94.33 | 85.47 | 95.96 | 68.98 | 87.29 | 10.67 | 27.89 | 28.54 | 41.64 | 87.76 | 96.56 |
| **LH-45** | 5 | 3.0 | RA | 80.13 | 94.31 | 85.10 | 95.99 | 68.19 | 86.65 | 10.89 | 27.57 | 28.35 | 41.35 | 87.08 | 96.20 |

Table 11: Second half of our hyper-parameter sweep. ViT-B models trained on ImageNet for 150 epochs; trained and tested at $224^2$. RA is for RandAugment and 3A for 3-Augment.

| Method | WD $10^{-2}$ | LR $10^{-3}$ | Data Aug | Val [1] top-1 | top-5 | ReaL [4] top-1 | top-5 | v2 [2] top-1 | top-5 | -A [3] top-1 | top-5 | -R [5] top-1 | top-5 | -HR (ours) top-1 | top-5 |
|---|---|---|---|---|---|---|---|---|---|---|---|---|---|---|---|
| 1D-learn | 2 | 1.5 | 3A | 77.31 | 92.41 | 82.95 | 94.79 | 64.85 | 84.22 | 6.95 | 22.00 | 26.14 | 38.81 | 85.68 | 95.84 |
| 2D-sincos | 2 | 1.5 | 3A | 77.50 | 92.64 | 83.14 | 94.93 | 65.57 | 84.70 | 7.23 | 22.91 | 26.66 | 39.92 | 85.78 | 95.60 |
| Factorized | 2 | 1.5 | 3A | 76.88 | 92.43 | 82.82 | 94.88 | 64.96 | 84.43 | 5.99 | 19.96 | 26.47 | 39.91 | 85.36 | 95.38 |
| Fourier | 2 | 1.5 | 3A | 77.15 | 92.31 | 82.82 | 94.63 | 64.92 | 84.24 | 7.09 | 22.67 | 26.28 | 39.49 | 85.38 | 95.48 |
| RPE-learn | 2 | 1.5 | 3A | 77.07 | 92.61 | 82.97 | 95.00 | 65.13 | 84.52 | 6.52 | 21.40 | 24.75 | 37.91 | 85.16 | 95.62 |
| 2D-ALiBi | 2 | 1.5 | 3A | 77.72 | 93.05 | 83.40 | 95.38 | 66.23 | 85.56 | 7.59 | 22.76 | 25.78 | 38.89 | 85.88 | 96.14 |
| 2D-RoPE | 2 | 1.5 | 3A | 78.14 | 93.19 | 83.74 | 95.40 | 66.67 | 85.57 | 8.20 | 25.76 | 28.78 | 42.64 | 86.26 | 96.14 |
| LH-180 | 2 | 1.5 | 3A | 80.14 | 94.19 | 85.51 | 96.12 | 68.87 | 87.25 | 11.03 | 27.84 | 29.73 | 42.81 | 88.14 | 96.82 |
| LH-90 | 2 | 1.5 | 3A | 79.88 | 94.18 | 85.51 | 96.12 | 69.34 | 87.07 | 10.83 | 28.32 | 30.88 | 44.23 | 88.32 | 96.92 |
| LH-45 | 2 | 1.5 | 3A | 79.57 | 94.06 | 85.22 | 96.01 | 68.40 | 87.02 | 9.43 | 27.60 | 30.69 | 44.85 | 87.86 | 96.88 |
| 1D-learn | 5 | 1.5 | 3A | 77.87 | 93.31 | 83.56 | 95.47 | 66.56 | 85.69 | 8.64 | 25.03 | 27.16 | 40.67 | 86.40 | 95.86 |
| 2D-sincos | 5 | 1.5 | 3A | 78.48 | 93.50 | 83.99 | 95.65 | 66.65 | 86.19 | 8.85 | 25.75 | 27.72 | 41.90 | 86.88 | 96.46 |
| Factorized | 5 | 1.5 | 3A | 77.34 | 92.98 | 83.12 | 95.32 | 65.62 | 85.33 | 7.24 | 23.51 | 26.86 | 40.51 | 86.42 | 96.02 |
| Fourier | 5 | 1.5 | 3A | 77.89 | 93.18 | 83.59 | 95.30 | 66.02 | 84.99 | 8.49 | 25.13 | 26.44 | 39.74 | 86.04 | 96.04 |
| RPE-learn | 5 | 1.5 | 3A | 77.71 | 93.31 | 83.50 | 95.44 | 66.12 | 85.43 | 7.91 | 24.67 | 25.23 | 38.30 | 86.60 | 96.02 |
| 2D-ALiBi | 5 | 1.5 | 3A | 78.56 | 93.77 | 84.32 | 95.88 | 66.34 | 86.57 | 8.60 | 27.16 | 26.97 | 40.76 | 86.62 | 96.60 |
| 2D-RoPE | 5 | 1.5 | 3A | 78.74 | 93.79 | 84.45 | 95.86 | 67.12 | 86.61 | 9.69 | 27.61 | 29.82 | 43.58 | 87.62 | 96.60 |
| LH-180 | 5 | 1.5 | 3A | 80.53 | 94.65 | 85.82 | 96.47 | 69.38 | 87.70 | 11.63 | 29.75 | 30.07 | 43.02 | 88.66 | 97.06 |
| LH-90 | 5 | 1.5 | 3A | 80.34 | 94.68 | 85.81 | 96.48 | 68.92 | 87.89 | 11.55 | 30.31 | 30.85 | 44.73 | 88.56 | 97.00 |
| LH-45 | 5 | 1.5 | 3A | 80.32 | 94.60 | 85.59 | 96.34 | 68.78 | 87.33 | 10.71 | 29.65 | 31.25 | 45.14 | 88.82 | 97.06 |
| 1D-learn | 2 | 1.5 | RA | 75.02 | 90.83 | 80.68 | 93.26 | 62.38 | 81.45 | 4.55 | 16.04 | 21.92 | 33.64 | 82.50 | 94.44 |
| 2D-sincos | 2 | 1.5 | RA | 75.72 | 91.12 | 81.32 | 93.59 | 62.60 | 82.06 | 5.61 | 18.31 | 22.82 | 35.10 | 82.82 | 94.00 |
| Factorized | 2 | 1.5 | RA | 74.47 | 90.62 | 80.40 | 93.21 | 61.73 | 81.29 | 4.63 | 15.31 | 22.10 | 33.95 | 82.16 | 93.40 |
| Fourier | 2 | 1.5 | RA | 74.95 | 90.58 | 80.59 | 93.19 | 61.66 | 81.23 | 4.97 | 17.11 | 22.08 | 34.01 | 82.84 | 94.14 |
| RPE-learn | 2 | 1.5 | RA | 74.65 | 90.50 | 80.28 | 93.11 | 61.42 | 81.04 | 4.44 | 15.64 | 20.22 | 31.96 | 82.14 | 94.12 |
| 2D-ALiBi | 2 | 1.5 | RA | 74.95 | 90.75 | 80.69 | 93.40 | 61.92 | 81.07 | 5.01 | 16.52 | 20.45 | 32.03 | 83.18 | 93.82 |
| 2D-RoPE | 2 | 1.5 | RA | 76.59 | 91.56 | 81.98 | 93.90 | 63.96 | 83.06 | 6.13 | 19.84 | 24.69 | 37.57 | 83.94 | 95.04 |
| LH-180 | 2 | 1.5 | RA | 78.42 | 93.04 | 83.68 | 95.03 | 66.46 | 85.12 | 8.20 | 22.65 | 26.07 | 38.42 | 86.18 | 95.58 |
| LH-90 | 2 | 1.5 | RA | 78.62 | 93.21 | 84.09 | 95.16 | 66.64 | 85.15 | 8.77 | 24.07 | 27.22 | 39.94 | 86.50 | 95.82 |
| LH-45 | 2 | 1.5 | RA | 78.17 | 93.12 | 83.61 | 95.22 | 66.54 | 85.01 | 7.49 | 23.24 | 26.56 | 39.61 | 85.98 | 95.78 |
| 1D-learn | 5 | 1.5 | RA | 76.07 | 91.79 | 81.66 | 94.11 | 63.03 | 83.12 | 5.73 | 19.24 | 23.20 | 35.88 | 83.12 | 94.62 |
| 2D-sincos | 5 | 1.5 | RA | 76.50 | 92.22 | 81.96 | 94.49 | 64.10 | 83.72 | 6.21 | 19.16 | 24.23 | 37.13 | 84.02 | 94.68 |
| Factorized | 5 | 1.5 | RA | 76.38 | 92.25 | 82.05 | 94.69 | 63.25 | 83.35 | 5.41 | 17.81 | 23.88 | 36.33 | 83.72 | 95.16 |
| Fourier | 5 | 1.5 | RA | 75.78 | 91.66 | 81.31 | 94.04 | 63.62 | 83.10 | 5.29 | 18.76 | 22.95 | 35.26 | 83.70 | 94.80 |
| RPE-learn | 5 | 1.5 | RA | 75.33 | 91.29 | 80.98 | 93.77 | 62.04 | 82.19 | 5.07 | 18.47 | 20.80 | 32.74 | 83.00 | 94.32 |
| 2D-ALiBi | 5 | 1.5 | RA | 76.02 | 91.94 | 81.61 | 94.18 | 63.16 | 82.93 | 5.01 | 17.97 | 21.53 | 33.85 | 83.80 | 94.78 |
| 2D-RoPE | 5 | 1.5 | RA | 77.13 | 92.53 | 82.47 | 94.80 | 64.63 | 84.68 | 6.47 | 20.95 | 26.08 | 39.35 | 85.00 | 95.14 |
| LH-180 | 5 | 1.5 | RA | 78.68 | 93.68 | 84.26 | 95.69 | 66.76 | 85.67 | 7.93 | 23.17 | 26.98 | 40.13 | 85.74 | 96.12 |
| LH-90 | 5 | 1.5 | RA | 78.77 | 93.63 | 84.09 | 95.37 | 66.69 | 85.83 | 9.15 | 25.33 | 27.45 | 40.08 | 85.70 | 95.98 |
| LH-45 | 5 | 1.5 | RA | 78.39 | 93.45 | 83.85 | 95.37 | 66.30 | 85.61 | 8.97 | 24.84 | 27.11 | 40.44 | 84.90 | 95.66 |

### A.6 Ablations

We train 18 models to ablate the LookHere design. Each run uses our best 150 epoch training recipe. We test models without extrapolation at $224^2$ px (Table 12) and with extrapolation at $1024^2$ px (Table 13). Before running extrapolation tests, we tune the global slope of each model at $1024^2$ px to fairly compare with our three default variants. To fit in the tables, we use short forms explained here: "undir$\rightarrow$ 90" means replacing the four undirected heads with four 90° FOV heads, "undir$\rightarrow$ no dist" means removing the distance penalties on the four undirected heads, "invert" means inverting the layer-wise slope pattern such that $s_l$ linearly increases from 0.5 to 1.5 with depth, "mask:$\infty \rightarrow 0$" means replacing $\infty$ with 0 in equation 1, and "dist$\rightarrow$no dist" means removing the distance penalties on all heads.

Table 12: LookHere design ablations *without* extrapolation. ViT-B models trained on ImageNet for 150 epochs; trained and tested at $224^2$.

| Variant | Change | Val [1] | | ReaL [4] | | v2 [2] | | -A [3] | | -R [5] | | -HR (ours) | |
|---|---|---|---|---|---|---|---|---|---|---|---|---|---|
| | | top-1 | top-5 | top-1 | top-5 | top-1 | top-5 | top-1 | top-5 | top-1 | top-5 | top-1 | top-5 |
| LH-45 | undir$\rightarrow$ 90° | 80.53 | 94.92 | 86.27 | 96.81 | 69.42 | 88.29 | 10.33 | 29.60 | 32.49 | 46.78 | 89.42 | 97.38 |
| LH-45 | undir$\rightarrow$ 180° | 80.72 | 94.90 | 86.19 | 96.74 | 69.66 | 88.51 | 10.81 | 29.59 | 32.14 | 46.13 | 89.44 | 97.26 |
| LH-45 | undir$\rightarrow$no dist | 81.14 | 95.08 | 86.44 | 96.74 | 70.53 | 88.48 | 14.17 | 34.07 | 32.61 | 46.14 | 89.84 | 97.54 |
| LH-90 | undir$\rightarrow$ 90° | 81.00 | 94.98 | 86.59 | 96.78 | 70.15 | 88.46 | 10.84 | 29.40 | 32.44 | 46.46 | 89.10 | 97.62 |
| LH-90 | undir$\rightarrow$ 180° | 80.94 | 95.06 | 86.54 | 96.78 | 69.99 | 88.55 | 12.29 | 30.83 | 31.73 | 45.64 | 89.46 | 97.06 |
| LH-90 | undir$\rightarrow$no dist | 81.01 | 95.13 | 86.38 | 96.79 | 70.37 | 88.68 | 12.41 | 32.39 | 32.27 | 46.51 | 89.34 | 97.52 |
| LH-180 | undir$\rightarrow$ 90° | 80.82 | 95.02 | 86.56 | 96.78 | 69.50 | 88.57 | 11.85 | 29.99 | 31.85 | 45.77 | 88.98 | 97.18 |
| LH-180 | undir$\rightarrow$ 180° | 80.88 | 95.11 | 86.56 | 96.87 | 70.36 | 88.33 | 11.96 | 30.55 | 31.63 | 45.52 | 89.28 | 97.32 |
| LH-180 | undir$\rightarrow$no dist | 81.39 | 95.11 | 86.78 | 96.77 | 70.66 | 88.43 | 12.49 | 32.00 | 31.79 | 44.93 | 89.84 | 97.50 |
| LH-90 | $s_g : 1 \rightarrow 0.125$ | 81.20 | 95.03 | 86.48 | 96.69 | 70.14 | 88.33 | 13.63 | 33.27 | 32.14 | 45.44 | 89.22 | 97.08 |
| LH-90 | $s_g : 1 \rightarrow 0.25$ | 81.08 | 94.92 | 86.28 | 96.53 | 70.02 | 88.05 | 12.64 | 31.43 | 31.46 | 44.73 | 88.88 | 97.06 |
| LH-90 | $s_g : 1 \rightarrow 0.5$ | 81.09 | 94.97 | 86.47 | 96.58 | 70.18 | 88.40 | 13.04 | 33.00 | 32.02 | 45.67 | 89.56 | 97.30 |
| LH-90 | $s_g : 1 \rightarrow 4$ | 80.91 | 95.10 | 86.58 | 96.92 | 70.16 | 88.69 | 11.40 | 30.31 | 32.13 | 46.33 | 89.40 | 97.46 |
| LH-90 | $s_l$ : invert | 81.37 | 95.02 | 86.43 | 96.72 | 70.30 | 88.32 | 13.87 | 33.88 | 32.69 | 46.55 | 89.72 | 97.44 |
| LH-90 | dist $\rightarrow$ dist$^2$ | 80.98 | 95.17 | 86.50 | 96.88 | 70.34 | 88.50 | 11.45 | 30.44 | 32.13 | 46.43 | 89.66 | 97.48 |
| LH-90 | dist $\rightarrow \sqrt{\text{dist}}$ | 80.86 | 94.89 | 86.17 | 96.55 | 69.15 | 88.15 | 12.33 | 31.75 | 31.65 | 45.23 | 88.56 | 97.32 |
| LH-90 | mask: $\infty \rightarrow 0$ | 79.68 | 94.47 | 85.11 | 96.47 | 68.58 | 87.82 | 11.21 | 30.33 | 29.69 | 43.59 | 87.94 | 97.02 |
| LH-90 | dist$\rightarrow$no dist | 80.19 | 94.77 | 85.49 | 96.48 | 69.22 | 88.27 | 11.52 | 30.92 | 31.76 | 46.33 | 88.52 | 97.10 |
| LH-90 | $s_l$ : fix$\rightarrow$learn | 81.35 | 95.06 | 86.55 | 96.64 | 70.40 | 88.55 | 13.08 | 33.03 | 31.93 | 45.88 | 89.56 | 97.20 |

Table 13: LookHere design ablations *with* extrapolation. ViT-B models trained on ImageNet for 150 epochs; trained at $224^2$ and tested at $1024^2$.

| Variant | Change | Val [1] | | ReaL [4] | | v2 [2] | | -A [3] | | -R [5] | | -HR (ours) | |
|---|---|---|---|---|---|---|---|---|---|---|---|---|---|
| | | top-1 | top-5 | top-1 | top-5 | top-1 | top-5 | top-1 | top-5 | top-1 | top-5 | top-1 | top-5 |
| LH-45 | undir$\rightarrow$ 90° | 71.94 | 91.01 | 78.26 | 93.58 | 59.23 | 81.53 | 5.89 | 18.47 | 16.02 | 27.72 | 81.40 | 95.26 |
| LH-45 | undir$\rightarrow$ 180° | 69.97 | 89.42 | 76.05 | 92.44 | 56.19 | 79.03 | 5.33 | 16.71 | 13.70 | 24.44 | 78.72 | 94.12 |
| LH-45 | undir$\rightarrow$no dist | 69.72 | 89.35 | 75.69 | 92.33 | 55.97 | 78.76 | 4.77 | 15.76 | 12.92 | 23.42 | 79.14 | 93.46 |
| LH-90 | undir$\rightarrow$ 90° | 69.39 | 88.99 | 75.67 | 92.00 | 55.82 | 78.68 | 5.00 | 16.40 | 11.93 | 21.96 | 78.10 | 93.46 |
| LH-90 | undir$\rightarrow$ 180° | 68.80 | 88.62 | 74.95 | 91.72 | 53.89 | 77.43 | 4.16 | 14.49 | 12.72 | 22.84 | 76.60 | 92.48 |
| LH-90 | undir$\rightarrow$no dist | 69.24 | 89.40 | 75.29 | 92.46 | 56.03 | 79.38 | 4.84 | 15.65 | 12.67 | 23.08 | 78.24 | 94.08 |
| LH-180 | undir$\rightarrow$ 90° | 64.44 | 85.72 | 70.73 | 89.36 | 49.80 | 73.70 | 3.40 | 12.16 | 8.98 | 17.32 | 73.60 | 90.14 |
| LH-180 | undir$\rightarrow$ 180° | 54.13 | 77.21 | 59.97 | 81.66 | 39.38 | 63.14 | 1.61 | 5.51 | 4.23 | 8.80 | 66.44 | 84.56 |
| LH-180 | undir$\rightarrow$no dist | 66.35 | 86.89 | 72.67 | 90.27 | 51.97 | 75.62 | 4.92 | 14.69 | 9.33 | 17.43 | 74.30 | 91.20 |
| LH-90 | $s_g : 1 \rightarrow 0.125$ | 67.36 | 87.84 | 73.31 | 91.06 | 53.06 | 76.37 | 3.15 | 10.69 | 12.15 | 22.20 | 76.58 | 91.90 |
| LH-90 | $s_g : 1 \rightarrow 0.25$ | 70.46 | 89.68 | 76.43 | 92.52 | 57.11 | 80.17 | 5.43 | 16.19 | 12.45 | 22.25 | 79.32 | 93.14 |
| LH-90 | $s_g : 1 \rightarrow 0.5$ | 72.53 | 90.72 | 78.40 | 93.30 | 59.34 | 81.56 | 7.16 | 20.80 | 13.88 | 24.05 | 79.82 | 93.36 |
| LH-90 | $s_g : 1 \rightarrow 4$ | 55.16 | 78.88 | 61.14 | 83.31 | 41.30 | 66.02 | 2.85 | 9.59 | 5.19 | 10.95 | 68.02 | 86.92 |
| LH-90 | $s_l$ : invert | 70.03 | 89.28 | 76.09 | 92.30 | 55.69 | 79.06 | 6.21 | 19.29 | 11.85 | 21.36 | 76.80 | 92.26 |
| LH-90 | dist $\rightarrow$ dist$^2$ | 65.08 | 87.03 | 71.16 | 90.46 | 51.05 | 75.52 | 3.28 | 12.35 | 10.96 | 20.95 | 74.80 | 92.04 |
| LH-90 | dist $\rightarrow \sqrt{\text{dist}}$ | 66.83 | 87.59 | 72.53 | 90.70 | 52.88 | 76.41 | 3.91 | 12.57 | 10.93 | 19.83 | 76.80 | 92.60 |
| LH-90 | mask: $\infty \rightarrow 0$ | 40.52 | 66.37 | 45.24 | 70.96 | 28.48 | 52.11 | 1.23 | 5.84 | 2.58 | 6.58 | 50.18 | 75.08 |
| LH-90 | dist$\rightarrow$no dist | 44.42 | 69.57 | 49.20 | 74.05 | 31.31 | 55.09 | 1.03 | 5.11 | 6.02 | 13.10 | 53.92 | 78.48 |
| LH-90 | $s_l$ : fix$\rightarrow$learn | 66.13 | 86.72 | 71.96 | 89.82 | 52.54 | 75.71 | 4.44 | 13.80 | 10.59 | 19.27 | 75.82 | 91.66 |

## A.7 Logit Lens

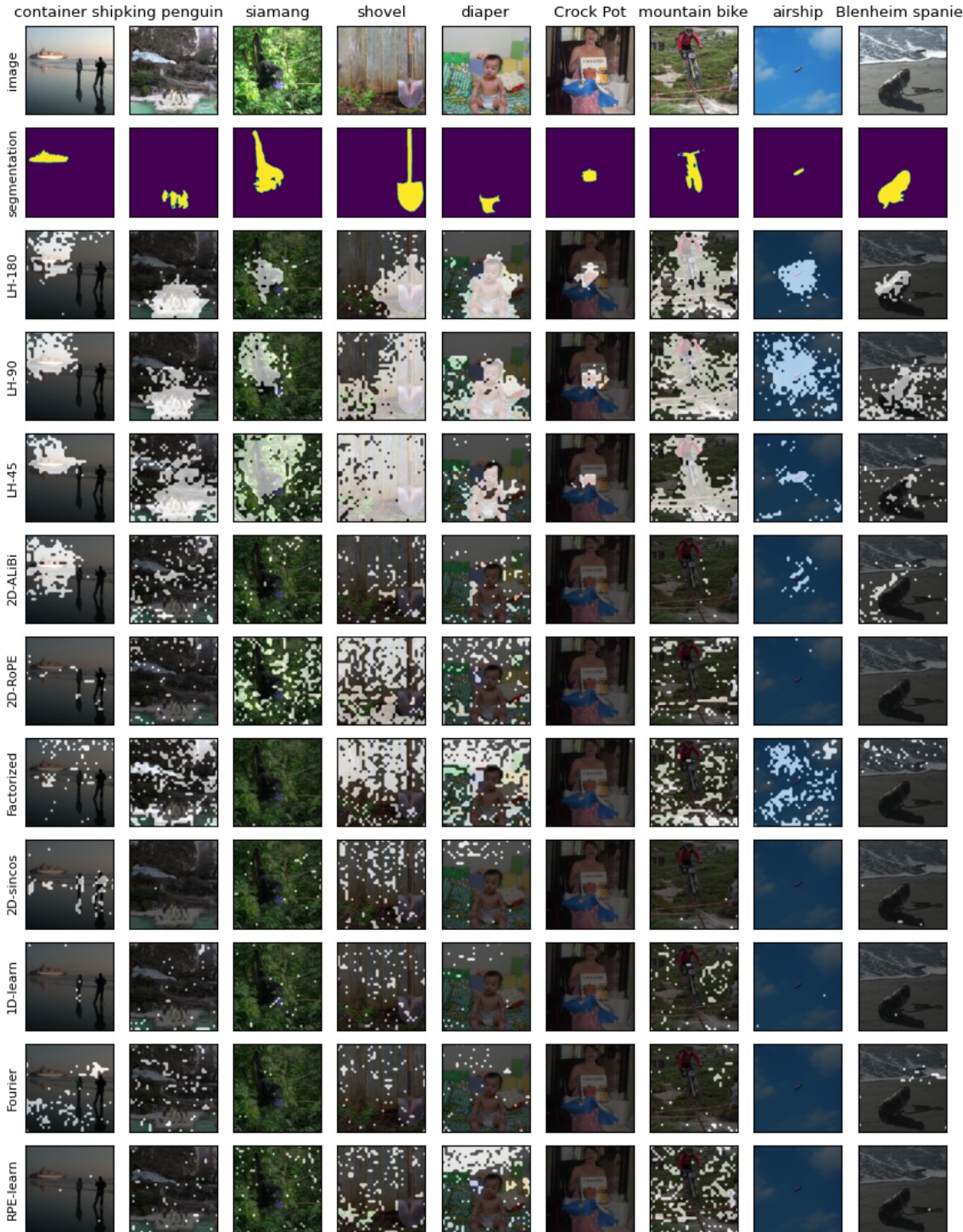

Figure 12: More examples from ImageNet-S and each model's logit lens predictions.

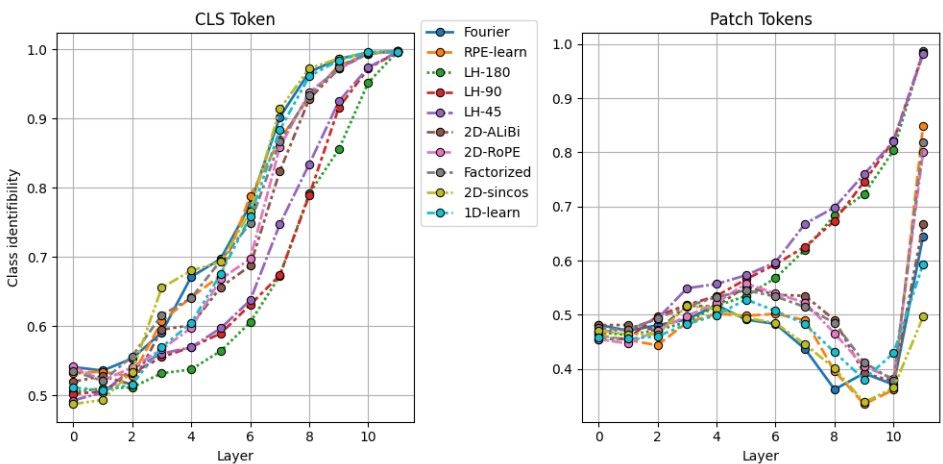

Figure 13: We plot the average class identifiability [89] across the model layers on 1000 images from Val for the class and patch tokens. This is a measure of how recoverable the correct class is from the class projection of the token. The score ranges from 0 to 1, with 1 denoting that the correct class has the highest logits and 0 the lowest.

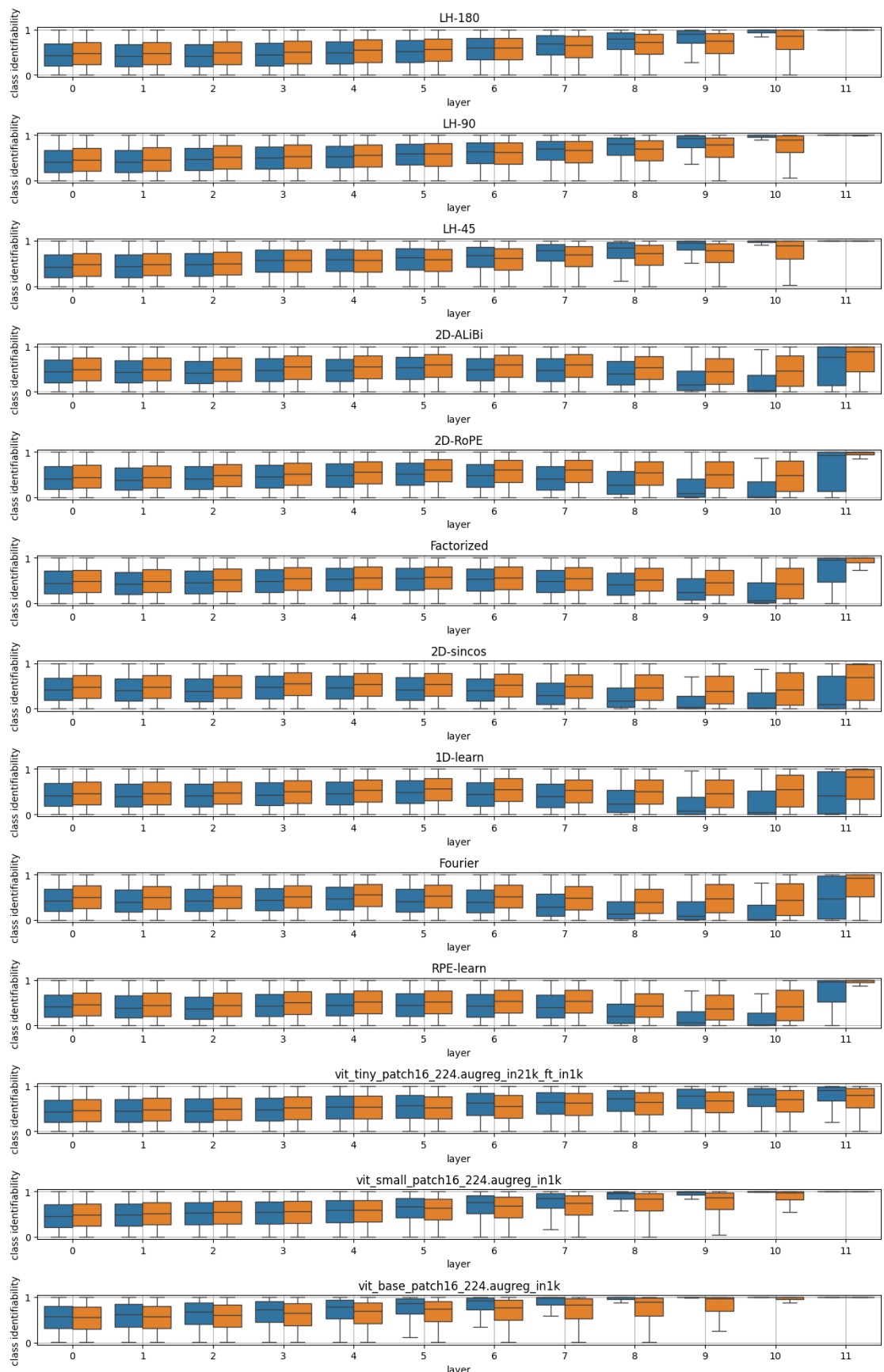

Figure 14: Leveraging the semantic segmentation labels from the ImageNet-S, we compared the identifiability rate of class patches (blue) vs non-class tokens [89] across the model layers on 1000 images from Val. LookHere can discriminate between class and non-class patches. Other positional encodings cannot unless they are trained for much longer.

### A.8 Head Diversity, Attention Distance, Patch Similarity and Head Visualizations

In our paper, we show that LookHere prevents attention collapse measured by JSD (Figure 4). Here, we measure attention diversity using $L_1$ and $L_2$ distance. We also measure attention distances and patch-wise representational similarity — both at $224^2$ px (Figure 15) and at all resolutions tested (Figure 16 & 17).

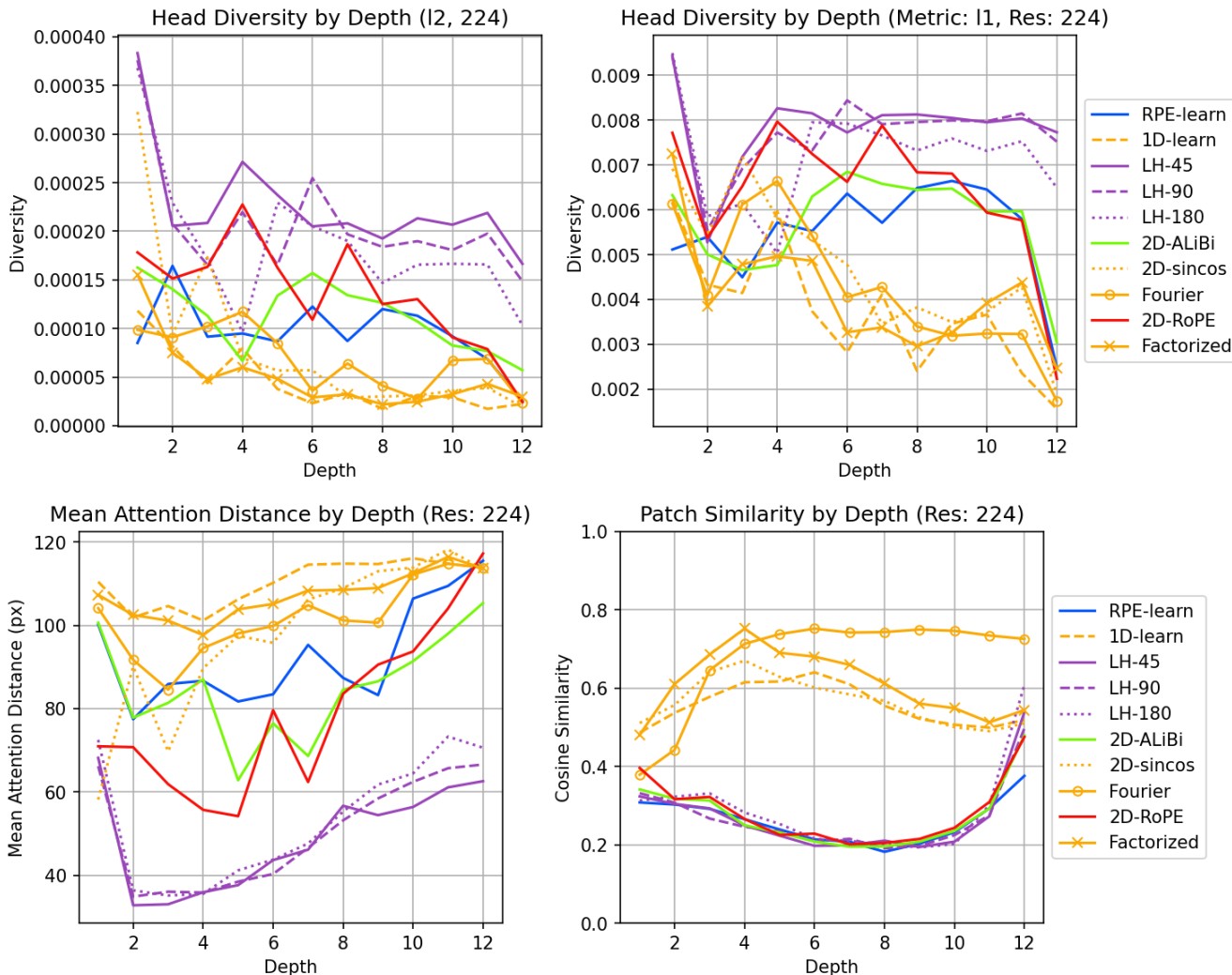

Figure 15: Measurements of head diversity, attention distance, and patch similarity by layer across position encoding methods.

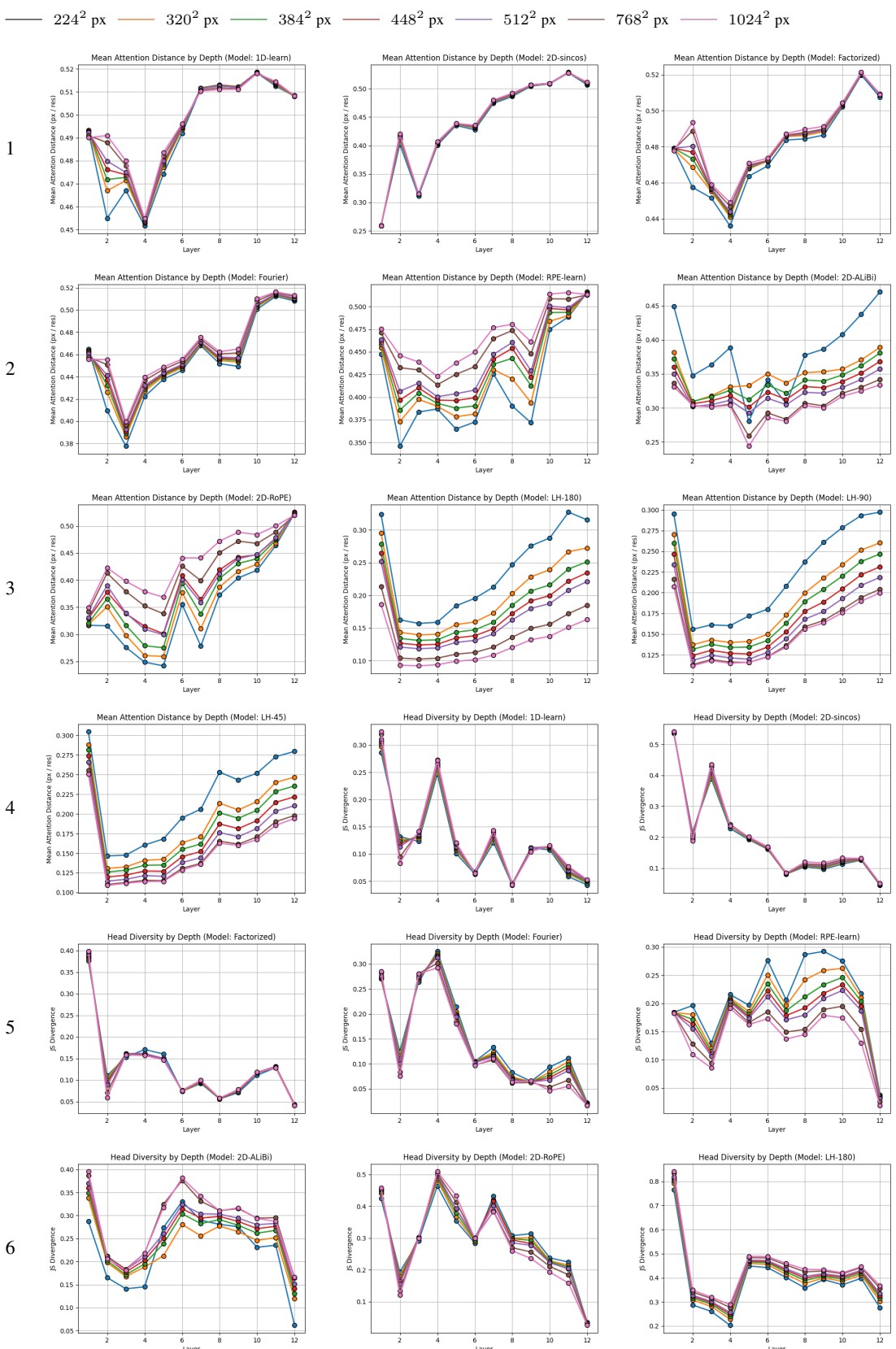

Figure 16: We measure the attention distance per depth, for each model and resolution, by taking the sum of patch distances weighted by attention scores, averaged across heads [(row: 1, col: 1) → (4, 1)]; and the head diversity as the generalized JSD of attention matrix rows for each head, averaged over rows [(4, 2) → (7, 2)]. We report the average over 500 randomly selected images from minival.

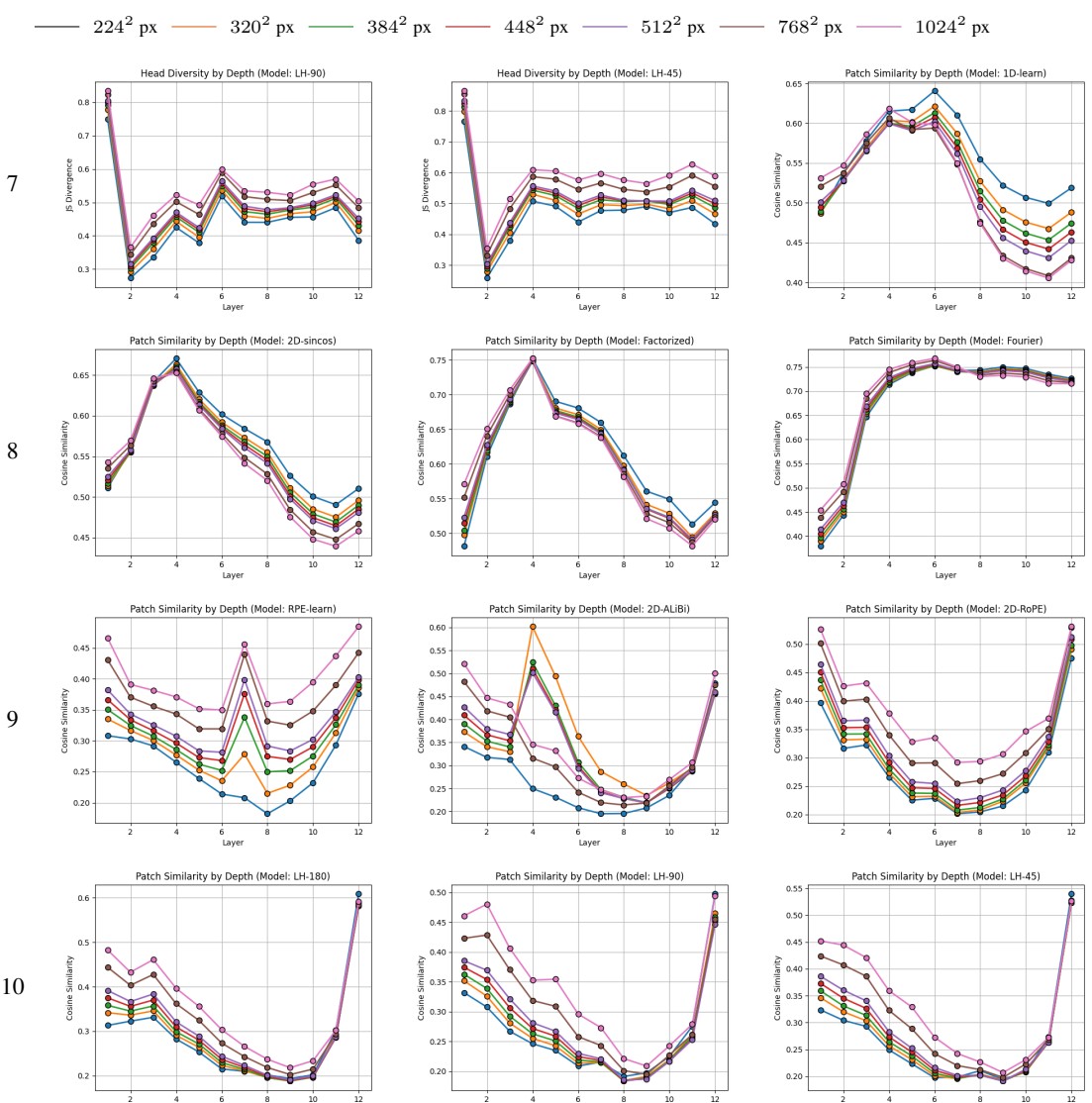

Figure 17: The patch similarity, for each model and resolution, is measured as the average of pairwise cosine-similarities between patch representations in each layer $[(7, 3) \rightarrow (10, 3)]$; we report the average over the same 500 minival images.

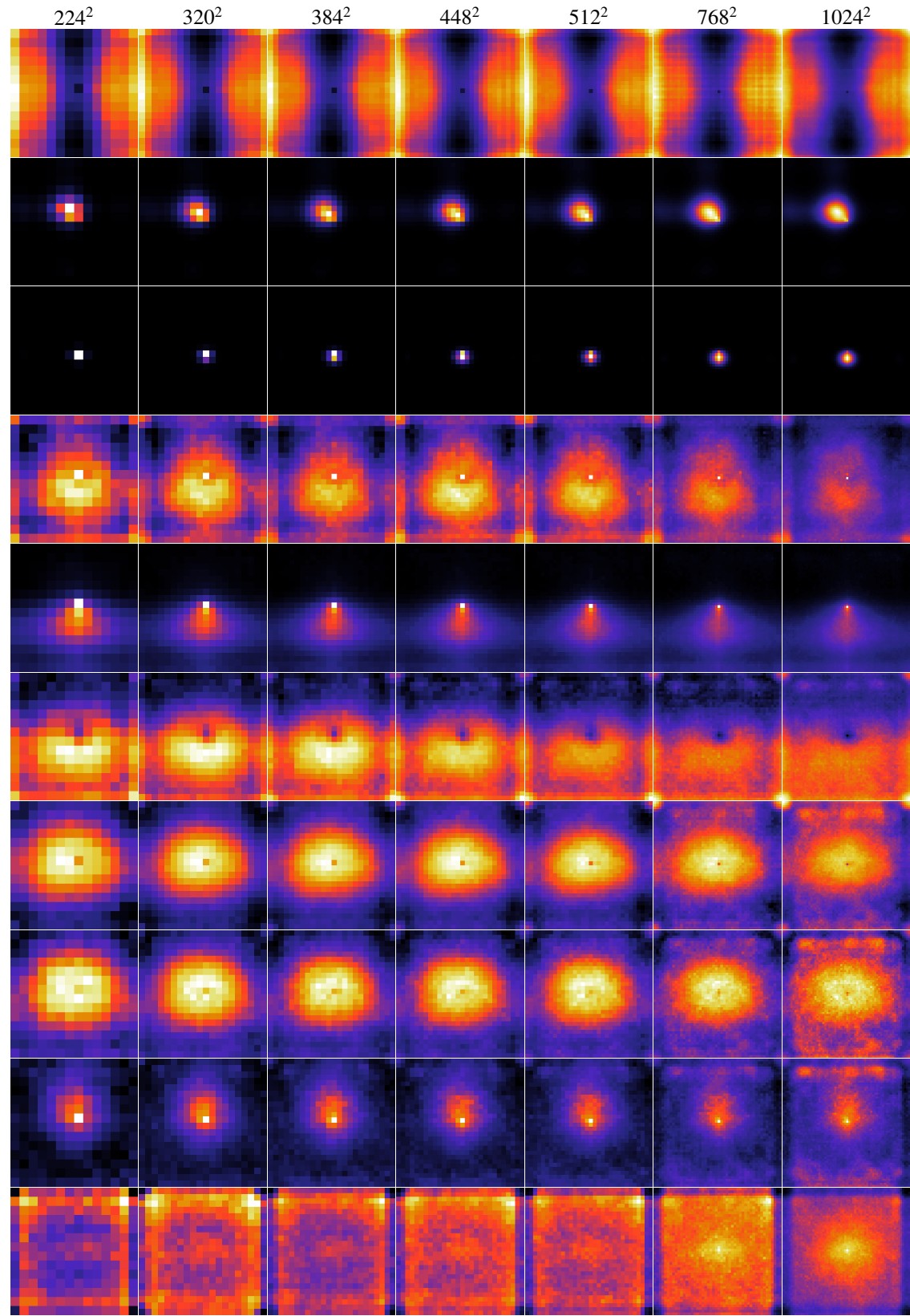

Figure 18: **1D-learn** attention maps of ten attention heads across seven resolutions, where the query is in the center. We use the colormap: ▬▬▬. Averaged over 5k images.

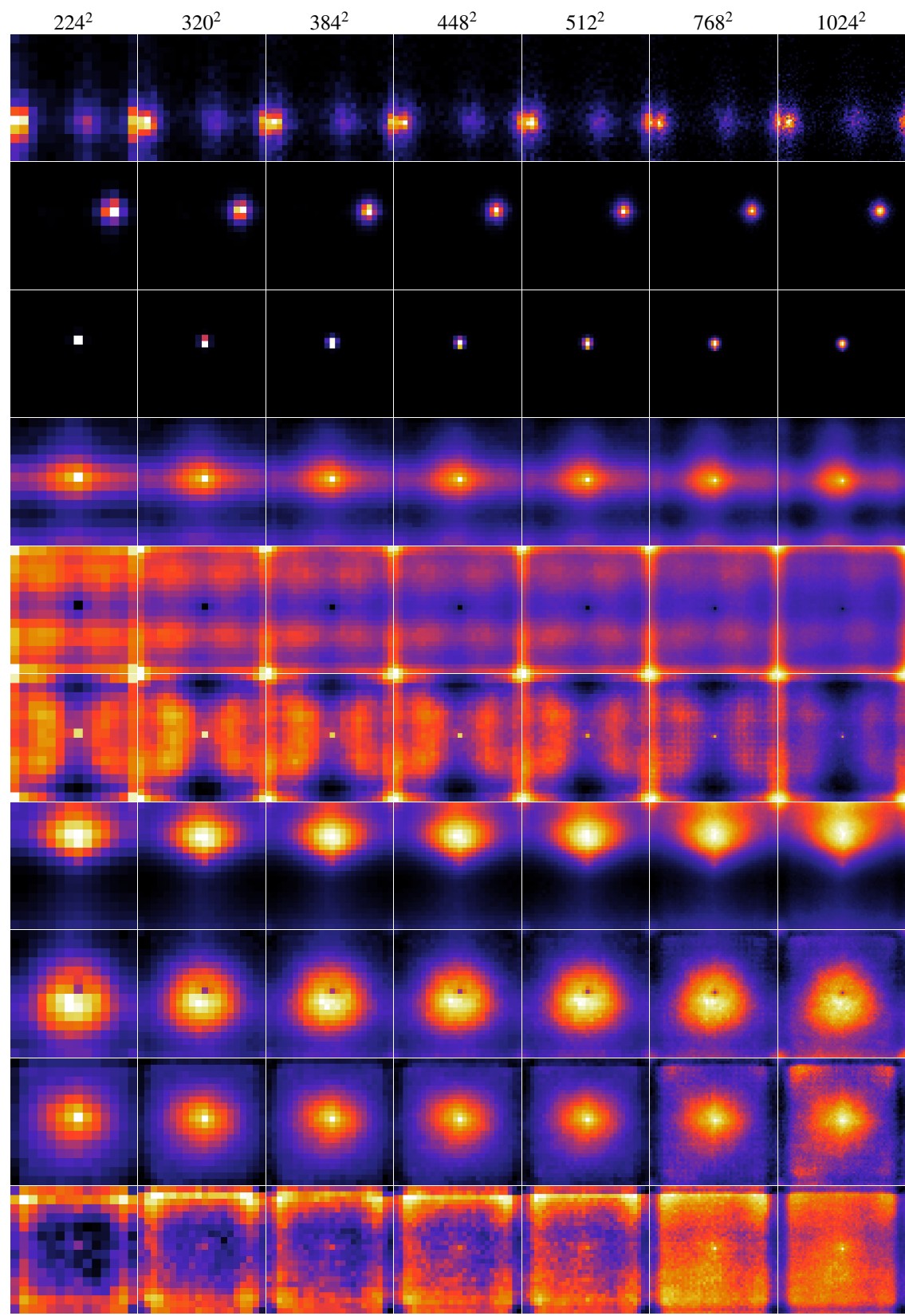

Figure 19: **2D-sincos** attention maps of ten attention heads across seven resolutions, where the query is in the center. We use the colormap: ▬▬▬▬. Averaged over 5k images.

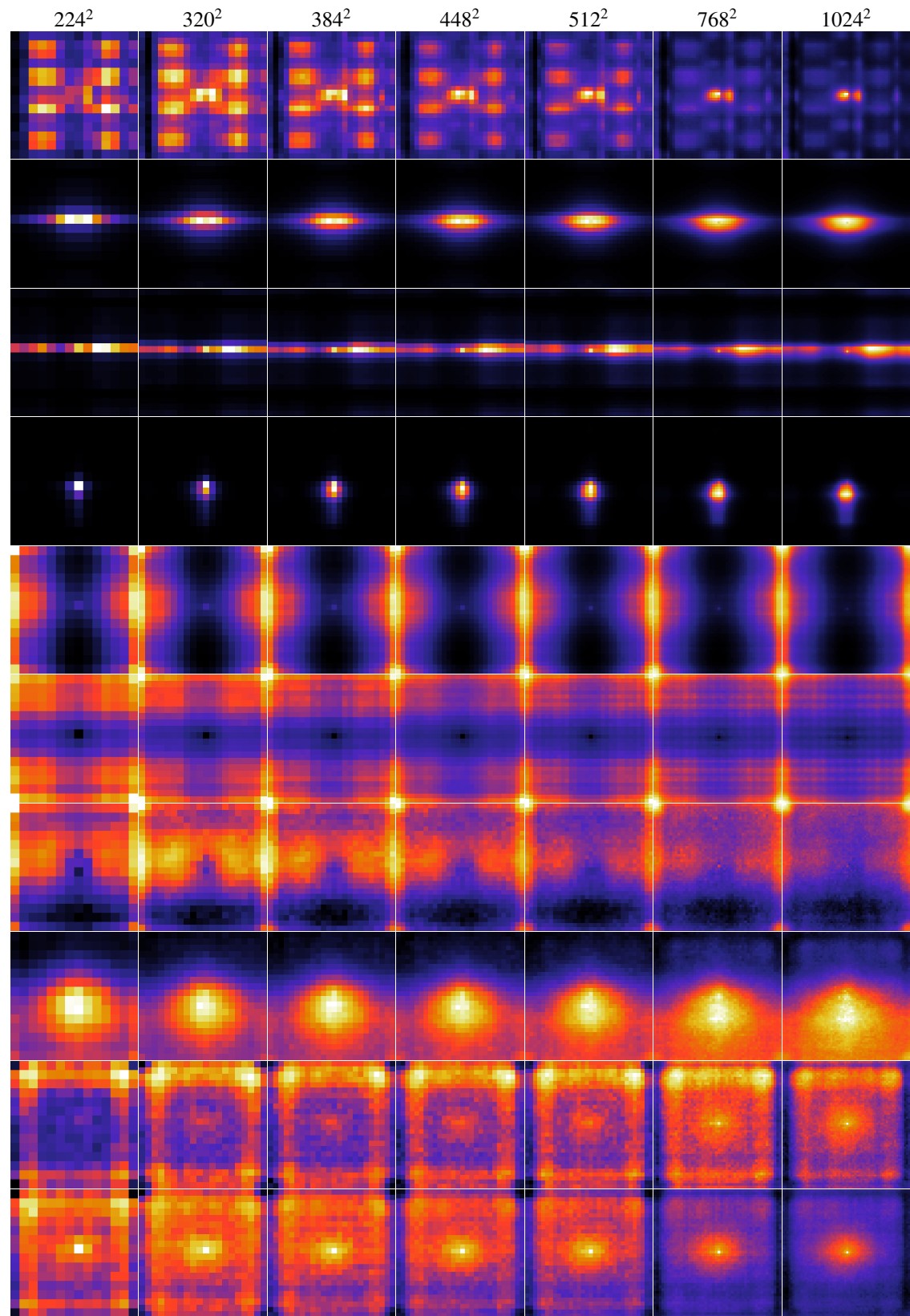

Figure 20: **Factorized** attention maps of ten attention heads across seven resolutions, where the query is in the center. We use the colormap: ▬▬▬▬. Averaged over 5k images.

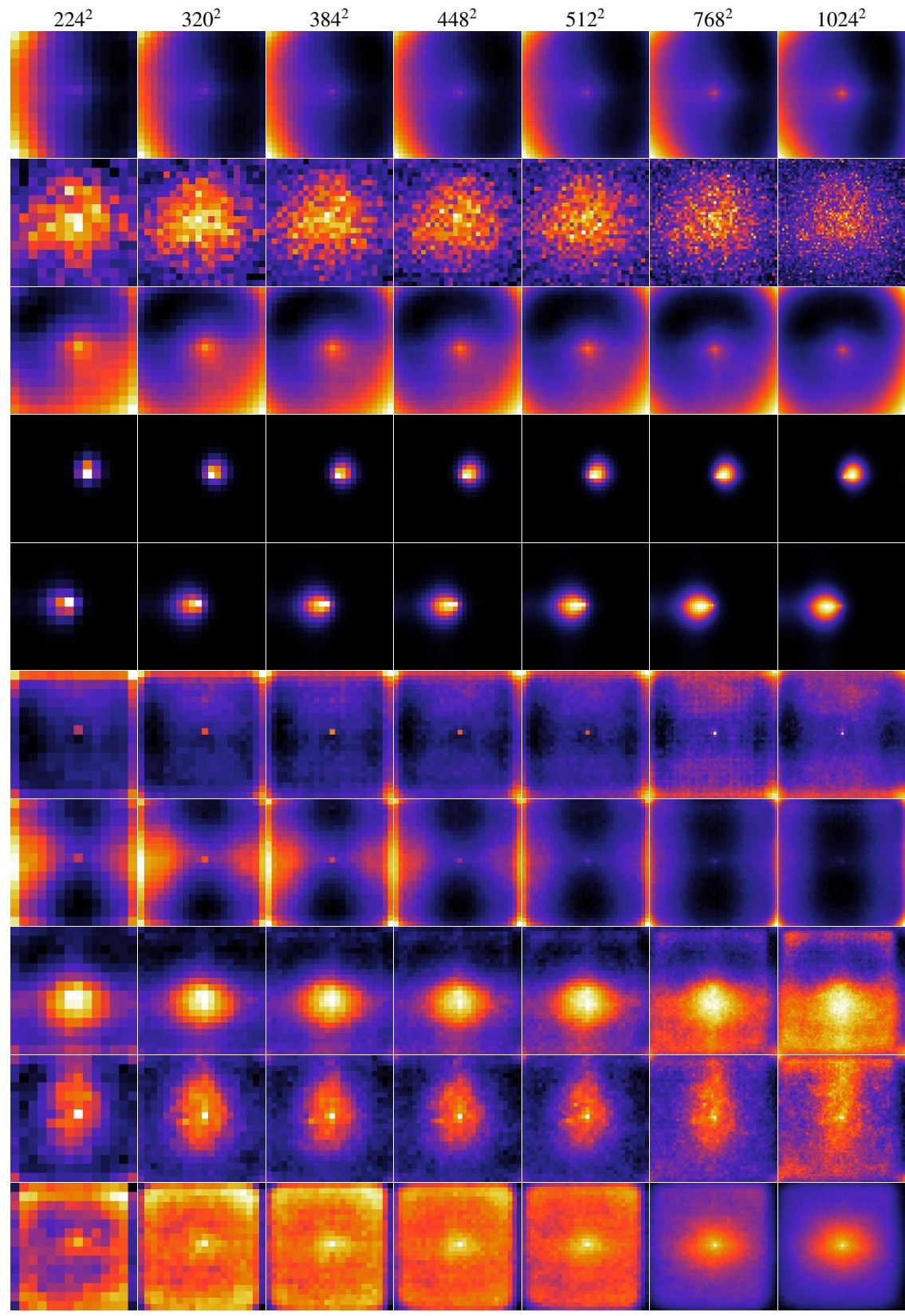

Figure 21: **Fourier** attention maps of ten attention heads across seven resolutions, where the query is in the center. We use the colormap: ▬▬▬▬. Averaged over 5k images.

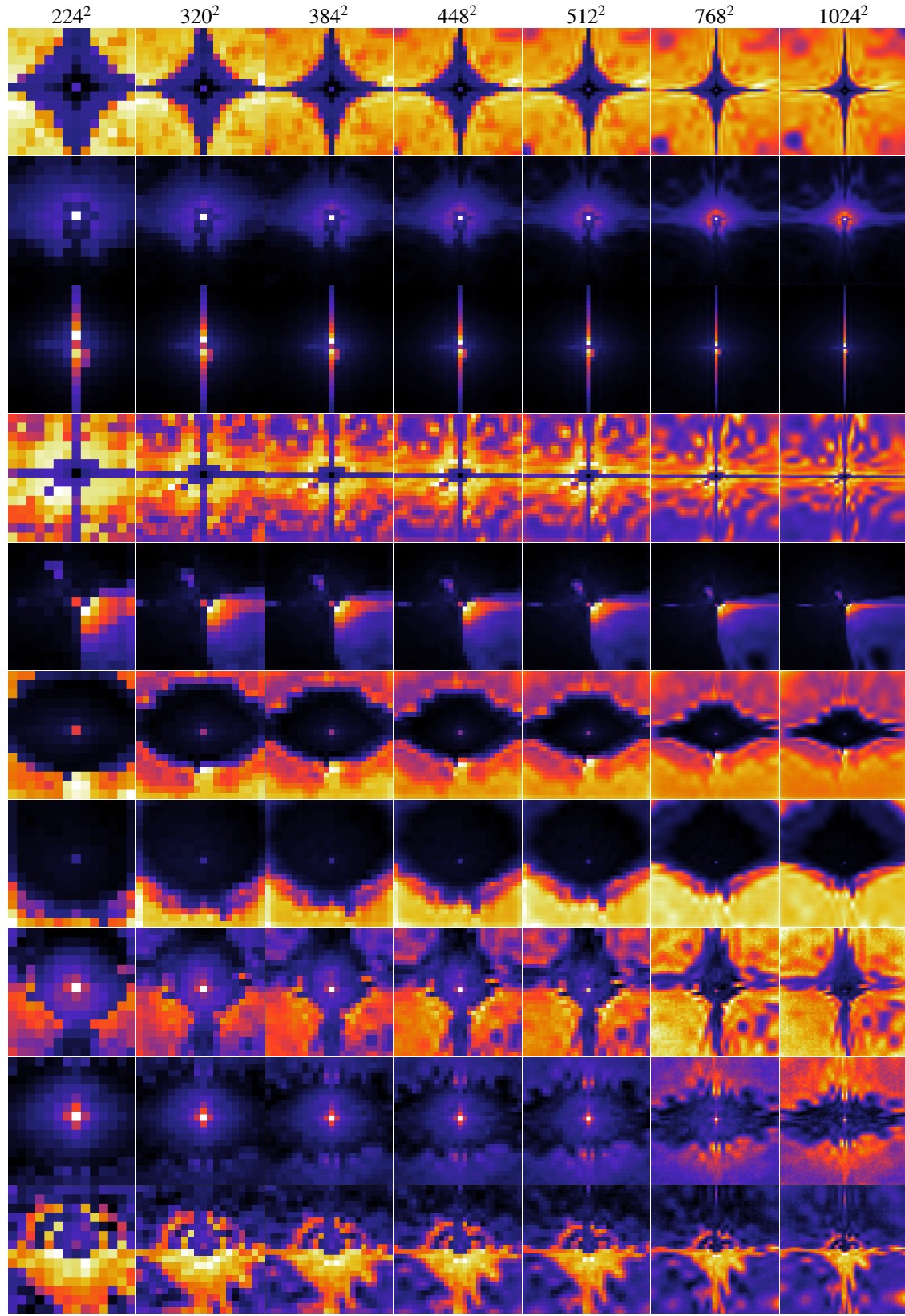

Figure 22: **RPE-learn** attention maps of ten attention heads across seven resolutions, where the query is in the center. We use the colormap: ▬▬▬▬▬▬. Averaged over 5k images.

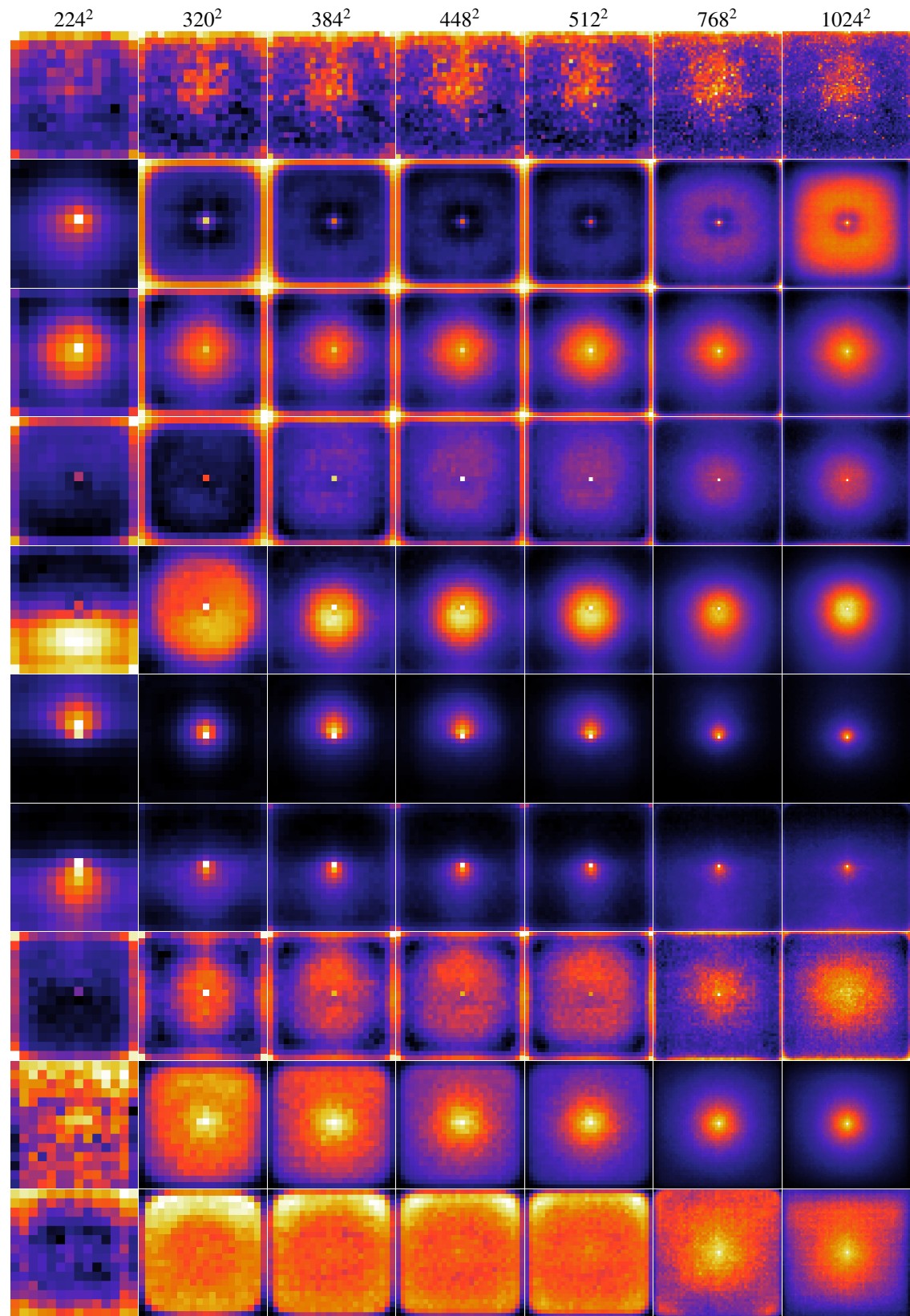

Figure 23: **2D-ALiBi** attention maps of ten attention heads across seven resolutions, where the query is in the center. We use the colormap: ▆▆▆▆▆▆. Averaged over 5k images.

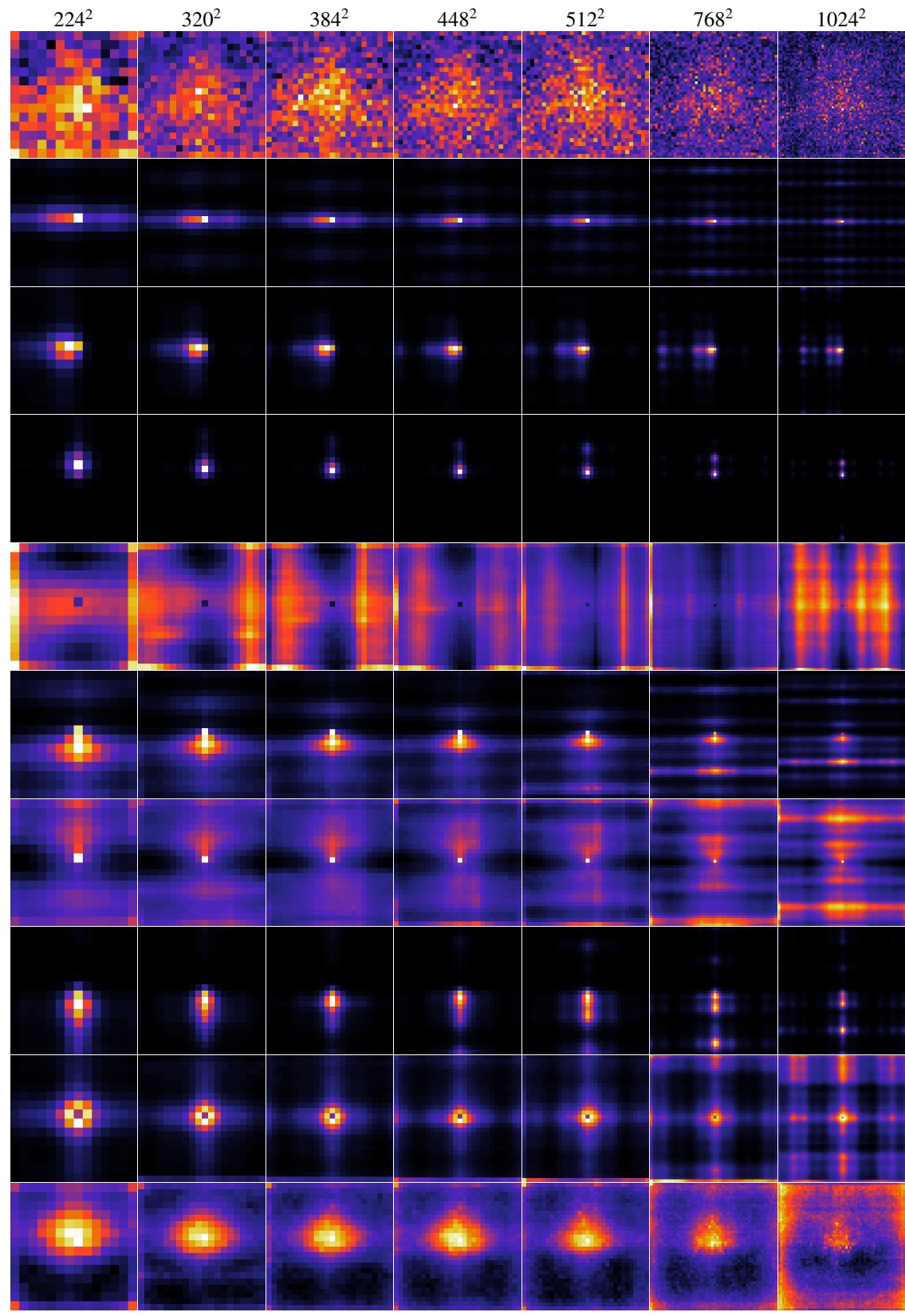

Figure 24: **2D-RoPE** attention maps of ten attention heads across seven resolutions, where the query is in the center. We use the colormap: ▮▮▮▮▮▮. Averaged over 5k images.

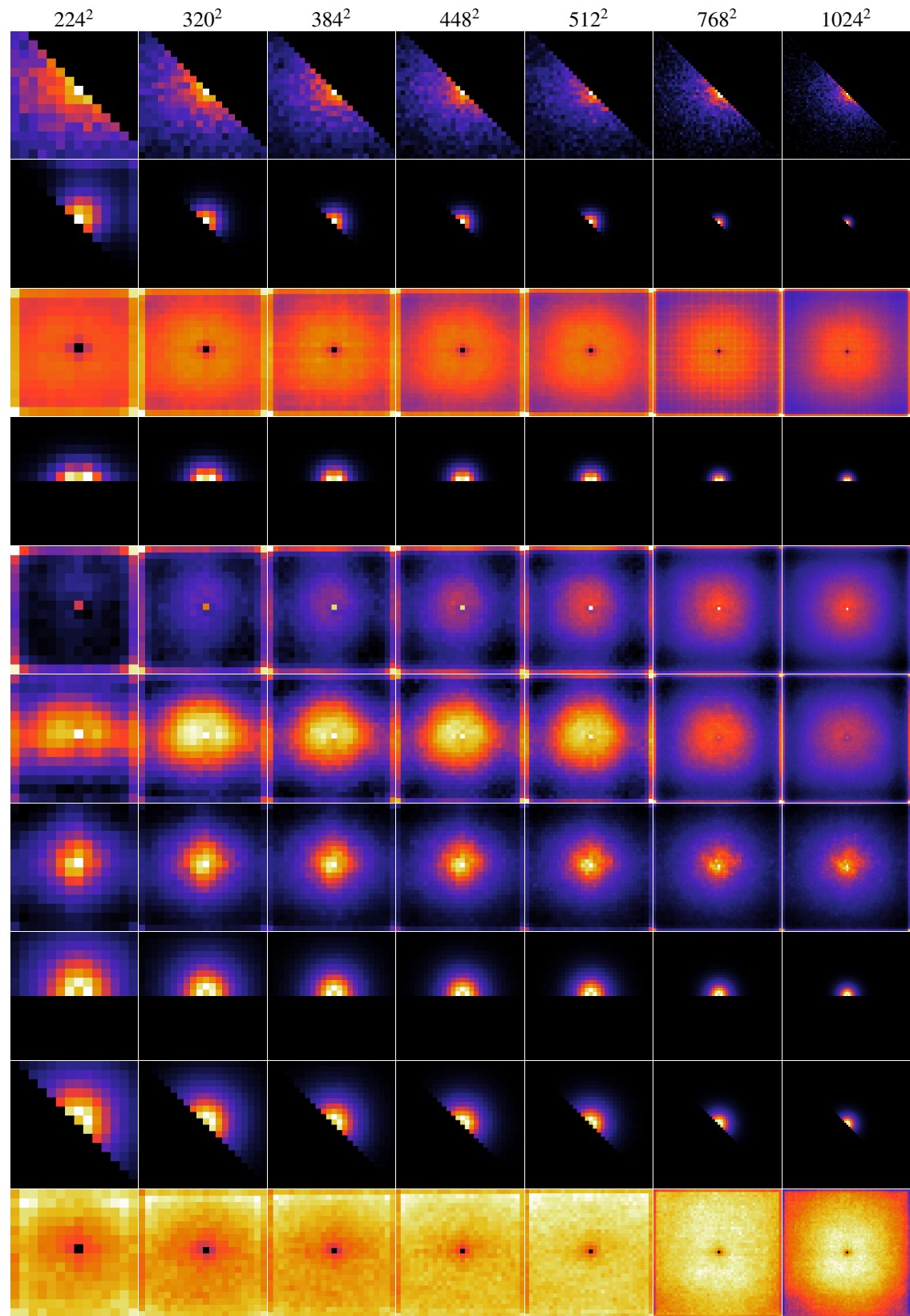

Figure 25: **LH-180** attention maps of ten attention heads across seven resolutions, where the query is in the center. We use the colormap: ▬▬▬▬▬. Averaged over 5k images.

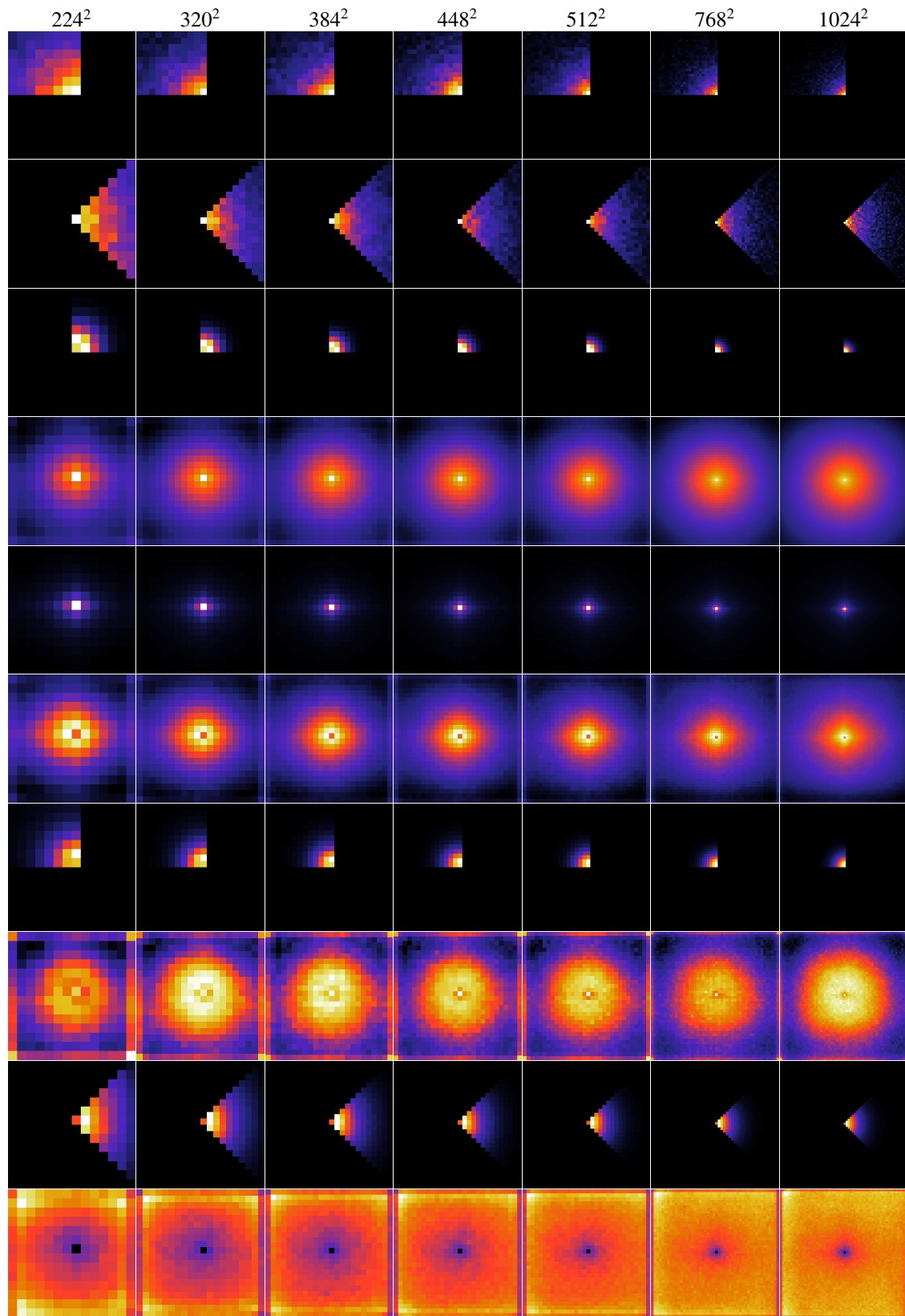

Figure 26: **LH-90** attention maps of ten attention heads across seven resolutions, where the query is in the center. We use the colormap: . Averaged over 5k images.

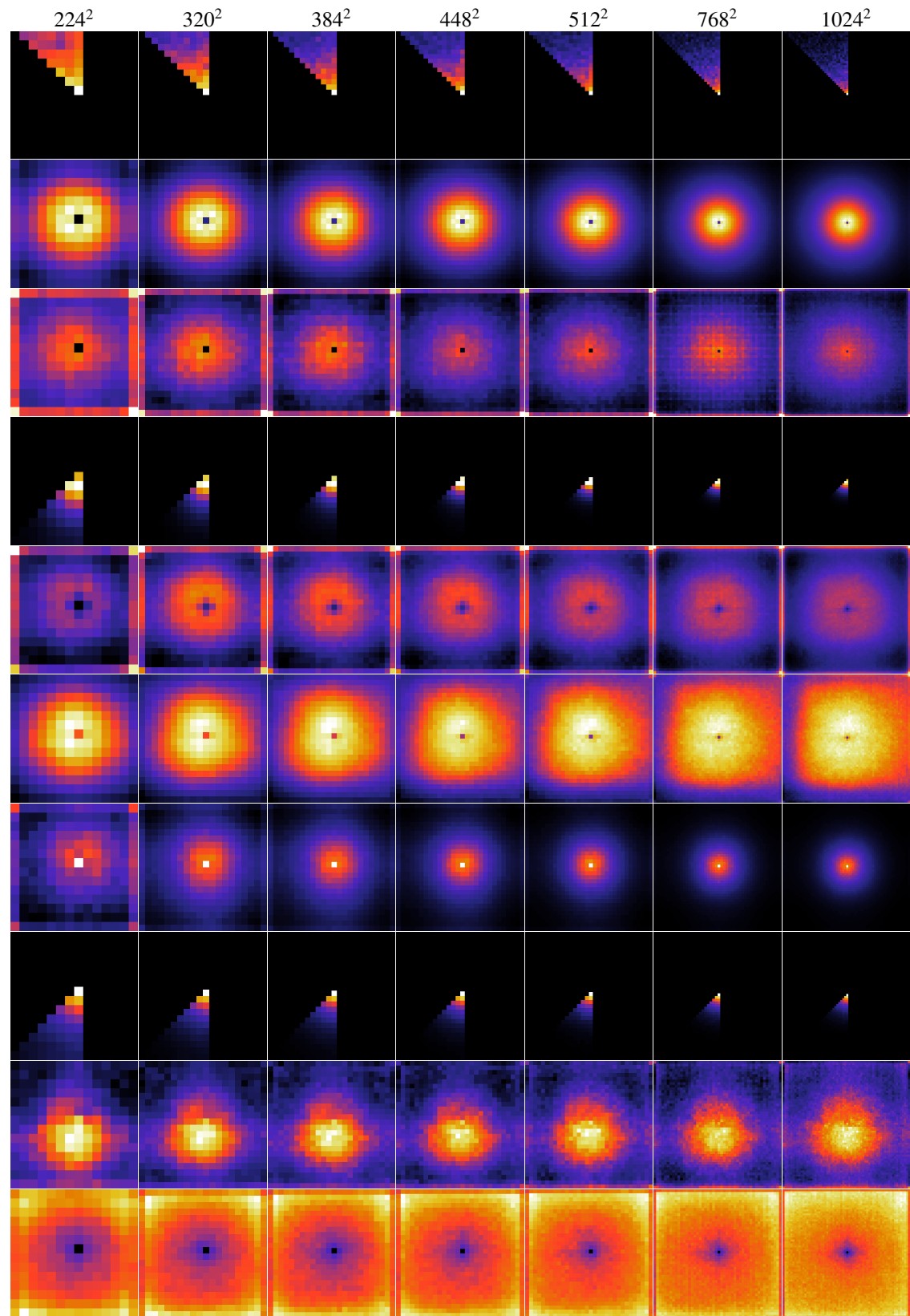

Figure 27: LH-45 attention maps of ten attention heads across seven resolutions, where the query is in the center. We use the colormap: ▇▇▇▇. Averaged over 5k images.

## A.9 Accuracy Gaps and Class-level Effects

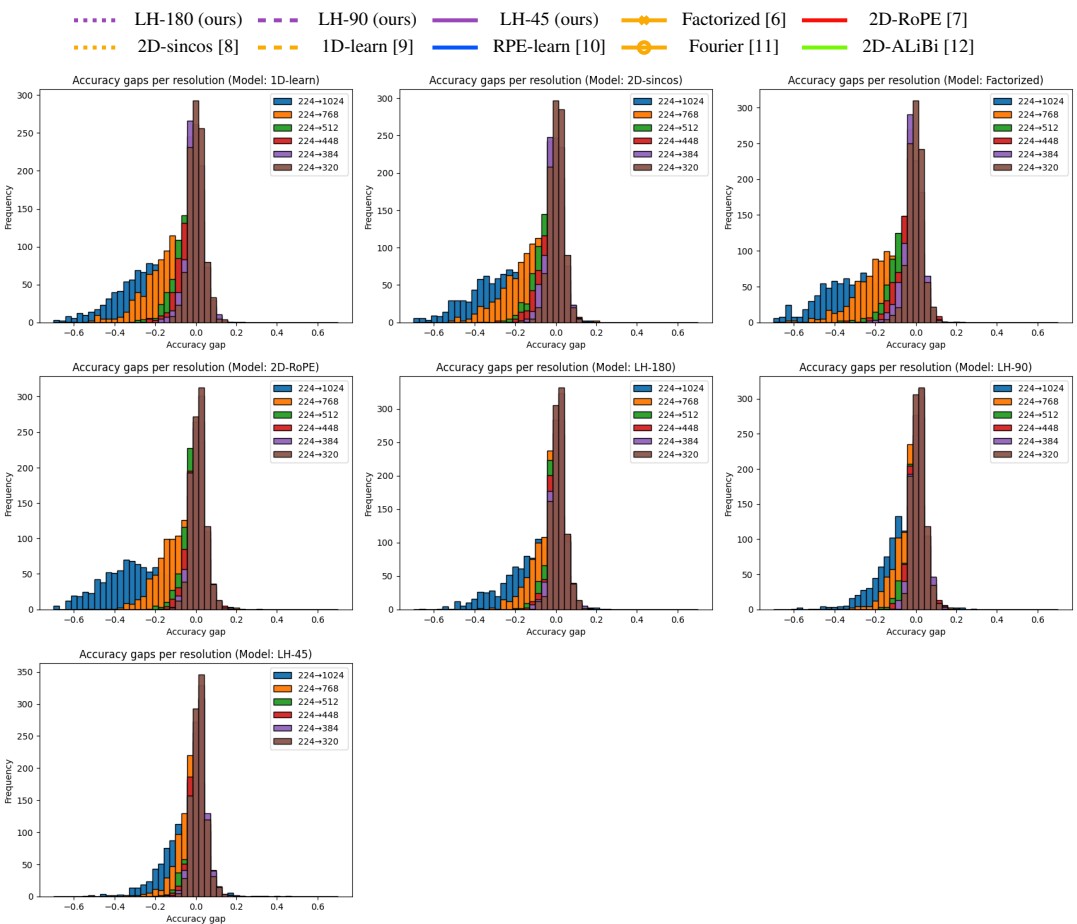

Figure 28: We calculate the difference in accuracy for the first 5k ImageNet [1] examples when extrapolating from $224^2$ px for models trained at $224^2$ px

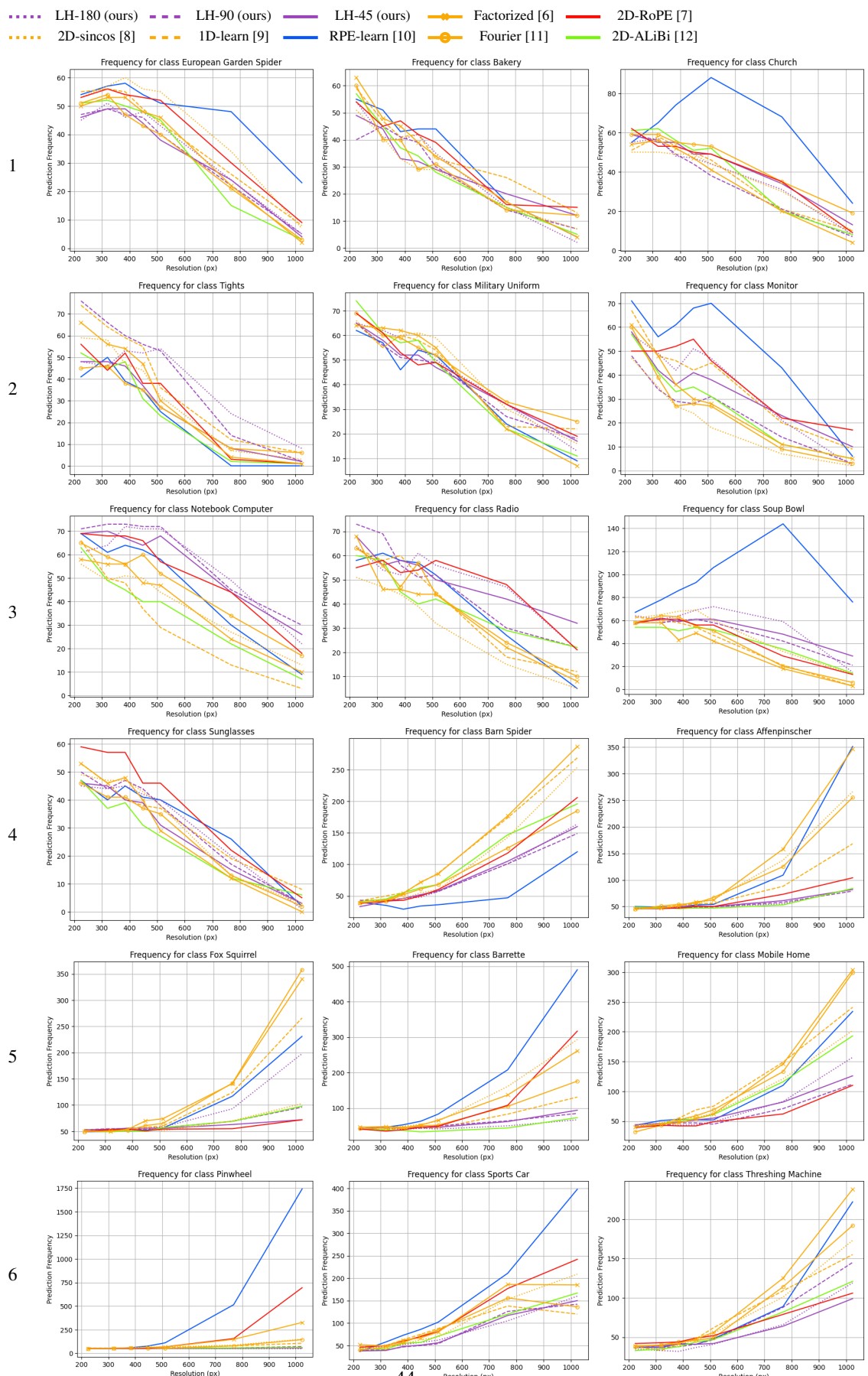

Figure 29: We calculate the prediction frequency for each class for the first 5k ImageNet [1], and plot those frequencies for the classes with the largest decrease $[(1, 1) \rightarrow (4, 2)]$, increase $[(4, 3) \rightarrow (7, 2)]$, and spread $[(7, 3) \rightarrow (10, 3)]$ at $1024^2$ px.

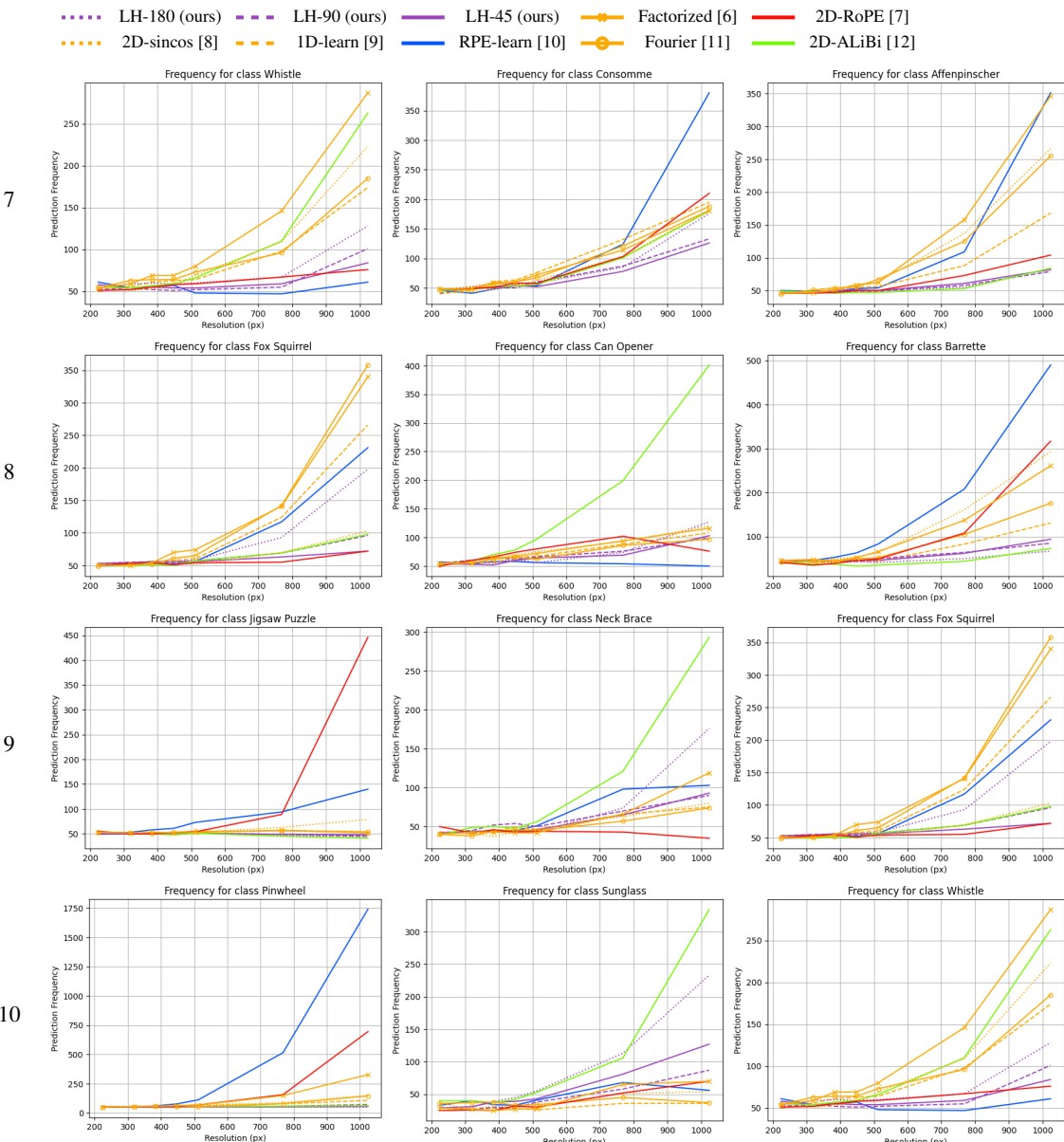

Figure 30: A continuation of increase, and spread plots.

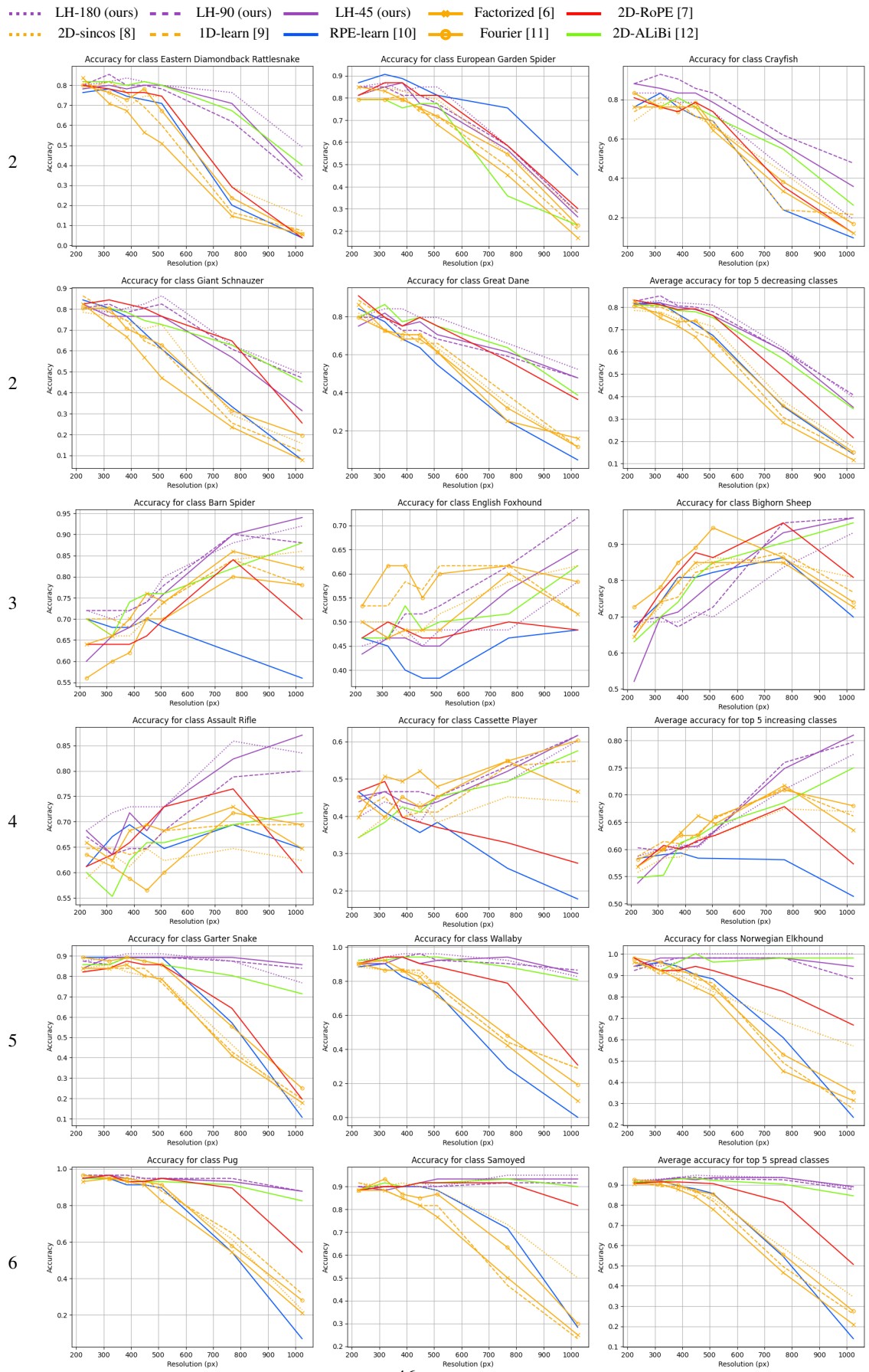

Figure 31: We calculate the prediction accuracy for each class among the first 5k ImageNet [1] examples, and plot these accuracies for classes with the top five classes with the largest decrease $[(1, 1) \rightarrow (2, 3)]$, increase $[(3, 1) \rightarrow (4, 3)]$, and spread $[(5, 1) \rightarrow (6, 3)]$ at $1024^2$ px, along with the average across all five.

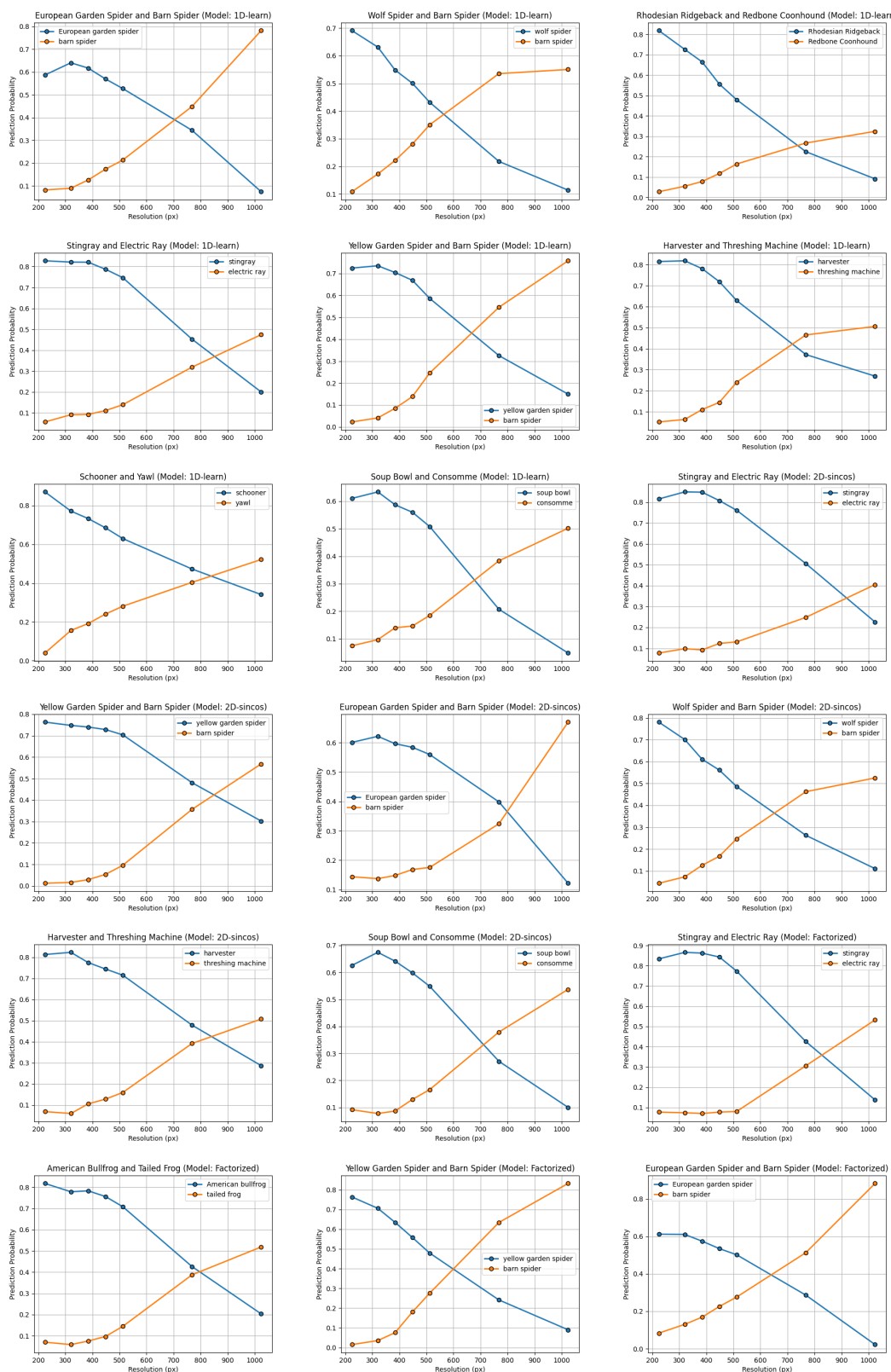

Figure 32: We find and plot cross-plots of class pairs that confuse models during extrapolation, indicated by a transfer in prediction probabilities. For the subset of ImageNet [1] examples, within the first 5k, with true class $X$ (blue) we select pairs $(X, Y)$ where $P(X|224^2) - P(X|1024^2) + P(Y|1024^2) - P(Y|224^2)$ is maximized.

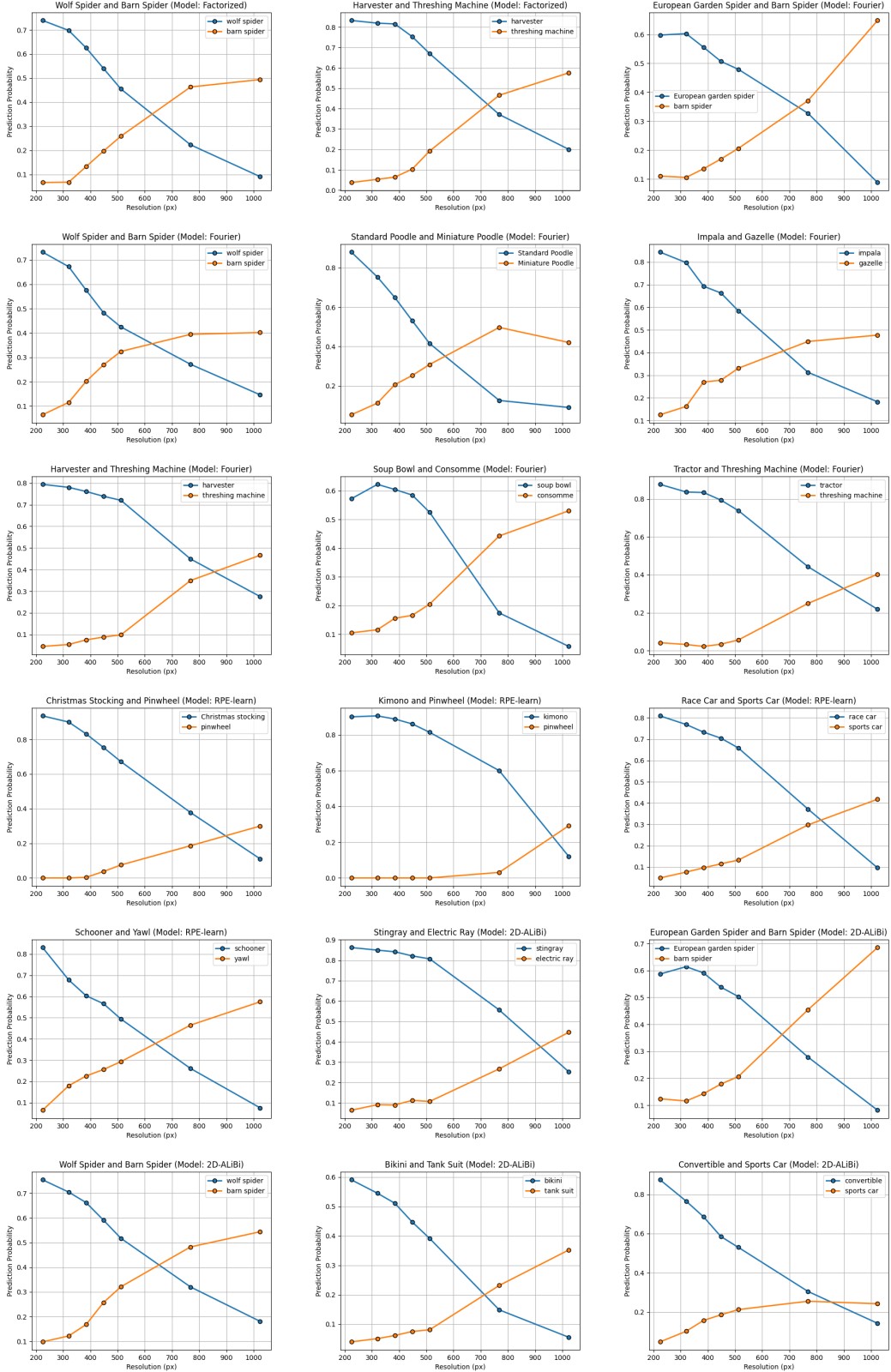

Figure 33: Cross-plots continued.

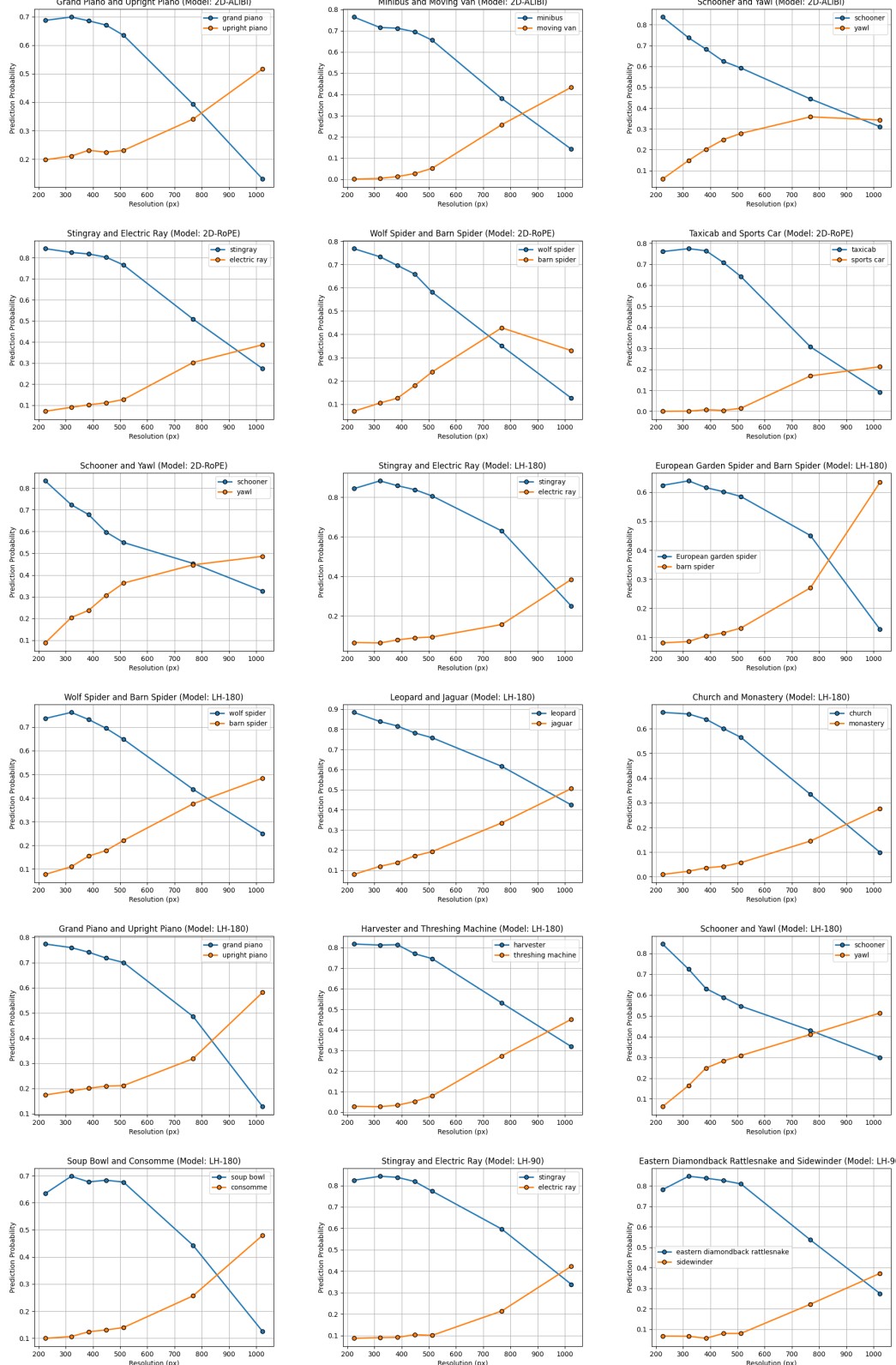

Figure 34: Cross-plots continued.

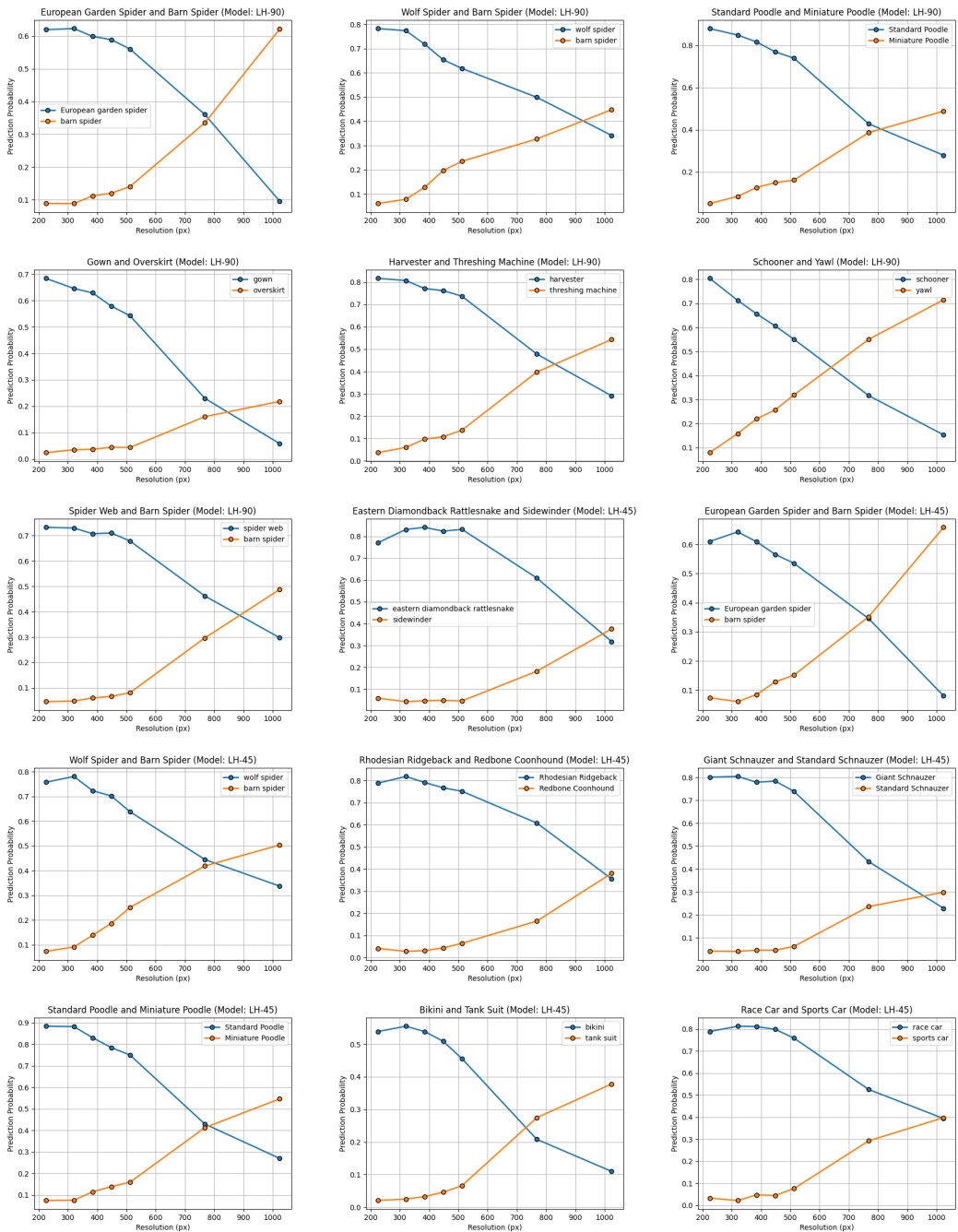

Figure 35: Cross-plots continued.

