# OpenReview forum: "LookHere: Vision Transformers with Directed Attention Generalize and Extrapolate"
_NeurIPS.cc/2024/Conference — NeurIPS 2024 poster_

### Official Review · Reviewer_irRJ · 2024-07-03

**Soundness:** 4
**Presentation:** 4
**Contribution:** 4
**Rating:** 7
**Confidence:** 5

**Summary:**

This paper proproses LookHere, a novel positional encoding method for Vision Transformers for dealing with high-resolution images. Specifically, LookHere explicitly constratin attention heads to attend certain directions via 2D masks. With comprehensive experiments. LookHere demonstrates strong performance across different benchmarks. It also achieves better performance than existing positional encoding methods on high-res images.

**Strengths:**

1. The idea is very clear. High-res images introduce new challenges on positoinal encoding due to the mismatch between the training and inference time. By introducing 2D attention masks, we can assign different heads a direction, such that they obtain explicit bias and will know how to attend on high-res images.
2. The experiemnts are very comprehensive and strong, which well demonstrate the effectiveness of the proposed method.
3. This paper is well-written and in a very good shape.

**Weaknesses:**

1. Actually the authors have clearly dicussed the weakness of their method, e.g. hand-designed masks, single scale of ViT. Beside that, have the authors considered a discussion with ConViT [50]? ConViT introduces convolutional bias to attention head, which also restricts the head to only attend to certain areas.
2. (Minor) It would be more promising to see if this technique can be applied into multi-modal LLMs.

**Questions:**

See the weakness.

**Limitations:**

Yes

---

> ### Author Rebuttal · Authors · 2024-08-07
>
> We thank the reviewer for their thoughtful review. In particular, their recognition of our clear ideas, comprehensive experiments, and writing, as well as suggestions to improve the paper. The review asks two questions, which we address in this rebuttal.
>
> $\textbf{Q1)}$ Have you considered a discussion with ConViT [50]?
>
> $\textbf{A1)}$ We agree there are some conceptual similarities between LookHere and ConViT — e.g., local and directional inductive biases. In ConViT, they arise from carefully initializing relative position encoding such that attention behaves like convolutions early in training. ConViT can learn to overcome the initially restricted views via learnable gating. In our estimation, ConViT is most similar to RPE-learn, with a clever and effective initialization scheme. We thank the reviewer for raising this point, and we will add a short discussion of ConViT to our final paper, should it be accepted.
>
> $\textbf{Q2)}$ Can you apply this technique to multi-modal LLMs?
>
> $\textbf{A2)}$ We do not have the resources to train multi-modal LLMs, for instance vision-language models (VLMs) with LookHere position encoding — at least during this rebuttal period. Given our extensive experiments, we fully expect that LookHere can significantly improve the extrapolation ability of VLMs. We also believe LookHere may improve VLMs in other ways. For example, compositional reasoning is a significant limitation in VLMs [R1, R2, R3]; e.g., they may find it challenging to differentiate between “a bed to the left of a dog” and “a dog to the left of a bed.” LookHere explicitly encourages ViTs to learn direction-aware representations since each head attends to a specific direction. We appreciate the reviewer’s suggestion and are excited to bring explicit directional inductive biases to VLMs in future work!
>
> [R1] “SugarCrepe: A benchmark for faithful vision-language compositionality evaluation”, Hsieh et al, NeurIPS 2023
>
> [R2] “Image Captioners Are Scalable Vision Learners Too”, Tschannen et al., NeurIPS 2023
>
> [R3] “CREPE: Can Vision-Language Foundation Models Reason Compositionally?”, Ma et al, CVPR 2023

---

> > ### Comment · Reviewer_irRJ · 2024-08-12
> > **Official Comment from Reviewer**
> >
> > Thanks for the authors' rebuttal. They have addressed my concerns.

---

### Official Review · Reviewer_Fbdg · 2024-07-09

**Soundness:** 3
**Presentation:** 2
**Contribution:** 3
**Rating:** 7
**Confidence:** 5

**Summary:**

This paper introduces Lookhere, a novel mask-based positional encoding designed to address the performance degradation of ViT when resolution changes. Lookhere restricts the receptive field of each head in the attention mechanism and enhances the diversity of information perceived by each head. Extensive experiments have demonstrated the effectiveness of Lookhere. Compared to other positional encodings, Lookhere has significantly better resolution scalability.

**Strengths:**

1. The author's writing and illustrations are very clear, making it easy to understand and follow the method.
2. The authors conducted numerous experiments, thoroughly validating the effectiveness of Lookhere.
3. The issue explored by the authors, namely the resolution generalization of vision models, is a very important but rarely investigated area. The authors' work could potentially guide future research in this direction.

**Weaknesses:**

In summary, I believe this is a very solid work with no obvious issues. I have only one personal concern. From Figure 6, it can be observed that the essence of the proposed positional encoding seems to be limiting the receptive field of each token, so that when the resolution changes, the number of tokens each token attends to does not change too drastically. However, this approach seems to reduce the model's overall receptive field. As a result, the fine-tuning results at a resolution of 384 do not show a particularly significant advantage compared to other positional encodings. I suspect that as the fine-tuning time increases, this advantage may disappear or even be worse than RoPE. Could the authors fine-tune for a longer period (for example, 30 epochs) to verify that this approach does not negatively impact the model's accuracy?

**Questions:**

Please refer to the weakness

---

> ### Author Rebuttal · Authors · 2024-08-07
>
> We thank the reviewer for their thoughtful review. In particular, their recognition of our writing and illustrations, extensive experiments, and the importance of resolution generalization in ViTs, as well as suggestions to improve the paper. The review lists one concern, which we address in this rebuttal.
>
> $\textbf{Q1)}$ Will the 2D masks that limit each token’s receptive field harm performance if finetuned for longer, relative to 2D-RoPE?
>
> $\textbf{A1)}$ First, a clarification: each of the eight directed heads are restricted to attend in a given direction, but the concatenation of all heads, including the four unrestricted heads, results in each token attending to the entire image. Second, the primary design motivation and performance advantage is LookHere’s extrapolation ability. We view LookHere’s other features, like performance at the training resolution and adversarial robustness, as an outcome of our 2D masks acting like a regularizer by encouraging attention diversity; in fact, we notice that LookHere consistently achieves higher training loss and lower validation loss than 2D-RoPE (which is the desired outcome of a regularizer).
>
> Regarding finetuning performance at 384, we believe LookHere’s ~1% performance advantage on ImageNet is significant (many papers have been published at NeurIPS showing a 1% improvement on ImageNet). That being said, this advantage at 384 might be due to LookHere’s inductive biases that could potentially harm performance when training for longer — as the reviewer points out. As requested by the reviewer, we finetune LookHere-90 and 2D-RoPE for 30 epochs on ImageNet at 384x384 px. The gap narrows significantly; LookHere-90 achieves 83.18% / 88.31% and 2D-RoPE achieves 83.18% / 88.00% on ImageNet-Val / ReaL. Interestingly, LookHere-90 achieves comparable results — 83.08% / 87.99% (Table 5 in our main submission) — after 5 epochs of finetuning, underscoring LookHere’s sample efficiency. Since finetuning on ImageNet for 30 epochs at 384x384 is expensive — roughly 65 A100 hours or 90% the cost of training for 150 epochs at 224x224 — we firmly believe the ability to quickly finetune models at higher-resolutions is significantly valuable to users.
>
> However, the reviewer raises an excellent question: Do LookHere’s inductive biases achieved via attention masks and distance penalties, that improve ViT sample efficiency and enable extrapolation, eventually limit its performance? To answer this question we train LookHere-90 and 2D-RoPE models from scratch for 600 epochs using function matching [R1] with a highly accurate teacher. LookHere-90 achieves 84.94% / 89.39% and 2D-RoPE achieves 85.06% / 89.39% on ImageNet-Val / ReaL. These results are around the empirical upper bound of ViT-B/16 models trained on ImageNet-1k at 224x224 px. Thus, we demonstrate that LookHere’s inductive biases do not limit its performance. We also find that this LookHere model outperforms this 2D-RoPE model by 15% when tested on ImageNet-Val at 1024x1024 px — maintaining a vast extrapolation advantage.
>
> [R1] “Knowledge distillation: A good teacher is patient and consistent”, Beyer et al., CVPR 2022

---

> > ### Comment · Reviewer_Fbdg · 2024-08-11
> > **final review**
> >
> > Thanks the author for the rebuttal. My concerns are well addressed. I have raised my score to 7.

---

### Official Review · Reviewer_cHUb · 2024-07-11

**Soundness:** 3
**Presentation:** 4
**Contribution:** 4
**Rating:** 8
**Confidence:** 3

**Summary:**

The authors explore an alternative to existing positional encoding methods for vision transformers. Within attention operations, 2D attention masks (subtracted from attention matrix i.e. key-query inner product, prior to softmax) limits feature mixing to fixed fields of view pointed in different directions. ViTs resulting from proposed method exhibit improved performance across diverse tasks, especially when extrapolating to input image dimensions larger than the train data. The authors also contribute a new high-resolution ImageNet dataset variant for evaluation.

**Strengths:**

1. The paper is well written with clear motivation, thorough discussion of prior works, and detailed explanation of methodology.
2. The proposed method is simple and elegant, providing better OOD performance to ViTs. This can be highly useful to the community.
3. Rigorous experimentation including hyper-parameter tuning to all prior works on a common training setup.
4. Extensive evaluations and ablations to establish usefulness of proposed method and to highlight its various interesting aspects.
5. Useful high-res ImageNet dataset that is already released anonymously (in appendix)

**Weaknesses:**

1. **Translation Equivariance:** Authors claim this to be a result of proposed method. However, this is not proved theoretically or demonstrated through experiments. Please explain this in method section, compare to CNNs, and provide experiments to quantify this is possible. In case I missed something related from appendix, please highlight it better in main paper.

2. **[Minor] Adversarial Attacks:** Only FGSM - maybe try newer methods like AutoAttack?

**Questions:**

1. **Learning Parameters**: Have you tried learning the slope values? And maybe even the entire attention masks? These could be learned using some SSL setup like MAE.

**Limitations:**

Discussed.

---

> ### Author Rebuttal · Authors · 2024-08-07
>
> We thank the reviewer for their thoughtful review. In particular, their recognition of our writing, proposed method, rigorous experimentation, extensive ablations, and ImageNet-HR, as well as suggestions to improve the paper. The review lists two concerns, which we address in this rebuttal.
>
> $\textbf{Q1)}$ Is LookHere translation-equivariant?
>
> $\textbf{A1)}$ LookHere masks and biases are translation-equivariant w.r.t. patches. However, the model’s representations are not translation-equivariant due to patchification and edges. In our view, the motivation for translation-equivariant position encoding follows from the motivation for translation-equivariant convolutions — i.e., we want the feature map of an object or a region to be insensitive to its location within the image. This reasoning motivated CNN-ViT hybrids that only encode positions via convolutions [43], also achieving translation-equivariance.
>
> To show that LookHere is translation-equivariant, we may consider two arbitrary patches at P1(x1, y1) and P2(x2, y2); then LookHere’s contribution to the attention matrix from query P1 to key P2 is A. Now impose a patch-level translation T such that the two patches are at P1’(x1 + Tx, y1 + Ty) and P2’(x2 + Tx, y2 + Ty); then LookHere’s contribution to the attention matrix from P1’ to P2’ is B. Where A=B because LookHere’s two components depend only on the relative positions of P1' and P2', which are maintained under translation — i.e., masks are a function of the direction between query-key pairs, and biases are a function of the Euclidean distance between query-key pairs. A formal proof is straightforward from here, which we are happy to provide should the reviewer wish to see it. We thank the reviewer for this question and will elaborate on LookHere’s translation-equivariance in our main paper if it is accepted. We feel this may be valuable to readers.
>
> $\textbf{Q2)}$ Why not try newer adversarial attack methods?
>
> $\textbf{A2)}$ We chose FGSM because it is the most well-known and is extremely fast — allowing us to attack all 80 ViTs on 50k Val images, using multiple strengths in a reasonable amount of time. During this rebuttal period, we first considered AutoAttack’s official implementation but found that their fastest attack takes ~2 hours (on an RTX 4090) per model and strength. Alternatively, we test all 10 position encoding methods against PGD attacks during this rebuttal period. Averaged across 4 attack strengths, LookHere-180 / 90 / 45 outperforms 2D-RoPE by 3.5% / 2.8% / 4.3%, and 1D embeddings by 6.8% / 6.2% / 7.6%. Please see Table 1 in our rebuttal PDF for full results. We thank the reviewer for this question and agree that more sophisticated attacks will strengthen our claim of LookHere’s adversarial robustness — space-permitting, we will add these new results to the appendix or our main paper.
>
> The reviewer also asked about learning, rather than hard-coding, components of LookHere. In preliminary experiments, we found that learnable slopes did not improve performance at 224x224 px. During this rebuttal period, we re-ran this experiment and confirmed that learnable slopes perform roughly on-par with fixed slopes at 224x224 px and hurt performance when extrapolating to 1024x1024 px (see Table 3 in our rebuttal PDF). We thank the reviewer for this question and will add it to our ablations section.
>
> We did not try learning masks with LookHere, mainly because other learnable relative position encoding methods do not extrapolate well. For instance, RPE-learn could theoretically learn LookHere’s masks by biasing attention scores with large negative numbers whose entries become zero after the softmax. In preliminary experiments, we also tried another learnable RPE method that did not perform well; this method calculates the relative positions between query-key pairs and processes them with an MLP, which outputs attention biases (inspired by position encoding used in hierarchical ViTs [54]). However, we are excited for the community to build on LookHere and help us develop better position encoding methods — and we welcome learnable solutions!

---

> > ### Comment · Reviewer_cHUb · 2024-08-08
> > **Final Review**
> >
> > The authors address all concerns; no changes to my original rating of 8 (strong accept).
> >
> > Highly encourage authors to release results on stronger adversarial attacks at a later point in time as feasible.

---

### Official Review · Reviewer_U7cz · 2024-07-12

**Soundness:** 3
**Presentation:** 3
**Contribution:** 2
**Rating:** 6
**Confidence:** 4

**Summary:**

Vision transformer is known for its constrained scalability across various image resolutions and this work is designed to address the generalization ability of ViT at high-resolution images. Specifically, the authors propose LookHere, a drop-in replacement for the positional encoding of standard ViTs, which restricts attention heads to fixed fields of view oriented in different directions using 2D attention masks. It offers translation-equivariance, guarantees attention head diversity, and minimizes the distribution shift encountered by attention heads during extrapolation. Extensive experiments validate the effectiveness of the proposed method.

**Strengths:**

1. Evaluation is extensive. The authors provide validations across different datasets, and tasks, and offer detailed analysis to demonstrate the effectiveness of the proposed approach.

2. The results seem promising. LookHere exhibits the best performance compared to other baseline methods.

3. The presentation is clear and the paper is easy to follow.

**Weaknesses:**

1. The generalization ability of LookHere to smaller-scale images remains unverified. Although the authors particularly focus on testing at high-resolution images, it would be better if LookHere could be generalized to smaller images as well.

2. The ablation study on the design choices seems missing. Although the authors provide abundant experiments and analysis, the ablation study on the specific designs is missing and their efficacy is unverified.

3. The size of the proposed ImageNet-HR is too small (5 images per class). The limited number of images in each class may not be representative enough and could have a strong bias implicitly.

**Questions:**

Apart from the questions in weakness, the reviewer has one additional question:

Is there a particular reason that the authors did not compare LookHere with NaViT, which can also be tested at higher resolutions?

---

> ### Author Rebuttal · Authors · 2024-08-07
>
> We thank the reviewer for their thoughtful review. In particular, their recognition of our extensive experiments, LookHere’s performance, and our submission’s presentation, as well as suggestions to improve the paper. The review lists three concerns, which we address in this rebuttal.
>
> $\textbf{Q1)}$ Does LookHere generalize to smaller images?
>
> $\textbf{A1)}$ Our main goal is to improve ViT performance on larger images, which is an exciting challenge with great potential impact, given the long trend of applying computer vision methods to higher-resolution imagery that contain more detailed scene information. Although deploying models on smaller images will hurt performance, some users will benefit from reduced computational costs. We thus thank the reviewer for raising this point and perform new evaluations to address it. We test all 10 position encoding methods on images ranging from 64x64 px to 208x208 px and all sizes in between (at 16 px increments). LookHere outperforms all baselines at every size, averaged across all 6 datasets. In particular, LookHere-45 outperforms 2D-RoPE by an average of 4.1% (top-1 accuracy) at 64x64 px. We display these 6 plots in Figure 1 of our rebuttal PDF, and we will add this Figure to the 10th page of our main paper, should our submission be accepted.
>
> $\textbf{Q2)}$ Is there a missing ablation section?
>
> $\textbf{A2)}$ We believe the reviewer missed our extensive ablations in the appendix. On page 25, we report the performance of 18 ablation runs. Specifically, we report performance on all 6 benchmarks at 224x224 px and 1024x1024 px. In our main paper, we only had the space to summarize the results of these experiments; please see lines 171 - 180. We will move some of these ablation results to our main paper if accepted. Our ablations demonstrate that our novel 2D direction masks and distance penalties are crucial to extrapolate effectively — and that LookHere is robust to both 2D direction mask types and slopes. We believe these ablations are a strength of our submission. Additionally, during this rebuttal period, we ran a 19th ablation (experimenting with learnable slopes), which reviewer cHUb enquired about.
>
> $\textbf{Q3)}$ Is ImageNet-HR too small?
>
> $\textbf{A3)}$ Our introduced ImageNet-HR dataset consists of 5k total images, with 5 images per ImageNet class; this is smaller than most ImageNet benchmarks. However, we do not believe it is too small.
>
> First, although ImageNet-Val is substantially larger, we note that ImageNet-A and ImageNet-v2 comprise 7.5k and 10k images, respectively. Both are widely used (1K+ citations each). Second, we focused on quality over quantity, agreeing with other experts in the field; for instance, Lucas Beyer publicly encouraged the creation of “small, very high-quality test data [R1].” Although ImageNet-Val comprises 50k images, it contains significant annotation error and label ambiguity [4, 105]. For instance, one study estimates that 44% of the “mistakes” made by a ViT are actually correct predictions on mislabeled samples [R2]. Labeling quality has not been thoroughly studied for v2; however, we have observed annotation error and ambiguity via visual inspection. For ImageNet-HR, we manually selected and cropped high-quality and diverse images from Unsplash and flickr. Several rounds of quality control further reduced annotation error and ambiguity. Finally, to demonstrate that performance on ImageNet-HR is a meaningful measure of a model’s capability, we measure the correlation between ImageNet-HR / v2 and ImageNet-ReaL (a re-annotated version of ImageNet-Val). Across 240 paired evaluations, ImageNet-HR top-1 accuracy is highly correlated with ImageNet-ReaL top-1 accuracy ($R^2$=0.997), even more so than between ImageNet-v2 and ImageNet-ReaL ($R^2$=0.983). Please see Figure 3 of our rebuttal PDF.
>
> We believe we’ve addressed all concerns raised by the reviewer. The reviewer also asked why we do not compare LookHere to NaViT, which we answer below.
>
> NaViT is primarily a recipe to efficiently train ViTs on images with variable aspect ratios by processing non-square grids, i.e., variable grid-height and grid-width. It achieves this by “packing” multiple images into a sequence, preventing attention between images, and then creating a batch of these sequences to process. We believe this recipe, along with other innovative ViT training recipes, may be complementary to LookHere. Furthermore, NaViT does not see improved performance when testing on longer sequences than it saw during training (see Figure 10 b in [6]) — whereas LookHere does see improved performance when extrapolating. NaViT also introduces additive factorized position encoding — which is one of the seven baselines in our submission. We thank the reviewer for this question, and we will add some of this text to our main paper, should it be accepted, since we feel it may be valuable to readers.
>
> Based on the reviewer’s question, we examined whether LookHere can also generalize to non-square images like NaViT. During this rebuttal period, we tested the 10 position encoding methods on ImageNet-Val by resizing the largest dimension to 384 and maintaining the native aspect ratio for each image. LookHere generalizes the most effectively to non-square images (see Table 2 in our rebuttal PDF).
>
> [R1] Twitter: https://tinyurl.com/4azematv
>
> [R2] “When does dough become a bagel? Analyzing the remaining mistakes on ImageNet”, Vasudevan et al., NeurIPS 2022

---

> > ### Comment · Reviewer_U7cz · 2024-08-13
> > **Post-rebuttal comment**
> >
> > The reviewer appreciates the detailed response provided by the authors. The concerns are addressed and the rating is upgraded to weak accept.

---

### Author Rebuttal · Authors · 2024-08-07

We thank all reviewers for their thoughtful comments and are pleased with the positive reception of our submission. In particular, we appreciate the recognition of our writing, presentation, and extensive experiments from all reviewers. Additionally, we appreciate the suggestions to improve our paper. Should our submission be accepted, we believe we will have space for these additions on a 10th page. There were no shared concerns among reviewers, so we formulate our rebuttal as replies to specific reviewers.

During this rebuttal period, we ran additional experiments that reviewers enquired about. These results can be found in a 1-page PDF attached to this comment.

---

### Decision · Program_Chairs · 2024-09-25

**Decision:**

Accept (poster)

**Comment:**

In the discussion period, the authors addressed the concerns of the reviewers and the reviewers are happy to accept this paper. The AC also enjoyed reading this paper and the proposed idea of directionally limiting the fields of view to better process high-resolution images.